# Fluid-injection-induced earthquakes characterized by hybrid-frequency waveforms manifest the transition from aseismic to seismic slip

Hongyu Yu [1,2]✉, Rebecca M. Harrington[2], Honn Kao [1,3]✉, Yajing Liu[4] & Bei Wang [1,3]

Aseismic slip loading has recently been proposed as a complementary mechanism to induce moderate-sized earthquakes located within a few kilometers of the wellbore over the time-scales of hydraulic stimulation. However, aseismic slip signals linked to injection-induced earthquakes remain largely undocumented to date. Here we report a new type of earthquake characterized by hybrid-frequency waveforms (EHWs). Distinguishing features from typical induced earthquakes include broader P and S-pulses and relatively lower-frequency coda content. Both features may be causally related to lower corner frequencies, implying longer source durations, thus, either slower rupture speeds, lower stress drop values, or a combination of both. The source characteristics of EHWs are identical to those of low-frequency earthquakes widely documented in plate boundary fault transition zones. The distribution of EHWs further suggests a possible role of aseismic slip in fault loading. EHWs could thus represent the manifestation of slow rupture transitioning from aseismic to seismic slip.

[1] Geological Survey of Canada, Sidney, BC V8L 4B2, Canada. [2] Institut für Geologie, Mineralogie und Geophysik, Ruhr-Universität Bochum, Bochum 44801, Germany. [3] School of Earth and Ocean Sciences, University of Victoria, Victoria, BC V8W 3V6, Canada. [4] Department of Earth and Planetary Sciences, McGill University, Montréal, QC H3A 0E8, Canada. ✉email: hongyu.yu@rub.de; honn.kao@canada.ca

Industrial fluid injection related to unconventional oil and gas production induces earthquakes[1,2]. The most common perception is arguably that M4+ events generally result from large fluid volumes related to wastewater disposal, particularly in the Central and Eastern United States[1]. However, induced earthquakes associated with hydraulic fracturing (HF) injection have recently challenged the conventional wisdom[3] by successively generating larger and larger maximum magnitude earthquakes. For example, three M4.5+ HF earthquakes occurred since 2015 in the Western Canada Sedimentary Basin (WCSB)[4–7], as well as the 2017 $M_W$ 4.7, 2018 $M_L$ 5.7, 2019 $M_L$ 4.9, and 2019 $M_L$ 5.3 earthquakes in the Sichuan Basin, China[8–10]. Studies suggest many M3+ events near HF wellbores are generated on seismogenic faults that are critically stressed, where injection facilitates rupture by shifting the stress state toward failure through pore pressure increase[11,12] or poroelastic stress transfer[10,13,14]. However, geomechanical considerations often make it challenging to explain M3+ HF-induced earthquakes with the classical concepts of pore pressure and poroelastic stress change. For example, some observations show migration speeds of induced seismicity that outpace pore pressure diffusion[15]. Moreover, the relatively small volumes of total injected fluids are often likely insufficient or unable to generate significant poroelastic stress perturbations[6,15].

An alternative, or complementary interpretation that has emerged from recent modeling and experimental results are that injection can initiate aseismic slip in the weak, fractured volume proximal to the wellbore, which then further transmits stress to surrounding (pre-existing) faults[15,16]. Laboratory and experimental work, including larger mesoscale experiments, have shown that loading can generate a continuum of slip behavior ranging from aseismic to seismic, and in some cases, aseismic slip can transition to seismic slip within a single slip episode[17,18]. Under the optimal geomechanical (e.g., critically stressed host fault) and compositional conditions (e.g., high clay and total organic carbon content, 19), aseismic slip fronts may interact with nearby, larger faults to trigger significant events[15]. In addition, modeling studies of aseismic slip in fluid injection environments have quantitatively validated laboratory aseismic slip observations, proving that (a) the aseismic slip front could outpace the fluid diffusion front[16,20], and (b) a limited amount of aseismic slip is sufficient to trigger larger magnitude events in cases where it interacts with nearby, larger faults under effective dynamic weakening, such as thermal pressurization[15,21]. However, to the best of our knowledge, direct field observations of aseismic or slow-slip signals prior to the onset of seismic slip have not been clearly documented outside of experimental injection environments. One reason might be that slow-slip-induced seismic signals are likely low-amplitude, similar to tectonic tremors observed at the up- and downdip ends of seismogenic zones[22]. Detecting such low-amplitude signals would require dense arrays of sensitive instruments in close proximity to injection wells, which are only recently becoming more commonplace for scientific purposes.

Using a high-density seismic array surrounding an active HF well in the Montney Shale formation, British Columbia, Canada, we report a new type of seismic signal that may represent slow rupture related to HF injection (Fig. 1). Qualitatively, the observed waveforms consist of two portions with differing frequency content compared to the typical induced events of comparable magnitude occurring in the same area (Fig. 1b–e): (1) an impulsive broadband onset with visible P- and S-phase arrivals, but with slightly broader pulse widths, and (2) a coda with lower-frequency energy that follows the body-wave phases. We term the combined signals "earthquakes characterized by hybrid-frequency waveforms" (hereafter, referred to as "EHWs"). In the following, we document and quantify the distinctive features of the EHW

waveforms to identify plausible source mechanisms. We first check the correlation between the injection and the spatio-temporal distribution of the EHWs. Second, we test whether the EHW signals stem from source or path effects by analyzing the duration pattern of a low-frequency coda. After confirming that the EHWs originate from source processes, we provide evidence that EHWs result from slow ruptures through analyzing the moment-duration scaling and source features, including corner frequency, rupture speed, and static stress drop, based on the broadband portion of the waveforms. We also conduct coupled pore pressure and poroelastic stress modeling to infer the likely role of aseismic slip loading in inducing EHWs. Finally, we propose that the variation of source properties with distance from the well may be best interpreted as EHWs representing part of the continuum slip behavior ranging from aseismic slip to seismic rupture in a fluid injection environment.

## Results

**Distribution of EHWs.** The EHWs analyzed in this study were recorded at eight broadband seismograph stations deployed around an HF well pad from May 28 to Oct 15, 2015 (MG01-08, Fig. 2). The instruments were installed with the explicit purpose of capturing seismicity associated with HF treatments at one well site in the Montney Shale formation. We thoroughly inspect continuous waveforms to identify a total of 31 EHWs (Table S1; Data and Methods), all of which are confirmed to have occurred near the wellbore (Text S1; Fig. S1).

Among the 31 detected EHWs, 25 events are sufficiently well constrained for us to determine their precise hypocenters (Data and Methods). As shown in Fig. 2a, they can be roughly divided into three groups: (a) in close proximity to the horizontal wellbore (3 EHWs), (b) near station MG08 (5 EHWs), and (c) located ~2 km south of the end of the wellbore (13 EHWs, the "southern cluster"). EHWs from the southern cluster show high waveform similarity (Fig. S2). Their hypocentral distribution outlines a plane with strike and dip angles of 150° and 66°, respectively. The fitted plane is optimally oriented for reactivation in the ambient regional stress field (Fig. 2b). Similar high-angle dipping structures have also been reported recently in the northeast Montney Shale formation[6,23]. The average distance between each hypocenter to the fitted plane is 0.13 ± 0.10 km. The planar structure implied by the southern EHW cluster also intersects a group of 128 typical induced events[24] with an average distance of 0.27 ± 0.22 km to the fitted structure (Fig. 2a). The short distances to the fitted plane could be owing to the uncertainty of event hypocenters, or the presence of a diffuse deformation zone.

All of the EHWs and typical induced events in the study area shown in Fig. 1 are likely related to a longer injection history. Here, we also apply a classical spatiotemporal analysis to associate EHWs with HF stimulation at a specific well. As shown in Fig. 2c, well operations, including preparation, pressure testing, and HF, started on Jul 1, 2015 and lasted for 20 days (Table S2). The number of EHWs is the highest during this period[24], which is considered as the nearly instantaneous response to the injection. Subsequent EHWs migrated approximately following a hydraulic diffusivity of ~0.2 m²/s (see Data and Methods), which is consistent with estimations based on typical induced events in our study area[24]. The similar migration patterns of the two different types of events suggest that they may share common driving mechanisms. Further investigation of the role of diffusivity in controlling the spatiotemporal distribution of pore pressure and poroelastic stress changes is given in the Discussion under the scope of EHWs following the migration pattern.

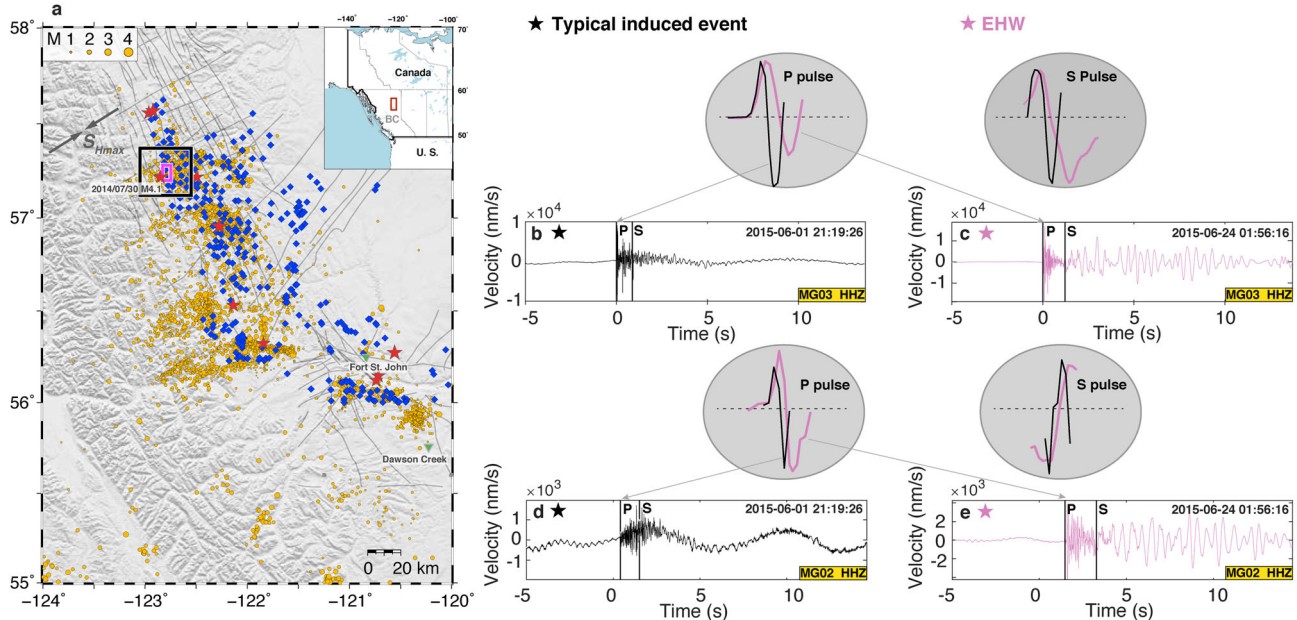

**Fig. 1 Study area and EHWs. a** Hydraulic fracturing activity and seismicity in northeast British Columbia, Canada. The top-right inset shows the geographic location of our study area (red rectangle). Blue diamonds: hydraulic fracturing injection wells between 1 Jan 2014 and Dec 31, 2016, reported by the British Columbia Oil and Gas Commission. Yellow dots: earthquakes during the same period[5]. Stars: 10 M 4–5 earthquakes since 2008[5,6]. Black rectangle: study area in which EHWs occurred. Pink rectangle: the area plotted in Fig. 2a. **b, c** Comparison between representative examples of a typical induced event (black trace) and an EHW (purple trace), marked in Fig. 2a. Both waveforms are from the vertical component of station MG03 with comparable source-station distance (1.32 km vs. 1.65 km) and magnitudes (~$M_W$1.5). Manually picked P/S-arrivals are marked. The comparison of P/S-pulse shape demonstrates the relatively wider pulses for EHWs (see text). **d, e** The same as **b** and **c** but for waveforms recorded at station MG02. Both events show longer coda durations with slightly larger epicentral distances. Note that although both types of events have extended coda durations, the EHW contains a relatively larger proportion of lower-frequency energy in the coda.

We also detect EHWs prior to well activity in July (Fig. 2c). EHWs preceding the July HF stimulation are suspected to be latent seismicity related to previous nearby injection activity (e.g., [25,26]). Specifically, HF injection was conducted at four wellpads between 2013 and 2014 (W1–W4 in Fig. 2b; Table S3). Among them, W1 was operational in September 2013 and is co-located with the monitored well pad of this study. W3 and W4 appear to be close to the cluster near MG08 and the southern cluster, respectively. Several EHWs that were nearly concurrent with the onset of the July HF stimulation but located at ~3 km from the well (Fig. 2c) also imply a previously critical stress state in our study area. In fact, the surge of seismicity in northeastern BC since 2008 is largely associated with the drastic increase of HF injection (Fig. 1a)[2]. Therefore, from a regional perspective, all the detected events in our study area can be considered injection-related. Consequently, it is rational to assume that all detected EHWs have a similar seismogenic origin and to study them as one group. Specifically, we only consider migrating EHWs within 4 km from the instrumented wellbore and within 100 days following the onset of HF injection as reliably linked to the well here (Fig. 2c). The spatiotemporal limit is chosen to reflect the maximum range of stress perturbation expected from HF injection in our study area, according to ref. [24].

**EHW source mechanism.** Seismic signals characterized by similar hybrid-frequency energy have been reported as "volcanic hybrids" in volcanic environments. Two broad interpretations have been proposed to explain their origin[27–34]. The first rests on a number of studies that interpret the low-frequency portion of the waveform as resulting from either low rupture velocities and/

or travel path through shallow low-velocity material (i.e., path effects)[27–31]. The second ascribes the low-frequency coda to the coupling between turbulent flow and the walls of an open crack[32–34]. The coda duration in the latter interpretation is thought to be dictated by the pressure gradient across the crack and is independent of the duration of brittle failure. Therefore, verifying the origin of the extended low-frequency coda is potentially the key to identifying the physical mechanism(s) of the EHWs.

To measure the low-frequency portion of the EHW waveforms, we define the coda duration, $T_a$, as the time required for the amplitude envelope of the low-frequency portion to decay to $e^{-1}$ of the peak value (Fig. S3). We observe that $T_a$ can vary from one station to another depending on the event and station locations (Fig. S4; see Data and Methods). The observed $T_a$ variation cannot be explained by source radiation pattern (Fig. S5) or site effects. Instead, it is consistent with travel path effects associated with the very nature of injection into a low-permeability shale formation. Namely, hydraulic stimulation creates localized fracture networks, increases fluid overpressure, and thus increases attenuation effects for rays passing through such regions. At the same time, the fine-scale fluid-pressurized structures would enhance the dispersive effect by disproportionately attenuating higher frequency energy. The seismic wave traveling through a heterogeneous, fluid overpressured rock matrix would therefore likely have a longer coda relative to that across the less fractured rock.

We, therefore, hypothesize that the variation of $T_a$ stems from velocity heterogeneities along the travel path. That is, if the ray path extends primarily through the low-velocity heterogeneity, its coda duration would be protracted, and vice versa. The distribution of $T_a$ consistently suggests two volumes of low-

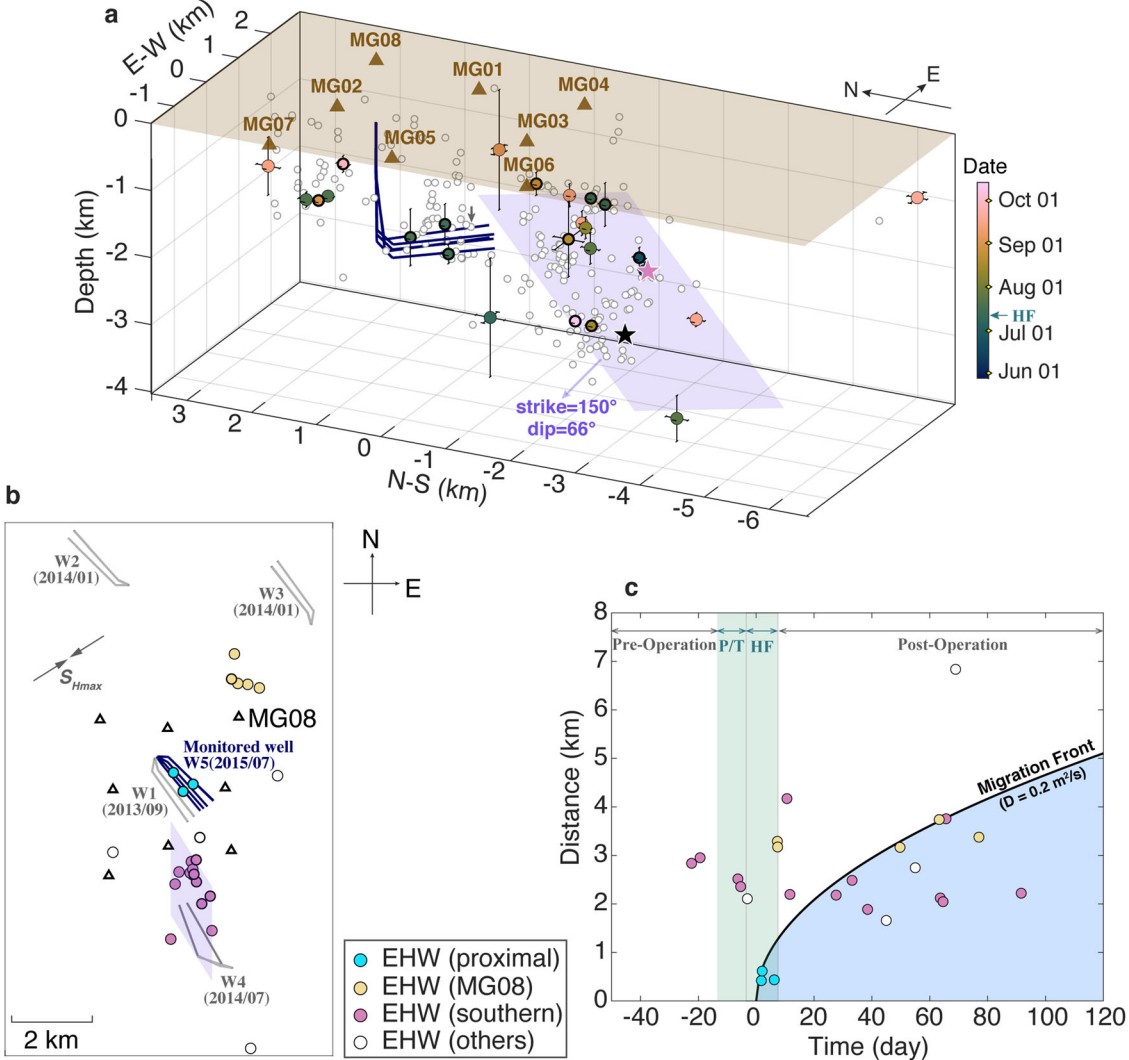

**Fig. 2 Distribution of EHWs. a** Spatial distribution of EHWs. Dots: 25 relocated EHWs. Dot color refers to the origin time. Black/Purple star: representative example of typical induced event/EHW in Fig. 1b–e. Purple shaded plane: least-squares plane fit of the southern EHW cluster. Thick blue lines: HF wells. Gray circles: 285 typical induced events near the wellbore[24]. **b** Locations of three EHW clusters in map view, the well pads and horizontal well trajectories of the monitored well (W5, blue lines) and four nearby wells (W1–W4; gray lines) of HF injections during 2013–2015. **c** Migration of EHWs. Parabolic envelopes: the "diffusion front" calculated using diffusivity values of 0.2 m²/s. The reference time ($t = 0$) here is 13 July 2015, 10:52:22 (origin time of the first near-well event following the injection[24], marked in **a** with a small gray arrow). Blue shaded areas: EHWs following the migration pattern. Green shaded areas: preparation/pressure testing (P/T) and hydraulic fracturing (HF) periods.

velocity heterogeneity surrounding the wellbore (Fig. S4, see more details in Data and Methods). Furthermore, an overall positive correlation exists between the hypocentral distance and $T_a$ (see example for the farthest station, MG08, in Fig. S6). The coda durations of typical induced events are similarly prolonged when their ray paths pass through a significant portion of low-velocity heterogeneities (e.g., Fig. S4d), although the frequency content of their coda is not as low as those of EHWs (e.g., Fig. 1d, e). In summary, the prolonged coda duration of both EHWs and typical induced events likely results from dispersion rather than pressure gradient-driven fluid flow in a crack. The relatively lower-frequency content of the EHW coda, on the other hand, seems to be a manifestation of source physics. In the following, we try to delineate its source physics based on the broadband portion of EHWs.

We estimate the source properties of seismic moment ($M_0$) and spectral corner frequency ($f_c$) based on the spectra of the broadband onsets (See Data and Methods). We find the EHW spectral corner frequencies ($f_c$) are discernibly lower than those of

typical induced events within the same magnitude range and same source-station distances obtained from the same data set (Fig. 3; [35]). Fig. S7 shows representative examples of the spectral characteristics difference between the two types of events. These examples also demonstrate that the range of corner frequency values for different types of events is resolvable with our data set and represents real differences between event types. Considering source duration as the reciprocal of $f_c$ for the broadband onset (0.05–0.14 s), Fig. 3 shows that the moment-duration scaling of 15 robustly constrained EHWs likely follows the scaling expected for typical (fast-rupture) earthquakes ($M_0 \propto T^3$). Although the limited magnitude range ($M_W \sim 0.5$–2.0) does not allow us to completely rule out a linear moment-duration scaling ($M_0 \propto T$) that is commonly inferred for a broad range of slow-slip phenomena[36], the root-mean-squared (RMS) error is 86% smaller when fitted with the $M_0 \propto T^3$ seismic rupture scaling versus the $M_0 \propto T$ scaling. Therefore, it is rational to apply a circular crack rupture model to further discuss the source features of EHWs[37–39].

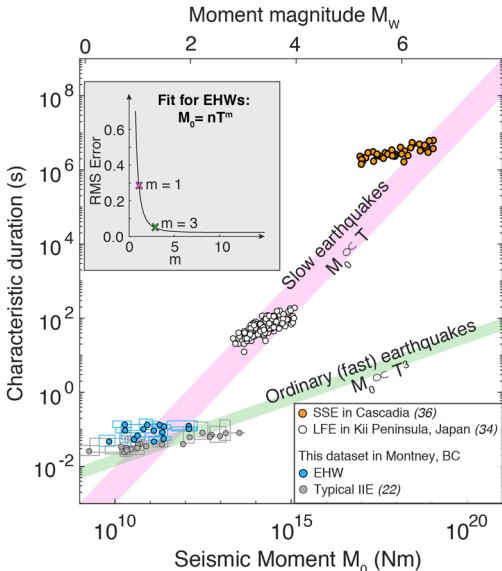

**Fig. 3 Source parameter estimates of the broadband onset.** Pink/green shaded bar: proposed moment-duration scaling for slow and typical (fast) earthquakes[36]. Blue/gray dots with rectangles: EHWs/typical injection-induced events in this study. Horizontal uncertainty: estimated from the standard error of low-frequency amplitude variance. Vertical uncertainty: reciprocal of frequency band with 5% increase of fit variance. White dots: low-frequency earthquakes in Kii Peninsula, Japan;[37] orange dots: slow-slip events in Cascadia[39]. Inset: fit for moment-duration scaling of EHWs assuming $M_O = nT^m$. The RMS error decreases 86% when the exponent of T increases from $m = 1$ (pink cross) to $m = 3$ (green cross), respectively.

**EHW rupture characteristics.** We estimate the rupture characteristics of EHWs under the assumption of a circular crack model with a constant rupture speed[40]. The source dimension (source radius $r$) would be the product of rupture speed ($v_r$) and source duration ($T \approx 1/f_c$). A longer source duration could be related to a larger source dimension (i.e., a lower stress drop value), a lower rupture speed, or both. In other words, with robust $f_c$ estimates only, it is beyond the scope of this study to determine the value of the source dimension (and hence stress drop) or the rupture speed individually. However, our previous study constrained source features of collocated typical induced events from the same data set[35]. By referencing these events, we are able to discuss the upper and lower bounds of EHW source parameters in the following two scenarios.

In Scenario 1, we assume EHWs have rupture speeds identical to typical induced earthquakes. By using the same constant $v_r$ ($= 0.9v_s$)[35], we obtain the maximum estimates of the EHW source dimensions and the corresponding minimum estimates for stress drop values (see Data and Methods). We then compare the difference between the stress drop values of EHWs and typical induced earthquakes detected in the same area. Figure 4a shows that EHWs exhibit an average stress drop value of 0.29 MPa (ranging from 0.02 to 1.08 MPa), compared with 4.86 MPa for typical events from the same field test[35]. The stress drop values of EHWs near the wellbore are lower compared to those at greater distances, which is consistent with the trend of typical induced earthquakes[35]. Moreover, stress drop values of the more distant clusters (near MG08 and the southern cluster) show a wider range between 0.03 and 1 MPa, suggesting a wider slip continuum.

In Scenario 2, we assume the EHWs have the same stress drop as typical induced events with similar seismic moments. Under this assumption, we are able to constrain the lower bound of the

rupture speed for EHWs (See Data and Methods). Figure 4b shows that EHWs exhibit an average $v_r$ value of 1.13 km/s (ranging from 0.25 to 1.96 km/s), compared with a roughly uniform value of 2.34 km/s for the typical induced earthquakes. The rupture speeds of EHWs are comparable to that of low-frequency earthquakes (LFEs; on the order of 1 km/s), which are tectonically driven slow earthquakes in fault transition zones[36]. Similarly, rupture speeds of distant EHW clusters are generally higher than the EHW near the wellbore, and show a wider range of rupture speeds, suggesting a transition from slow-slip to seismic slip.

## Discussion

We observe that the EHWs exhibit evidently longer source durations than the typical induced earthquakes (Fig. 3), albeit over a limited magnitude range of $M_W \sim 0.5$–2.0. Admittedly, the absolute source duration values could vary depending on the approach and parameters applied in the estimation. However, the relative differences between these two types of event, estimated using the same data set and methodology, should be robust. The difference is also consistent with the first-order observation of relatively broader P and S-pulses, and lower-frequency coda content of the EHWs (Fig. 1). Lower stress drop values of fluid-induced earthquakes have been widely observed and explained by many factors. For example, relatively shallow focal depths[41–43], in situ fluid-induced effective stress change[44], and heterogeneous slip within rocks with variable mechanical properties[15,19] may all be viable explanations in certain settings. Studies with improved data resolution report a positive relationship between the stress drop and event-to-well distance as the injection-induced stress perturbation decreases with distance[35,45,46]. However, this is not the case with our observation of co-located EHWs and typical induced events (Fig. S7). The lower $f_c$ values of EHWs compared with those of co-located typical induced events determined using the same data set and estimation method in the same study area makes it difficult to explain the observations with pore pressure and/or poroelastic stress changes alone. Thus, the most compelling explanation in light of the relative source parameter differences is that EHWs are the manifestation of a source process with lower stress drop and/or slower rupture speed that is fundamentally different from that of typical induced events.

To evaluate the role of different physical processes related to the HF injection in generating EHWs, we test the null hypothesis that classical concepts of coupled pore pressure and poroelastic stress change are sufficient to account for the distribution of EHWs. We model the evolution of coupled pore pressure and poroelastic stress caused by HF injections, using the COMSOL Multiphysics software (see details in Data and Methods). In our layered model (Fig. 5a), we embed a 300 m-thick shale layer to represent the Montney shale formation. To match the migration of seismicity, we assume the permeability within the shale layer to be $1 \times 10^{-14}$ m[2], equivalent to a hydraulic diffusivity of 0.2 m[2]/s. The value is much higher than reported for tight shale formations ($10^{-23} \sim 10^{-16}$ m[2])[47,48], indicating that the southern cluster and cluster near MG08 are probably hydraulically connected to the volume near wellbore[3,6]. Our simulation applies industry-reported injection information in each HF stage (Table S2). To evaluate the triggering capacity for microseismicity, we require the Coulomb failure stress change ($\Delta CFS$) to be at least $\sim 0.1$ bar[49–51]. Similar values are used by previous dynamic triggering studies in the WCSB[52,53].

We specifically look into the evolution of $\Delta CFS$ at three locations: near the horizontal wellbore, at the center of the planar structure fitted to the southern cluster, and near station MG08 (Fig. 5b–d). By adopting a diffusivity value of 0.2 m[2]/s (Fig. 2c),

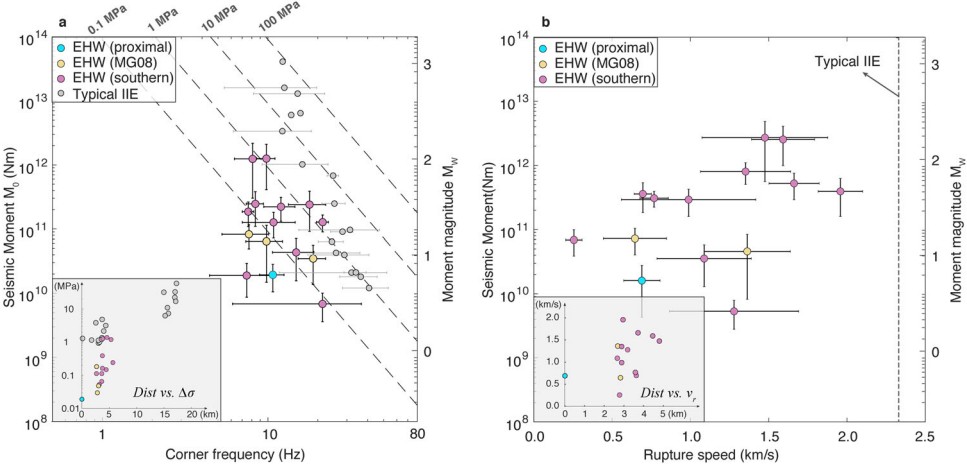

**Fig. 4 Rupture characteristics of EHWs. a** Scaling between corner frequency $f_c$ and seismic moment $M_O$. Colored/gray dots: $f_c$ and $M_O$ estimated based on spectral ratio fitting of S-phase for EHWs/typical induced events constrained using the same approach and data set. Error bars are consistent with Fig. 3. Inset: stress drop as a function of well distance. **b** Estimated rupture speed of EHWs. The dots and error bars are defined same as in **a**. Inset: rupture speed as a function of well distance.

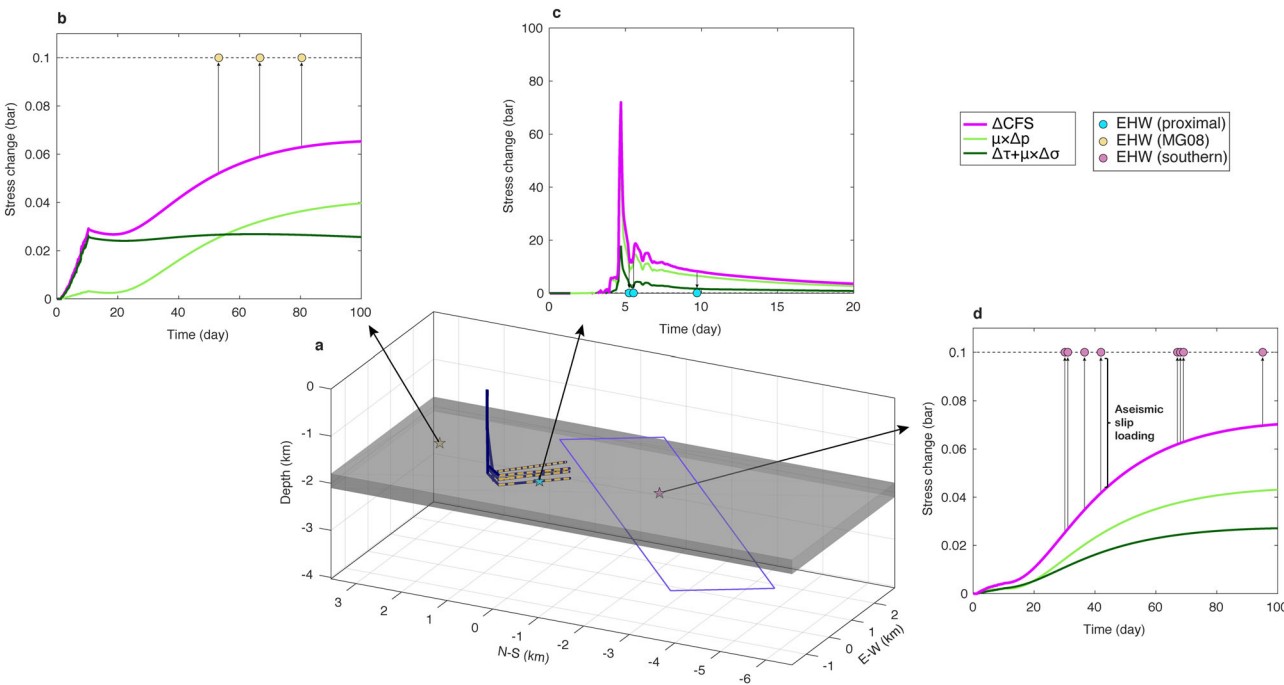

**Fig. 5 Evolution of coupled pore pressure and poroelastic stress change. a** Sketch of model. Yellow dashed line: location of injection intervals. Gray surface: shale layer. Purple plane: fitted structure in Fig. 2a. Yellow/Blue/purple star: representative locations near station MG08/wellbore/at the center of the fitted plane. **b–d** Stress evolution of example locations. Note that the light and dark green curves show the ΔCFS caused by pore pressure ($\mu\Delta p$) and poroelastic stress ($\Delta\tau + \mu\Delta\sigma$) changes, respectively. Dots: occurrence times of EHWs beneath the diffusion envelope illustrated in Fig. 2c (blue shaded area). The arrow lines connecting the pink curve (ΔCFS) to individual circles indicate the discrepancy between the calculated ΔCFS and the triggering threshold, which could be attributed to aseismic slip loading.

the coupled pore pressure and poroelastic stress change lead to a wellbore-proximal ΔCFS of ~10 bars, which is sufficient to induce EHWs (Fig. 5c). At further distances, near the station MG08 (Fig. 5b) or the source location of the southern EHW cluster (Fig. 5d), it takes ~50 days for ΔCFS to reach 0.05 bar because of the slow build-up of pore pressure change. The ΔCFS thereafter may be eventually capable of triggering the further EHW clusters. However, the four EHWs that occurred within the first 50 days are unlikely to be induced by the low values of ΔCFS

(0.02~0.04 bar). Our modeling results thus indicate the coupled effect of pore pressure and poroelastic stress change is insufficient to induce those "early" EHWs at greater distances, even if an abnormally high permeability is assumed to allow the relatively fast fluid diffusion. An alternative mechanism(s) is required to make a comparable contribution to the ΔCFS.

Aseismic slip loading could be a viable mechanism to explain the early occurrence of EHWs at greater distances (Fig. 5d). As the active pumping would lower the effective normal stress

through an increase of pore pressure, it could possibly reactivate the fault/fractures near the wellbore or a structure that is hydraulically connected to the volume near the wellbore (e.g., the fault outlined by the southern cluster). As slip accelerates, dilatancy accompanied by shear deformation could generate newly fractured rock volume[17,54]. The pores dilate and the absence of fluid flow would temporarily reduce the pore pressure, and hence increase the effective normal stress. Consequently, the strengthened fault tends to hinder, or even inhibit slip acceleration that, in turn, creates a mechanical condition in favor of aseismic/slow slip. On one hand, an aseismic slip would load adjacent areas along the fault[15,17]. With an injection rate of ~9 m$^3$/min (Table S2), the pressure front of aseismic slip loading could propagate at twice the rate of fluid diffusion[20], which is consistent with the occurrence of the "early" EHWs (Fig. 5d). On the other hand, as the increased pore space is connected, the enhanced permeability could further accelerate the migration of fluid diffusion[20,55]. Both effects could help shorten the time scale of a build-up in stress perturbation.

Moreover, the build-up in ΔCFS along the plane outlined by the southern cluster also favors the nucleation of seismic events, including EHWs. The fault is stable in the early stage when the stress perturbation is insufficient to overcome the residual fault strength. As the stress perturbation further loads the fault to overcome the residual fault strength, the fault would eventually slip seismically[20,56].

EHWs that occurred before the July 2015 HF injection (mainly from the southern cluster) are likely latent seismicity (Fig. 2c). Given the timing, they cannot be interpreted to have been triggered by pore pressure diffusion or poroelastic stress transfer from injections. Rather, we propose that aseismic slip driven by fluids from prior injections (W1–W4) may play a role. The fluids trapped in fault zones within low-permeability formations could retain a localized, elevated stress state for periods of months to years[26,53]. The altered stress state may help generate aseismic slip to repeatedly load neighboring unstable areas along faults, and thereby lead to latent EHWs/typical induced events occurring at relatively steady rates[25,26]. The localized elevated stress-state scenario may also work for EHWs that occur behind the curve of the migration front (Fig. 2c), as they could be the on-going process of either latent EHWs or the earlier triggered events.

EHWs with distinctly longer source durations may represent the seismic signature in a continuum of rupture speeds between aseismic slip to seismic rupture. Recent results of both laboratory and mesoscale experiments show that aseismic slip gradually transitions to seismic slip along faults near the wellbore with a continuum of slip speeds[17,18]. Consistent with laboratory observations, we report that the stress drop and rupture speed of EHWs along with the fitted southern plane exhibit a wide range of characteristic rupture behavior (Fig. 4). These source features, including lower stress drop values and slower rupture speeds, are commonly interpreted to represent the transition from aseismic to seismic regimes in a tectonic environment[36]. Furthermore, the $M_0 \propto T^3$ source scaling of EHWs is similar to that inferred from individual groups of LFEs[37,38] and from more recently revised moment and duration estimates of Cascadia slow-slip events (SSEs)[39]. Linking the geodetically detected SSEs across a broad magnitude range to the seismically observed EHWs (i.e., from $M_W$ 0 to 6, Fig. 3b) implies that EHWs also appear to be compatible with the linear moment-duration scaling ($M_0 \propto T$)[36]. The common features between EHWs and tectonically driven slow-slip phenomena also suggest that EHWs may bridge aseismic and seismic slip near injection wells.

Finally, from the perspective of mitigating seismic hazards due to fluid injection, EHWs might provide the first clue to how injection operations can be controlled to keep sliding aseismic.

The purpose of HF stimulation is to enhance the permeability of tight shale layers by enlarging the surface area of the fracture network. In this context, the sourcing process of EHWs can be more efficient and safer in accomplishing the objective of HF stimulation owing to their relatively larger rupture area and slower rupture speed (Fig. 4), compared to typical induced events of similar magnitude. If injection commonly first induces aseismic slip near the wellbore, EHWs should occur on a widespread basis, and are likely not limited to the Montney shale formation. One possible hurdle to identifying EHWs is that the characteristic low-frequency, low-amplitude coda will be quickly buried by body-wave dispersion and coda noise when recorded at regional distances from injection wells. Indeed, there are emerging reports of long-period transients similar to EHWs at stations in close proximity to injection[57,58]. We expect an increase in the reports of EHW signals with increasing seismic observations at closer distances to injection wells.

## Data and methods

**Data.** The EHWs analyzed in this study were recorded at eight broadband seismograph stations equipped with Nanometrics Trillium Compact 20 s sensors (MG01-08 in Fig. 2) deployed 1–3 km from an HF well pad from May 28 to Oct 15, 2015. In addition to 350 well-constrained hypocentral locations of typical induced earthquakes (with waveforms resembling standard, tectonic earthquakes)[24], the detection procedure yielded a catalog of 31 EHWs (Table S1). The HF injection at the instrumented well pad lasted for 10 days from Jul 11 to Jul 20, 2015 (Table S2).

**EHW detection.** The EHW event detection is performed in two steps. The first uses an automated, recursive STA/LTA detection module in the ObsPy toolbox[59]. We bandpass filter continuous waveforms between 10 and 20 Hz and set the length of short- and long-time windows to be 0.5 s and 10 s, respectively. Trigger-on and trigger-off thresholds of the STA/LTA ratios are 3.5 and 1, respectively, and we require a coincidence sum larger than three stations to claim an event. The initial detection step identifies a variety of seismic signals, including induced and natural earthquakes. We then visually identify EHWs as a cluster of events with waveforms showing the following two characteristics: (1) an impulsive broadband onset with visible P- and S-phase arrivals, but with slightly broader pulse widths compared to typical induced events in the same area, and (2) a sustained lower-frequency coda wave (~10 seconds; <5–7 Hz) after the body-wave phases.

Next, we use EHW detections from the first step as template events in a Multi-station Matched-Filter (MMF) approach to detect additional EHWs. We cut template waveforms in time windows 3 s before and after the S-arrival so that both P- and S-phases are included on the east component of MG stations. We use the east component because it has a better SNR and thus optimizes the detection compared to using all three components. Both the templates and continuous waveforms are bandpass filtered between 5 and 15 Hz to assist visual identification of local seismic signals[60]. After cross-correlating (CC) templates with continuous waveforms at each respective station, we sum the normalized CC values. To help detect earthquakes in the vicinity of the template events with slightly different locations, we weaken the assumption of co-location between the template and detected events by allowing up to 0.5 s shift in calculating the summed CC values to maximize the CC sum. A detection is declared when the sum exceeds 16 times the mean absolute deviation, with a corresponding probability of exceedance equal to $1.9 \times 10^{-27}$, equivalent to $8 \times 10^{-21}$ events/day being detected by random

chance[61,62]. We refer readers to ref. [24]. for further details of the MMF detection.

**EHW location**. We perform double-difference relocations (hypoDD)[63] of the initial locations constrained with Hypocenter[64]. Both location procedures use the Crust 1.0 velocity model with 1°-by-1° cell centered at (57.5°N, 122.5°W)[65]. The relocation calculation uses both manually picked and cross-correlated phase arrivals at all MG stations (with timing correction applied). We include the relative travel time differences between EHWs and typical induced events from the same data set[24]. There are 25 EHWs (of the original 31 detections) that surpass an error cutoff threshold of 0.5 km and 1 km for horizontal and vertical errors, respectively. Relocation uncertainty estimates stem from a bootstrap approach based on 100 trials[66,67].

**EHW migration**. Assuming isotropic hydraulic diffusivity ($D$) for the bulk crustal rock near the HF well, the relation between hypocenter distance ($r$) and occurrence time ($t$) follows a diffusional parabolic envelope, $r = \sqrt{4\pi Dt}$[68] We set the origin time and hypocenter of the first detected induced event following the commencement of injection as $r = 0$, $t = 0$[24] As shown in Fig. 2c, 59% of EHWs follows the diffusivity value of ~0.2 m²/s, which is consistent with the estimation based on typical induced events[24]. A few EHWs located beneath the station MG08 indicate a diffusivity value of ~1.5 m²/s. A cluster of typical induced events near MG08 also initiated shortly after the commencement of injection[24].

**Low-frequency coda of the EHW: seemingly a result of path effects**. The distribution of the coda duration $T_a$ at individual seismic stations roughly follows two patterns that are consistent with the influence of path effects (Fig. S4). As shown by a representative EHW that occurred proximal to the wellbore, MG03 consistently records the longest $T_a$, MG05 and MG07 intermediate $T_a$ values, and stations MG01, MG06, and MG08 record the shortest $T_a$. Similarly, events in the southern EHW cluster exhibit the longest $T_a$ on MG01, MG02, MG03, intermediate on MG07, MG04, MG05, and MG08, and the shortest on MG04.

We first consider the role of the source radiation pattern in determining the distribution of $T_a$. Because the EHWs are all small in magnitude ($M < 2$), it is not possible to robustly constrain focal mechanism solutions with these surface stations. As such, the observable effects of source directivity would likely also be negligible. To further explore the effects of radiation pattern on $T_a$, we perform a consistency check, in which the inconsistency of the azimuthal correlation between amplitude and $T_a$ rules out a significant influence of radiation pattern on $T_a$. For example, if we assume that the observed $T_a$ is indeed controlled by the focal mechanism, a positive correlation should exist between the $T_a$ and the radiated energy, as well as between the waveform amplitude and the radiated energy. That is, if we compare two stations at the same epicentral distance, the stations at azimuths receiving more radiated surface wave energy are more likely to record waveforms with larger amplitude and longer $T_a$. Under the above hypothesis, we should observe consistency between the $T_a$ and amplitude. We show the variation of $T_a$ and amplitude at nearby stations for two EHW events in Fig. S5. For the well-proximal EHW, MG01, and MG03 have similar distances and comparable amplitudes (38 nm/s vs. 34 nm/s, Fig. S5c), but values of $T_a$ are largely different (1.9 s. vs. 4.6 s, Fig. S5a). The anticorrelation invalidates the hypothesis that the $T_a$ is governed by the radiated energy. Similarly, for the EHW among the southern cluster, stations MG02 and MG05 show inconsistent results when we infer the amount of radiative energy from $T_a$ or

amplitudes (Fig. S5f). The discrepancy between waveform amplitude and $T_a$ for both EHW examples suggests that the focal mechanism, if not irrelevant, is not the key factor controlling the $T_a$ pattern (Fig. S5e, f). Site effects are also not the most likely justification for the measured variations in $T_a$, given that MG stations are all spaced within ~2 km of each other on similar geological material (Fig. 2a).

We then hypothesize that the $T_a$ variation stems from velocity heterogeneity along the travel path. To test the hypothesis, we construct a conceptual model using the distribution of ray paths and their coda durations as constraints to outline where lower-velocity/higher-porosity heterogeneities would be expected. We first document the $T_a$ of all the waveform records with SNR larger than 2 and calculate the ray path of these records by applying a layered velocity model[24,65]. The ray paths and observed $T_a$ suggest two individual velocity heterogeneities. As shown in Fig. S4a, one heterogeneous volume has dimensions of ~2.0 km × 0.5 km × 0.4 km and is located directly above and extends parallel along the horizontal wellbore direction (where ray paths allow no constraint at greater depth). The other heterogeneity has dimensions of ~3.2 km × 1.6 km × 0.8 km, at depths shallower than 1 km. We do note that a detailed forward modeling study to quantitatively evaluate the structural heterogeneities, including the detailed 3D shape and the extent of velocity/quality anomaly, is beyond the scope of this work, and ideally suited to a follow-up study.

**Corner frequency estimation (S-phases)**. We first estimate the size and duration of the initial (broadband) portion of EHW waveforms by constraining seismic moment ($M_0$) and corner frequency ($f_c$) using a spectral ratio approach (Fig. 3a). By applying signal-to-noise and additional quality control criteria, we obtain $f_c$ estimates for 15 EHWs based on the S-phase spectrum of the broadband portion of the waveform. Spectral ratio techniques take advantage of a co-located event pair in order to cancel travel path, site, and all other non-source related effects between the spectra of a smaller (empirical Green function, EGF) and larger event (target event). Thus, using the method requires two events that meet both requirements of co-location and magnitude difference (as well as high SNR over a sufficient frequency bandwidth). To make sure the co-location assumption is satisfied, we require the inter-event distance of <1 km (accounting for location error), and more decisively, the full waveform similarity (6 s window length waveform) of the event pair as exhibited by the cross-correlation coefficient must be higher than 0.7 after applying a bandpass filter of 1–20 Hz. The corner frequency estimation is considered stable when the CC threshold is set in the range between 0.7 or higher[69,70]. Following ref. [35], we also require a magnitude difference of >0.5 and a ratio of low-frequency amplitudes ($\Omega_0$) larger than 2 in order to ensure that the corner frequency and long-period spectral amplitudes between the two events are resolvable in the spectral ratio fitting. All the typical induced events reported in ref. [24] are taken as potential EGF candidates. The fitting frequency band is defined where (a) SNR exceeds 2, (b) the minimum and maximum fitting frequencies satisfy $0.5\,Hz \leq f_{min} \leq 5\,Hz$ and $10\,Hz \leq f_{max} \leq 80\,Hz$ respectively, and (c) the entire bandwidth is larger than 20 Hz.

The displacement spectral ratio is calculated by stacking the single-station spectral ratio. An example is shown in Fig. 3a. We then fit the spectral ratio using a Brune model[71,72],

$$\frac{\Omega_1(f)}{\Omega_2(f)} = \frac{M_1}{M_2}\left(\frac{1 + \left(\frac{f}{f_{c2}}\right)^2}{1 + \left(\frac{f}{f_{c1}}\right)^2}\right) \tag{1}$$

where $M_1$ and $M_2$ are seismic moment values of the target and EGF events, respectively. The estimated seismic moment is

based on low-frequency amplitudes ($\Omega_0$) of the single spectrum estimate[71,72],

$$M_0 = \frac{4\pi\rho c^3 R\Omega_0}{U_{\phi\theta}} \quad (2)$$

where density, $\rho$, is set to 2790 kg/m$^3$, shear wave velocity $c$ is chosen according to the value in the velocity model at the focal depth of each respective event, and $R$ is the hypocentral distance. The mean radiation pattern[73], $U_{\phi\theta}$, is set to be 0.63.

The uncertainty of estimated ($f_{c1}$) is set at the range where the corresponding variance increases by 5%. Two quality control criteria are required to assure a robust fit: (1) the RMS value of spectral ratio fitting is smaller than 0.3, and (2) $\delta f_{c1}/f_{c1}$ is no larger than 1 (to guarantee that our fitting is sensitive to $f_{c1}$). For cases where a target event has several EGFs, a weighted $f_{c1}$ is calculated based on inverse-variance weighting[69]. Fig. S7 provides representative examples to show the quality of spectral ratio fitting for the EHWs.

**EHW source parameter estimation**

*Stress drop estimation.* Under the assumption of a circular crack model[40], we calculate the stress drop ($\Delta\sigma$) using

$$r = \frac{kc}{f_c} \quad (3)$$

$$\Delta\sigma = \frac{7M_0}{16r^3} \quad (4)$$

where $k$, a constant related to the reciprocal relation between $f_c$ and $r$ (source radius), is set as 0.25 and 0.32 for P- and S-waves respectively by assuming the rupture velocity of 90% of the shear wave velocity[35,74–76].

*Rupture velocity estimation.* For an EHW and a typically induced earthquake with the same seismic moment ($M_0^e = M_0^H$; where the $H$ and $e$ superscripts denote the EHW and typical earthquakes, respectively), we assume they have the same stress drop value ($\Delta\sigma^e = \Delta\sigma^H$). The rupture speed of the EHWs is then estimated using

$$v_r^H = r^H f_c^H = r^e f_c^H = \frac{kc}{f_c^e} f_c^H \quad (5)$$

Where the $f_c^e$ is calculated from the averaged $f_c$ of all the typical earthquakes, with $M_0^H$ falling in the uncertainty range of $M_0^e$, i.e., $M_0^H \in [M_0^e - \delta M_0^e, M_0^e + \delta M_0^e]$.

**Coupled evolution of pore pressure and poroelastic stress.** We calculate the coupled evolution of pore pressure and poroelastic stress in relation to the HF injection parameters. The pore pressure evolution can be calculated by solving the coupled diffusion equations[77,78], assuming that the medium is homogeneous and isotropic,

$$\rho S \frac{\partial p}{\partial t} - \nabla \cdot \left(\rho \frac{\kappa}{\mu_d}\nabla p\right) = Q_m(x,t) - \rho\alpha\frac{\partial \varepsilon}{\partial t} \quad (6)$$

$$S = \chi_f \theta + \chi_p(1-\theta) \quad (7)$$

$$q = -\frac{\kappa}{\mu_d}\nabla p \quad (8)$$

Where $\rho$ is fluid density, $S$ is the linearized storage parameter, $p$ is pore pressure, $\kappa$ is the permeability of the medium, $\mu_d$ is dynamic viscosity, $Q_m$ is volumetric injection rate, $\alpha$ is Biot-Willis coefficient, and $\varepsilon$ is strain tensor, $\chi_f$, $\chi_p$ are fluid compressibility and bulk compressibility, respectively, $\theta$ is porosity, $q$ is Darcy flux,

which is injected flow rate per area according to the well report (m/s). The poroelastic stress variation can be constrained by,

$$-\nabla \cdot \boldsymbol{\sigma} = \boldsymbol{F_v} \quad (9)$$

$$\sigma_{ij} = \frac{2G\nu}{(1-2\nu)}\varepsilon_{kk}\delta_{ij} + 2G\varepsilon_{ij} - \alpha p\delta_{ij} \quad (10)$$

$$\varepsilon_{ij} = \frac{1}{2}((\nabla\boldsymbol{u})^T + \nabla\boldsymbol{u}) \quad (11)$$

Where $\sigma$ is the stress tensor, $F_v = (\rho\theta + \rho_b)g$ is the volume force vector, $\rho_b$ is the bulk density, $G$ is shear modulus (=30 GPa), $\nu$ is Poisson's ratio (=0.25), and $\boldsymbol{u}$ is the deformation vector.

We simulate the evolution of coupled stress changes using the COMSOL Multiphysics software by applying the solid mechanism module and Darcy's fluid flow module. We set the model as a layered elastic medium inferred from Crust 1.0[65] with an embedded 300-m-thick shale layer (Fig. 5a). As the stress changes do not depend on the initial stress state, we assume that (1) the initial normal stress follows lithostatic gradient, (2) pore pressure follows hydrostatic gradient, and (3) shear stress is the product of friction coefficient and normal stress. HF injection information is available in Table S2. Other aforementioned modeling parameters are listed in Table S4. Specifically, we set the permeability within the shale layer as $1 \times 10^{-14}$ m$^2$, which is equivalent to a diffusivity value of 0.2 m$^2$/s in our case (using Eqs. 7–8 from ref. [14]). We note that applying a strain-independent permeability may have minor effects on predicting the stress perturbation in the proximity of the well[79]. We set 2015/07/10 00:00:00 as $t = 0$ and run the model for 100 days. The model has 123, 147 cells with adjustable grids. It runs for 20.7 h with four-core parallel computing.

Next, we resolve the poroelastic stress and pore pressure change onto the fault plane. We assume the ruptures are (a) near the wellbore: normal slip along a fault striking parallel to $S_{Hmax}$ (strike = 60°, dip = 90°, rake = 0°), (b) near station MG08: thrust slip along a fault with preferred strike (strike = 30°, dip = 90°, rake = 90°), (c) at the center of the fitted structure: thrust slip along the fitted plane (strike = 150°, dip = 66°, rake = 90°). We use the following to calculate the corresponding $\Delta$CFS[80],

$$
\begin{aligned}
\Delta CFS = \Delta\tau + \mu(\Delta\sigma + \Delta p) = {} & \sin\lambda - \frac{1}{2}\sin^2\phi\,\sin(2\tilde{\delta})\sigma^{11} \\
& + \frac{1}{2}\sin(2\phi)\sin(2\tilde{\delta})\sigma^{12} + \sin\phi\cos(2\tilde{\delta})\sigma^{13} \\
& - \frac{1}{2}\cos^2\phi\,\sin(2\tilde{\delta})\sigma^{22} - \cos\phi\,\sin(2\tilde{\delta})\sigma^{23} + \frac{1}{2}\sin(2\tilde{\delta})\sigma^{33} \\
& + \cos\lambda - \left[\frac{1}{2}\sin(2\phi)\sin\tilde{\delta}\sigma^{11} + \cos(2\phi)\sin\tilde{\delta}\sigma^{12}\right. \\
& + \cos\phi\cos\tilde{\delta}\sigma^{13} + \frac{1}{2}\sin(2\phi)\sin\tilde{\delta}\sigma^{22} + \sin\phi\cos\tilde{\delta}\sigma^{23}\Big] \\
& + \mu\Big[\sin^2\phi\sin^2\tilde{\delta}\sigma^{11} - \sin(2\phi)\sin^2\tilde{\delta}\sigma^{12} \\
& - \sin\phi\,\sin(2\tilde{\delta})\sigma^{13} + \cos^2\phi\sin^2\tilde{\delta}\sigma^{22} \\
& + \cos\phi\,\sin(2\tilde{\delta})\sigma^{23} + \cos^2\phi\sigma^{33} + \Delta P\Big]
\end{aligned}
\quad (12)
$$

where $\mu = 0.6$ is the friction coefficient, $\phi$, $\tilde{\delta}$, and $\lambda$ are the strike, dip, and rake of the receiver fault, respectively, $\sigma^{ij}$ is the stress tensor, where $i, j = 1, 2, 3$ are the 3D components in the Cartesian coordinate system, and $\Delta p$ is the pore pressure change. In Fig. 5, we show stress evolution at three representative locations: near

the wellbore, near station MG08, and at the center of the fitted structure in the south.

## Data availability

All data needed to evaluate the conclusions in the paper are present in the paper and/or the Supplementary Materials. Waveform data of the 31 EHWs (Table S1) can be downloaded under the following link: https://www.geophysik.ruhr-uni-bochum.de/download/public/event_waveform_sac.zip. Well location, geometry data, and operation information are provided in Supplementary Materials. They can be downloaded from geoLOGIC database (https://www.geologic.com/), access by subscription.

## Code availability

*HypoDD* relocation package is open source, available at https://www.ldeo.columbia.edu/felixw/hypoDD.html. Modeling software COMSOL is available by subscription. Additional scripts related to this paper may be requested from the corresponding author.

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

## Acknowledgements

We thank Conan Drummond for his help in the deployment of seismic stations, as well as Progress Energy, in particular, Mark Norton, for assistance in collecting the waveform data, as well as the BC Oil and Gas Commission for providing injection data. H.Y. was supported by the Postdoctoral Research Program of Natural Resources Canada. R.M.H. was supported by start-up funds from the Ruhr University Bochum, H.K. was supported by the Environmental Geoscience Program of Natural Resources Canada, and Y.L. was supported by Natural Sciences and Engineering Research Council of Canada (NSERC) Strategic Grant STPGP/494141-2016. This paper is NRCan contribution 20210336.

## Author contributions

H.Y. wrote the manuscript. H.Y. and B.W. processed and analyzed the data. B.W. conducted the COMSOL numerical modeling. All authors (H.Y., R.M.H., H.K., Y.L., and B.W.) discussed, reviewed, and revised the article.

## Funding

## Competing interests

The authors declare no competing interests.

## Additional information

*Nature Communications* thanks Andrea Porcheddu and the anonymous reviewer(s) for their contribution to the peer review of this work. Peer reviewer reports are available.

