## [Peer Review File · Nature Communications]

REVIEWER COMMENTS

Reviewer #1 (Remarks to the Author):

I have reviewed the manuscript on "Fluid-injection induced earthquakes characterized by hybrid-frequency waveforms manifest the transition from aseismic to seismic slip". Overall, the manuscript is well-written and presents some very interesting ideas that are at the cutting edge of the science of induced seismicity. However, in my opinion, the evidence used to support the conclusions of the paper is pretty weak and principally based on one (not well presented: see comments) observation that has previously been reported for induced seismicity (i.e. that some of the events have a lower corner frequency and thus longer duration than expected). The potential errors in the estimations of the source duration are also not adequately discussed, which may significantly affect whether these events are truly different from "ordinary" events.

I do not wish to discourage the Authors; I do believe that a reworked version of this manuscript, taking into account some of the comments here, would be an interesting observational seismology paper in a more targeted journal; the observations are of interest to the community. I also commend the Authors on taking into account many of the alternative explanations for the observations that they report (e.g. low frequency coda caused by path effects, considering the hybrid event model from volcano seismology, etc.). However, currently I am not convinced that the results presented are of sufficient impact (or present a significant breakthrough) to be of interest for the broader audiences targeted by Nature Communications.

However, given the considerations above and the number of major issues listed below, unfortunately I have to recommend rejection of this manuscript for Nature Communications at this time. I have provided my major and minor comments below which I hope will help the Authors to revise the manuscript for a future submission.

Major issues:

1. The evidence provided in the manuscript to show that the EHWs are significantly different from "ordinary" events is quite weak. Ideally, stronger evidence should be provided (see detailed comment below).
2. All of the interpretation is based on observations of events with a lower corner frequency. This is not a new observation for fluid-induced seismicity (e.g. Abercrombie and Leary, 1993; Fehler and Phillips, 1991; Sumy et al., 2017 showed that in some cases stress drops can be over an order of magnitude lower than natural events), and therefore I question whether this manuscript presents results that are sufficiently impactful for Nature Communications.
3. Personally, I really like the interpretation that the Authors provide, but there is no more evidence for this interpretation than events with slightly low corner frequencies, which have been used by the other studies mentioned above to suggest other interpretations (e.g. due to high pore pressures (Sumy et al., 2017), tensile versus shear events (Fischer and Guest, 2011)). Alternatively, the events are taking place in a stratified sedimentary basin, where rock units may have very different properties including shear velocity. It is therefore not surprising to me that events may have different rupture velocities if they are occurring at slightly different depths, without needing necessarily to advocate an aseismic/slow driving process. Currently, it's the interpretation that distinguishes this manuscript from those studies, but there is no evidence provided that suggests that this explanation better fits the data. The other interpretations should equally be considered (including whether it could result from errors), and I also suggest toning down the language around this interpretation. I am currently not convinced that these are true "hybrid" events (from my knowledge "hybrid" events were first documented in volcano seismology and are usually events with significant portions of both high and low frequencies; however, the Authors have shown that in this study the main low frequency part presented is caused by path and not source effects), but they are presented as such.
4. I wonder if the coda duration patterns for the stations are focal mechanism-related. The potential effects of focal mechanisms are not discussed. For example, the proximal EHW in Fig S5 shows longer durations in a NNW-SSE region through stations 3,5 and 7, and the stations with longest duration for the further EHW are 1,2,7,8; all in the same region of the array. This may warrant some consideration, and in my mind better explains the observations: the fact that these wavetrains are low frequency and are described to have retrograde motion suggests that they are surface waves; I therefore think that it is difficult to see how they

could be generated at depth on velocity heterogeneities as hypothesized by the authors. However, certain focal mechanisms could generate stronger energy in different directions, with resulting variations in surface wave energy and duration.

5. Different possible error sources in the analysis are not adequately considered. Some events appear to be located > 4km away from the wells. Is this really likely? Higher resolution monitoring arrays do not usually observe induced events as far away from HF wells (e.g. Bao and Eaton, 2016; Eaton et al. 2018; Eyre et al., 2019; Kozłowska et al., 2018; Clarke et al., 2019; McKean et al., 2019). Location errors would affect both magnitude and attenuation estimates and could lead to the spread observed between "ordinary" and EHW events. Also, focal mechanisms could greatly affect magnitude estimates given that the Authors have assumed a constant factor to estimate magnitudes. This is valid and common but could still lead to significant errors, especially when comparing two main clusters which may have common internal features leading to different systematic errors for each cluster, even though the results are averaged over an array of stations. This could explain some observations (e.g. Fig 4). Some consideration of these aspects should be given in the discussion. These and other errors could also play into the slip velocity versus distance from wellbore pattern suggested in the manuscript. For example, the Authors' principle argument is about the observations from the two main clusters. But the differences in these clusters may be because of differences in the sampling of the focal mechanism by the array or due to differences in location errors. Also, events furthest from the array likely have the highest location errors and are more affected by errors in attenuation estimation etc. So some of these aspects may explain some of the observations presented and need to be properly considered. Finally, how reliable is the EGF method for this data? It seems to me that the events do not match some of the required criteria for good EGFs, i.e. > an order of magnitude magnitude difference (Frankel and Kanamori, 1983), located within one source dimension (Abercrombie, 2015), share the same focal mechanism (Kane et al., 2013).

Main issue 1 in detail:

1. - Figure 1 shows a comparison of the pulse shapes for P- and S- arrivals for two events of supposedly approximate equal magnitude, to show how the durations are different between the two events. The events are stated to be (~MW1.5). Is this the calculated value for both? The EHW is stated to be further away (1.65km vs 1.32km) but has a larger amplitude in the figure (~ 1.8×10^4 vs $\sim 1 \times 10^4$ nm/s), suggesting that it might be a significantly larger event. This may explain at least part of the longer duration/lower frequency of the EHW, at least without further evidence.

- Figure 2B shows a comparison between the spectra of an EHW and a normal event. However, given that the events have a different magnitude, it is very difficult to compare them. The squashed y-axis also exaggerates the relative corner frequency positions, but it does not look like the scaling between the events would be far from the expected $M_0 \propto f_c^{-3}$. Also, how were the corner frequencies estimated in this case, they seem much poorer estimates than that (A) and both look like they should be higher values?

- Given that there are not that many events and that the conclusions of this paper are based predominantly on the interpretations of these corner frequency and magnitude results, I would like to see the spectral fits for many more of the EHWs and ordinary events for comparison in the Supplementary Materials, to give a better idea of the quality of fits.

- Figure 3C shows a slight separation between the EHWs and ordinary events. However, the characteristic durations of the EHWs are so small that its not very convincing. Some of the smaller grey points also plot above green region, and some of the events/error bars overlap. When other, unconsidered error sources are taken into account (major point 5) I would suspect that it is even less convincing.

Other main comments:

L135: As described above, I need more convincing than currently provided that "EHW spectral corner frequencies (f_c) are discernibly lower than those of ordinary induced events within the same magnitude range obtained from the same dataset"

L139-145: So they don't act specifically like "slow" events: if this was the case I would be much more convinced of the Authors' arguments. I appreciate that it doesn't rule it out as described in the discussion, but the evidence seems very thin.

L324: I have some issues with this migration analysis currently as it is measured from the first event after injection. Which event is this and is it close to the injection? Also, the wells are over 1km long horizontally so some diffusivity patterns may be influenced by errors introduced by this simplification (maybe the 3 events at $D = 1.5 \text{ m}^2/\text{s}$). These factors will influence the reliability. Also, it seems like only hypocentral

distance is included, but diffusivity will also be influenced by depth. A repeat of Figure 2A with progression over time would also help (in the supplementary materials?).

L344: ratio of 0.5 seems low for EGF, most studies advocate for a range of 1.0-2.0 (e.g. Frankel and Kanamori, 1983).

L402-414: From this I'm still unclear on exactly how the two low velocity zones are found from the raypaths and durations. Is it inversion, trial and error? Please better clarify the methodology, especially as this appears to be a new technique. Also, consider my major comment 5 which may also explain the observations.

- The fact that EHWs seem to occur at certain times of day but other induced earthquakes don't slightly worries me in terms of their source. The only explanation that I can think of if they are truly HF-induced events is if they are strongly correlated with injection times. However, I'm not sure how common it is for HF operations to only occur in a few hours time window consistently for several days, to my knowledge they are usually conducted on a tight time schedule to optimise costs. Is there any injection data for this treatment that can be used for correlation.

- On a related note, the injection is stated as taking place between 11-13 July. Why are there events detected before this in Table S1? This is not mentioned but seems extremely important to me! Are there any other HF or disposal wells nearby that any of these events could be related to? Or could they be natural? And to a lesser extent, what about the events months after injection- this is uncommon for HF induced seismicity? If some of these events are natural it could significantly alter the interpretations.

- EHW detection – is the STA/LTA designed so that only EHW events are detected, or do ordinary earthquakes also get picked up? How do you determine whether these events are truly EHWs or whether they are ordinary events? I would think a frequency band of 10-20 Hz should pick up both types of events, why not use a lower frequency for the EHWs? I get the impression that the distinguishment is maybe based mainly on the long "ringing" coda, but you have stated that this is a path and not a source effect, so separating event types based on this seems unintuitive.

- It would be interesting to compare locations of the EHWs with locations of ordinary events, and this may provide further insights on source mechanisms, how path effects may play a role etc depending if there are any patterns. A figure showing this would be nice, although I appreciate the other catalogue was published in an earlier paper.

- Also looking at the previous paper of the same monitoring campaign, there were events with similar low stress drops as the EHWs (or are they the same EHWs?).

Minor comments:

L20 – "several kms away" – not apparent whether it would act over these distances

L47 – "distances of several km" -> "distances of up to several km"

L112-116 – why the diffusivity

L161: "encounter" -> "encounters"

Similarly, rupture

L204-205: "speeds of the southern EHW cluster are generally higher than those closer to the wellbore, suggesting a positive correlation with distance to the wellbore." – possible, but given only two clusters its difficult to draw firm conclusions as it could just be something else about the locations of the two different clusters – is there any evidence of this for events within each cluster? Also see main comments.

L217-218: "Either case is thought to be the seismic manifestation of slow slip in aseismic to seismic fault transition zones" – this sentence does not really make sense, by definition slow slip = slow rupture, a manifestation of this may be the interpretation of low stress drops too.

L252-254: "Fluid migration is supported by the abnormally high hydraulic diffusivity ($\sim 0.2 \text{ m}^2/\text{s}$) observed in the tight shale formation between the wellbore and the southern cluster (Fig. 2B) (13)." – I would think that an abnormally high diffusivity may suggest that a different mechanism than fluid migration may be the cause?

L267-269: "One possible difficulty in the detection of EHW is that the characteristic low-frequency, low-amplitude ringing will be quickly buried by body wave dispersion and coda noise when recorded on stations at regional distances from injection wells." – but my understanding of your conclusions is that this "ringing" is a path effect and therefore not indicative of the source process. The events shown here still have a broad frequency onset that should be just as easily detected for events of the same magnitude at distal stations. More important are events with low corner frequencies, and this is not necessarily a common technique, and

rarely analysed for enough (or even any) events to see such a pattern.

L397-401: basically an argument for why get lower frequencies in a fluid filled, heterogeneous environment. Maybe this should be considered in terms of interpreting lower corner frequencies for the events. I know that the EGF method in theory removes the path effect, but given the relatively small study region some of the EGFs seem to be a significant distance from the events analysed (0.7km), i.e. beyond the one source dimension suggested by Abercrombie (2015) for EGFs given the low event magnitudes. Given the HF and related pore pressure changes are isolated to relatively small regions, path effects could change significant over a small distance and this could also be a contributing factor in the results?

Figure 1A: Too small, some parts are very difficult to see (e.g. grey rectangle, all red stars etc.). and some labels (e.g. top-right inset)

Figure 2B: Define "distance", i.e. from first event.

L653: "Thick blue curve: HF well" -> "Thick blue lines: HF wells"

Figure 3C; S4: Y-axis label "duratuon" -> "duration"

L663: "displacement of two events" -> "displacement of the two events"

Figure 4: Why show ordinary IIE in (A) but not (B)?

Supplementary Materials:

L34: "short duration of low-frequency" -> "short duration of the low-frequency"

References:

- Abercrombie and Leary, 1993; Fehler and Phillips, 1991; Sumy et al., 2017 Fischer Guest
- Abercrombie, R., and P. Leary, 1993, Source parameters of small earthquakes recorded at 2.5 km depth, cajon pass, southern california: implications for earthquake scaling: *Geophysical Research Letters*, 20, 1511–1514.
- Abercrombie, R. E., 1995, Earthquake source scaling relationships from- 1 to 5 ml using seismograms recorded at 2.5-km depth: *Journal of Geophysical Research: Solid Earth*, 100, 24015–24036.
- Bao, X., and D. W. Eaton, 2016, Fault activation by hydraulic fracturing in western Canada: *Science*, 354, 1406–1409.
- Clarke, H., J. P. Verdon, T. Kettlely, A. F. Baird, and J.-M. Kendall, 2019, Real-Time Imaging, Forecasting, and Management of Human-Induced Seismicity at Preston New Road, Lancashire, England: *Seismological Research Letters*, 90, 1902–1915
- Eaton, D. W., N. Igonin, A. Poulin, R. Weir, H. Zhang, S. Pellegrino, and G. Rodriguez, 2018, Induced Seismicity Characterization during Hydraulic-Fracture Monitoring with a Shallow-Wellbore Geophone Array and Broadband Sensors: *Seismological Research Letters*, 89, 1641–1651.
- Eyre, T. S., D. W. Eaton, M. Zecevic, D. D'Amico, and D. Kolos, 2019a, Microseismicity reveals fault activation before Mw 4.1 hydraulic-fracturing induced earthquake: *Geophysical Journal International*, 218, 534–546.
- Fehler, M., and W. S. Phillips, 1991, Simultaneous inversion for q and source parameters of microearthquakes accompanying hydraulic fracturing in granitic rock: *Bulletin of the Seismological Society of America*, 81, 553–575.
- Frankel, A., and Kanamori, H., 1983. Determination of rupture duration and stress drop for earthquakes in southern California. *BSSA*, 73,(6), 1527-1551.
- Fischer, T., and A. Guest, 2011, Shear and tensile earthquakes caused by fluid injection: *Geophysical Research Letters*, 38.
- Kane, D.L., Kilb, D.L., and Vernon, F.L., 2013. Selecting Empirical Green's Functions in regions of fault complexity: A study of data from the San Jacinto Fault Zone, Southern California: *BSSA*, 66(3), 639-666.
- Kozłowska, M., M. R. Brudzinski, P. Friberg, R. J. Skoumal, N. D. Baxter, and B. S. Currie, 2018, Maturity of nearby faults influences seismic hazard from hydraulic fracturing: *Proceedings of the National Academy of Sciences*, 115, E1720–E1729.
- McKean, S. H., J. A. Priest, J. Dettmer, and D. W. Eaton, 2019, Quantifying Fracture Networks Inferred From Microseismic Point Clouds by a Gaussian Mixture Model With Physical Constraints: *Geophysical Research Letters*, 46, 11008–11017.
- Sumy, D. F., C. J. Neighbors, E. S. Cochran, and K. M. Keranen, 2017, Low stress drops observed for aftershocks of the 2011 mw 5.7 prague, oklahoma, earthquake: *Journal of Geophysical Research: Solid Earth*, 122, 3813–3834.

Thomas Eyre

Reviewer #2 (Remarks to the Author):

Thank you for the opportunity to review this manuscript. It is a very interesting study that describes and analyzes a new, potentially important signal of induced shear slip during HF stimulation. Overall, the data analysis is robust and rigorous, but the wording, interpretations, and figures could be strengthened to emphasize/clarify the significance of the results. Specific comments are provided in the annotated .pdf and several general comments are provided below.

1. I am not sure about the idea that the EHW 'bridges' the gap between inferred aseismic slip and seismic slip at greater distances. I understand the intent, but another way to approach this would be to directly address the physical processes that operate at different distances. For example, the authors could outline 3 regimes: wellbore, near-wellbore, and far away from wellbore and discuss how each of the mechanisms described in the discussion may or may not play a role.
2. One important idea missing from the manuscript is a discussion of effective normal stress. It is somewhat implied in the discussion of dilatant strengthening and slip on non-critically stressed faults, but it should be directly stated that low effective normal stresses can be achieved through active pumping near the wellbore or into a feature (e.g., southern cluster fault) that is hydraulically connected to the near wellbore region.
3. There is a missed opportunity here to add in some simple, but potentially impactful geomechanical analysis. In Fig. 1, the direction of SHmax is indicated on the map, but the authors don't return to this idea when discussing the southern cluster fault plane. I was left wondering if the best fit plane was or was not critically stressed and what magnitude of pore pressure change was needed to induce slip? Adding this simple analysis and plotting the best fit plane on the map in Fig. 1 would help connect this study to the referenced geomechanical studies in the same area. In addition, comparing the estimate of the pore pressure change to the diffusion model could yield insights into the timing of the events and strengthen the authors' interpretations of the formation diffusivity.
4. Is moment tensor inversion possible for this dataset? There are recent studies of HF seismicity that attempt to extract best fit planes and estimate principal stress directions (e.g., Kuang, W., Zoback, M., & Zhang, J. (2017). Estimating geomechanical parameters from microseismic plane focal mechanisms recorded during multistage hydraulic fracturing. *Geophysics*, 82(1), KS1-KS11. <https://doi.org/10.1190/geo2015-0691.1>). If possible, it would be interesting to perform the geomechanical analysis described in comment 3 on the EHWs and ordinary events.
5. The discussion of EHWs in context of seismic hazard needs to be reworked. The purpose of HF stimulation is to induce slip on faults in order to enhance the surface area of the pre-existing fracture network. In this context, there seem to be two conflicting objectives. If one was somehow able to reproduce the near-wellbore conditions of EHWs elsewhere in the formation, would the induced accomplish the necessary permeability enhancement? The authors could potentially recast Fig. 4A in terms of source dimension and slip in order to say something about whether aseismic slip achieves the same scale of surface area creation. On a related note - If the dataset of EHWs was larger, it would be interesting to see if the Gutenberg-Richter distribution was similar or different to that of the ordinary events.

Thank you again for the opportunity to review this manuscript.

Arjun Kohli

Reviewer #3 (Remarks to the Author):

A – Report Summary

The authors observed 31 slow slip induced seismic signals related to an industrial HF treatment in the Montney Play, Canada. This is an interesting new catalogue of such events which are difficult to measure in deep field contexts. The authors then make a “classical” source analysis of these events, and compare to collocated induced seismic events. Their clearest finding is that there is a significant difference in the corner frequency. They then used this measured parameter to estimate EHWs source stress drop and rupture velocity showing some analogy with tectonic slow slip events in literature. To summarize, this study is robust but classical. The high merit of the authors is to bring to the scientific community a nice catalogue of exotic events (they call EHWs) which are well located around a hydrofracking treatment well. Unfortunately, they are very little information about the hydrofracking treatment (volume, protocol, time duration, etc) that could allow the community to eventually reanalyze these data. This is to our opinion a strong weakness of the paper. We would recommend this contribution more as a report for the quality of the observations, if they can be completed with some crucial information on the HF treatment.

- What are the noteworthy results?

A catalogue of well localized and characterized hybrid frequency waveform seismic events. These events look like slow slip events observed in tectonic context. From this comparison, the author hypothesize that they might be proxies of the transition from aseismic to seismic slip around a fluid injection

- Will the work be of significance to the field and related fields? How does it compare to the established literature?

The originality of the work is the catalogue of 31 events, and the industrial context of a real HF treatment. The data processing is not original, and authors provide relevant references.

- Does the work support the conclusions and claims, or is additional evidence needed?

Additional evidence is needed. There are not enough information about the hydraulic treatment.

- Are there any flaws in the data analysis, interpretation and conclusions? - Do these prohibit publication or require revision?

Interpretation and conclusion seem to mainly rely on one parameter, and thus may look overinterpreted.

- Is the methodology sound? Does the work meet the expected standards in your field?

The work meets the usual standards in the field of IS analyses.

- Is there enough detail provided in the methods for the work to be reproduced?

No. There are no information on the hydraulic source

B - Additional detailed comments in the text

Introduction

Lines 45 to 50 – “However, the combination of factors that are associated with HF 46 injection, such as the relatively small total injected fluid volumes, the short injection time 47 history, and the often nearly instantaneous onset of seismicity at distances of several 48 kilometers from wellbores, makes it challenging to explain the occurrence of M3+ 49 earthquakes with the classical concept of pore pressure and/or poroelastic stress change”. We do not see the clear link between the affirmations in this sentence and the bibliographic references. In addition, the fact that an industrial HF treatment involves a relatively small total injected fluid volume is not true.

Lines 54-56 – “Under the optimal geomechanical (e.g., if the hosting fault is critically 55 stressed) and compositional (e.g., high content of clay and total organic carbon) 56 conditions, aseismic slip fronts can move ahead of the pore-pressure diffusion and may 57 interact with nearby, larger faults to trigger significant events.” Here again, the fact that the compositional fault conditions may play a role in the preferential development of aseismic slip front ahead of the pressurized zone is not as clear as stated by the authors. The references cited do not really support the idea (they do not deny that there could be a link either).

1 – Correlation of EHWs with HF treatment

Line 82 – “We first 82 check the correlation between the HF injection and the distribution of the EHWs.” We are not fully convinced by the correlation, may be because it is not enough documented in the paper (Text S2 is not bringing significant additional information to the main text). It would for example be interesting to get a time variation of the injected water volume, with the time occurrence of the EHWs and IS plotted.

Figure 2 – Dot color figuring the EHWs hour is not accurate enough to clearly the chronology of these events. The ‘classical’ IS events should be shown in the figure as it is in Fig. S2

Lines 112 – 116 – “As shown in Fig. 2B, the migration of EHWs mainly follows a fluid diffusivity value of

$\sim 0.2 \text{ m}^2 \text{ 113 /s}$ (see in Methods), which is consistent with the estimation based on ordinary 114 induced events (13). A few EHWs located near the station MG08 are compatible with a higher diffusivity value of $\sim 1.5 \text{ m}^2 \text{ 115 /s}$, suggesting the existence of a fluid conduit that 116 connects to the well." Seems to me like a lot of interpretations here. Fig. 2B mainly shows a high dispersion of the plot...

2 – EHWs Source mechanisms

Lines 119 to 127 should be moved to the later discussion

Except the difference in the corner frequencies, it is not very clear that there is a difference between the EHWs and the classical earthquakes. The moment-duration scaling is not conclusive.

Lines 146 – 162 - The apparent duration of the low-frequency portion of the EHW waveform is interpreted as related to geological heterogeneities and dispersion. This is mainly based on the poor correlation ($R = 60\%$) of the T_a vs hypocentral distance. May be the authors underestimate the possibility of fluid driven effect. Overall, the data make interpretations hard to achieve here.

3 – EHW rupture characteristics

Lines 171 – 181. These lines gives details about a basic way to estimate rupture characteristics. They should shorten this paragraph only stressing on the fact that they use collocated classical IS events to compare ruptures between EHWs and IS. In scenario 1, they take the same rupture speed for both events types, which is a strong assumption. Nevertheless, it shows that EHWs stress drop values are statistically 16 times smaller than for ordinary events. In sentence of line 190, it is not clear if this is the stress drop of the collocated IS of the same field test or an average value from literature. In the second case, it would be much more appropriate to compare to the stress drop estimated from the collocated events...In scenario 2, author assume the same stress drop between EHW and IS. This way they calculate a much smaller EHWS rupture velocity than IS, which is in the range of tectonically induced LFE. One question arises wether this paragraph about the EHW rupture characteristics should not be in the discussion chapter since it relies on many assumptions and only one field estimated parameter which is the corner frequency f_c . In addition, it is not fully clear that the comparison is done with the IS rupture characteristics of that experiment or with more general IS characteristics from the bibliography.

Discussion

Lines 208 – 218 are more a summary of the results than a discussion. Line 217-218 is not really proved, unless the authors can exclude any fluid coupling effects on rupture (they arguments are not fully clear to do this since they rely mainly on a poor T_a vs hypocentral distance correlation).

Lines 219-226 – I think that the author make a strong shortcut in their discussion by writing that these observed EHWs marl the boundary between aseismic and seismic slip just because they have common features with tectonic slow slip events.

Lines 229 – 232 – Pore pressure reduction inducing dilatant strengthening might depend a lot of the treatment injection rate. The authors should give information about the HF treatment adopted in their field study.

Line 252 – The hydraulic diffusivity value shown here was calculated from the fitting of a simple diffusivity model to the epicentral distance vs time plot. It would much better to use the hydraulic data to confirm or cross-check this high value.

Manuscript Number: NCOMMS-20-41246

Title: Fluid-injection induced earthquakes characterized by hybrid-frequency waveforms manifest the transition from aseismic to seismic slip

Authors: H. Yu, R. M. Harrington, H. Kao, Y. Liu, and B. Wang

We would like to thank the editor and three reviewers for their constructive comments that helped to significantly improve the quality of the manuscript. In this letter, we first describe our efforts to address the major concerns raised by the reviewers, followed by point-to-point responses to individual comments.

To emphasize and support our conclusion that the EHWs are indeed a new type of seismic signal not reported in any previous induced earthquake studies, and that they represent the manifestation of the transition from aseismic to seismic slip speeds, we now include the following new aspects and lines of evidence:

- ✧ Highlight the co-location of EHWs and typical induced events, a key line of evidence demonstrating that the range of stress drop values for different types of events is resolvable with our data set and represents real source differences (Lines 296-335).
- ✧ Include a new Fig. S3 to provide four pairs of representative examples between the co-located EHWs and typical induced events. We show the quality of spectral ratio fitting, as well as the differences in source parameters between the two (Lines 187-191, Lines 617-618).
- ✧ Conduct a coupled pore pressure and poroelastic stress modeling test using realistic multi-stage HF injection parameters (Fig. 2A; Table S2) to show compelling arguments for why aseismic slip loading best explains the generation of EHWs. (Lines 336-406, Lines 768-815, Fig. 5 and Tables S3).
- ✧ Interpret the EHW mechanisms in the context of three (newly) separated groups of varying proximity to the wellbore: 1) directly next to the well bore, 2) near the station MG08, and 3) in the southern cluster. The grouping helps highlight changes in source parameters of EHWs with distance that are also consistent with observations of typical induced events. (Lines 142-155).

Furthermore, we have added additional evidence to support our interpretation that the low-frequency ringing originates from velocity heterogeneity along the ray path, including:

- ✧ Additional analysis to show that the waveform duration of EHWs is not likely governed by focal mechanism radiation patterns (Lines 217-218, Lines 629-673, and Fig. S7).
- ✧ Moving the text describing the hypotheses that the variation of apparent duration stems from velocity heterogeneity along the travel paths from Method to the main text to improve clarity (Lines 218-229).
- ✧ Constructing a synthetic velocity heterogeneity model to simulate the waveforms of southern EHWs to verify that ray paths travelling through low-velocity volumes can generate low-frequency ringing (Figs. R1, R2).

Further major changes include:

- ✧ Removal of the discussion of the origin hour distribution of EHWs (Text S2; original Fig. 2A), as the distributions of origin hours for both EHWs and typical induced events appear to be unrelated to the injecting hours (discussed in detail below), but more likely a background seismic pattern. It is an intriguing observation but is beyond the scope of this study.
- ✧ Including additional information reported by industry to estimate the principal stress in our study area. We discuss how uncertainties related to inconsistency in operator stress measurement techniques render the corresponding stress state estimate unrealistic (Fig. R7).

Our point-to-point responses to individual comments are given below in the order of Reviewer #1, Reviewer #2 (we copied Reviewer #2's detailed comments from the annotated PDF), and Reviewer #3. The original review comments are in black, and our responses in blue. A "track-changes" version is also included for your convenience. Please refer to the PDF document of the revised manuscript with tracked changes for the corresponding line numbers mentioned in this response letter.

REVIEWERS' COMMENTS

Reviewer #1 (Remarks to the Author):

I have reviewed the manuscript on “Fluid-injection induced earthquakes characterized by hybrid-frequency waveforms manifest the transition from aseismic to seismic slip”. Overall, the manuscript is well-written and presents some very interesting ideas that are at the cutting edge of the science of induced seismicity. However, in my opinion, the evidence used to support the conclusions of the paper is pretty weak and principally based on one (not well presented: see comments) observation that has previously been reported for induced seismicity (i.e. that some of the events have a lower corner frequency and thus longer duration than expected). The potential errors in the estimations of the source duration are also not adequately discussed, which may significantly affect whether these events are truly different from "typical" events.

We would like to thank the reviewer for pointing out that the original version of the manuscript did not sufficiently emphasize that our observations of induced event corner frequencies (both EHWs and IIEs) are not exclusively low frequency. What makes the observation of EHWs consistent with the transition from aseismic to seismic slip is the temporal and spatial proximity to injection and the well, and the differences of corner frequencies between different types of events (EHWs and IIEs) that we constrain in the same crustal volume. As noted on Page 1 of this letter, we now highlight the co-location of EHWs and typical induced events. The collocation and range of measured source parameter values form a key line of evidence demonstrating that the range of stress drop values for different types of events is resolvable with our data set and represents real differences in rupture speeds manifesting the aseismic to seismic transition (Lines 296-335).

I do not wish to discourage the Authors; I do believe that a reworked version of this manuscript, taking into account some of the comments here, would be an interesting observational seismology paper in a more targeted journal; the observations are of interest to the community. I also commend the Authors on taking into account many of the alternative explanations for the observations that they report (e.g. low frequency coda caused by path effects, considering the hybrid event model from volcano seismology, etc.). However, currently I am not convinced that the results presented are of sufficient impact (or present a significant breakthrough) to be of interest for the broader audiences targeted by Nature Communications. However, given the considerations above and the number of major issues listed below, unfortunately I have to recommend rejection of this manuscript for Nature Communications at this time. I have provided my major and minor comments below which I hope will help the Authors to revise the manuscript for a future submission.

The changes and improvement we have made with the help of the reviewers' comments now better support that our study may be the first documentation of field seismic signals manifesting the complex interaction between seismic deformation and aseismic slip driven by HF well stimulation, a new triggering mechanism postulated by recent modeling studies and observed in meso-scale fluid injection experiments. We feel that this new discovery and its broad-ranging implications merit publication in Nature Communication.

Major issues:

1. The evidence provided in the manuscript to show that the EHWs are significantly different from “typical” events is quite weak. Ideally, stronger evidence should be provided (see detailed comment below).
 - a) Figure 1 shows a comparison of the pulse shapes for P- and S- arrivals for two events of supposedly approximate equal magnitude, to show how the durations are different between the two events. The events are stated to be ($\sim M_W 1.5$). Is this the calculated value for both? The EHW is stated to be further away (1.65km vs 1.32km) but has a larger amplitude in the figure ($\sim 1.8 \times 10^4$ vs $\sim 1 \times 10^4$ nm/s), suggesting that it might be a significantly larger event. This may explain at least part of the longer duration/lower frequency of the EHW, at least without further evidence.

The moment magnitudes (M_0) of events are estimated based on low-frequency amplitudes of single event spectra (Eq. 2). According to Eq. 2, $M_0 \propto c^3$, where shear wave velocity c is chosen according to the value in the velocity model at the focal depth of each respective event. In this case, the c value at the focal depth of the EHW event (1.93 km) is 2.59 km/s, but increases to 3.63 km/s at the focal depth of the typical induced event (2.83 km). The combination of a higher shear wave velocity at the source depth with a smaller low-frequency amplitude (for the case of a typical induced event) would be equivalent to that of a slower shear wave velocity and a larger amplitude (the EHW case), leading to approximately the same M_W for both events.

- b) Figure 2B shows a comparison between the spectra of an EHW and a normal event. However, given that the events have a different magnitude, it is very difficult to compare them. The squashed y-axis also exaggerates the relative corner frequency positions, but it does not look like the scaling between the events would be far from the expected $M_0 \propto f_c^{-3}$. Also, how were the corner

frequencies estimated in this case, they seem much poorer estimates than that (A) and both look like they should be higher values?

We suspect that the reference to Fig. 2B is probably a typo, instead the reviewer meant Fig. 3B. We have now removed Fig. 3B and have added a new Fig. S3. The new plot provides four representative example pairs of EHWs and typical induced events that are from each of the three clusters. The consistent lower corner frequency values of EHWs demonstrates the observable source property differences between the two types of events. Please refer to Comment #1(c) for more details.

- c) Given that there are not that many events and that the conclusions of this paper are based predominantly on the interpretations of these corner frequency and magnitude results, I would like to see the spectral fits for many more of the EHWs and typical events for comparison in the Supplementary Materials, to give a better idea of the quality of fits.

Thanks for the suggestion. We added the new Fig. S3 to show the quality of spectral ratio fittings for the EHWs and for typical induced events. For each type of signal, we provide four examples to represent events in each of the three clusters shown in the new Fig. 2C at a range of depths (i.e. near station MG08, near the wellbore, at the shallow portion of the southern structure, and at the deeper portion of the southern structure). Their locations are specifically marked in Fig. S3I. We elaborate this figure in Lines 187-191 as “Fig. S3 shows representative examples of spectral characteristics difference between the two types of events. These examples also demonstrate that the range of corner frequency values for different types of events is resolvable with our data set and represents real differences between event types.” and in Lines 617-618 as “Fig. S3 provides representative examples to show the quality of spectral ratio fitting for the EHWs.”

- d) Figure 3C shows a slight separation between the EHWs and typical events. However, the characteristic durations of the EHWs are so small that its not very convincing. Some of the smaller grey points also plot above green region, and some of the events/error bars overlap. When other, unconsidered error sources are taken into account (major point 5) I would suspect that it is even less convincing.

The new Fig. S3 and new Fig. 4 more clearly demonstrate that the f_c values of the EHWs are smaller than the typical induced events with similar moment magnitudes and similar hypocenter locations, which suggests that EHWs occur with possibly lower stress drop values compared to the typical induced

earthquakes (0.29 vs. 4.86 MPa), and/or slower rupture speeds on the order of 1 km/s (Fig. 4).

The difference in source parameters between the two types of events decreases with increasing distance from the well, likely for two reasons. One is a wider continuum of slip speeds of EHWs (Fig. 4), as suggested by the wide corner frequency values of EHWs among the further clusters (near MG08 and the southern cluster (Lines 279-281)). The other is that small magnitudes would naturally lead to smaller discrepancies in source durations between the EHWs and typical induced events. The source durations of the slow and typical (fast) earthquakes would cross below M2 (Ide et al., 2007; Fig. 3), which also agrees with our case study.

We addressed the reviewer's concern related to the errors in Comment #5.

2. All of the interpretation is based on observations of events with a lower corner frequency. This is not a new observation for fluid-induced seismicity (e.g. Abercrombie and Leary, 1993; Fehler and Phillips, 1991; Sumy et al., 2017 showed that in some cases stress drops can be over an order of magnitude lower than natural events), and therefore I question whether this manuscript presents results that are sufficiently impactful for Nature Communications.

The reviewer is correct that the absolute corner frequency values could vary depending on the approach and parameters applied in the estimation. However, the relative differences between two event types should be reliable when using the same dataset and methodology. Lower stress drop values of fluid-induced earthquakes have been widely observed and explained by many factors. For example, relatively shallow focal depths (Sato, 2006; Hough, 2014; Zhang et al., 2016), in-situ fluid-induced stress change (Abercrombie and Leary, 1993), heterogeneous slip along less well-developed structures (Fehler and Phillips, 1991), and different methods used in f_c estimation (Sumy et al., 2017). Studies with improved data resolution reported a positive relation between the stress drop and event-to-well distance as the injection-induced stress perturbation decreases with distance (Yu et al., 2020, Goertz-Allmann et al., 2011; Kwiatek et al., 2014). What distinguishes our case from previous studies is the relative differences between the co-located EHWs and typical induced events. This is now emphasized more clearly with changes in the text and the addition of Fig S3. The lower f_c values of EHWs compared with those of co-located typical induced events, using the same dataset and estimation method in the same study area, could not be explained by any of the above factors discussed in previous studies. Thus, the most compelling explanation in light of the relative source parameter differences is

that EHWs are the manifestation of the source process with lower stress drop and/or slower rupture speed that are fundamentally different from the typical induced events.

We added the above discussions in Lines 296-335.

3. Personally, I really like the interpretation that the Authors provide, but there is no more evidence for this interpretation than events with slightly low corner frequencies, which have been used by the other studies mentioned above to suggest other interpretations (e.g. due to high pore pressures (Sumy et al., 2017), tensile versus shear events (Fischer and Guest, 2011)). Alternatively, the events are taking place in a stratified sedimentary basin, where rock units may have very different properties including shear velocity. It is therefore not surprising to me that events may have different rupture velocities if they are occurring at slightly different depths, without needing necessarily to advocate an aseismic/slow driving process. Currently, it is the interpretation that distinguishes this manuscript from those studies, but there is no evidence provided that suggests that this explanation better fits the data. The other interpretations should equally be considered (including whether it could result from errors), and I also suggest toning down the language around this interpretation. I am currently not convinced that these are true “hybrid” events (from my knowledge “hybrid” events were first documented in volcano seismology and are usually events with significant portions of both high and low frequencies; however, the Authors have shown that in this study the main low frequency part presented is caused by path and not source effects), but they are presented as such.

The observations of EHWs (events co-located with ordinary induced earthquakes but characterized by hybrid-frequency waveforms) may not be exclusive in our study. In fact, if injection commonly first induces aseismic slip near the wellbore, EHWs could occur on a widespread basis rather than only being observable in the Montney formation. Here we try to provide a viable explanation to understand the genesis of these events. That is, they could be compatible with the aseismic slip driving process. To provide additional evidence to the source parameter observations, including the spatial distribution with respect to the well, we now demonstrate with numerical modeling that the EHW timing is better explained by aseismic slip loading from HF injection (summarized in the new Fig. 5). The results clearly indicate that the classical mechanisms of coupled pore pressure and poroelastic stress change are insufficient to address the temporal and spatial distribution of EHWs (Lines 352-406, Figs. 5B-D). Our interpretation may thus offer new insights into the physical conditions of the seismogenic process of injection-induced earthquakes.

Seismic signals characterized by similar hybrid-frequency content have been reported as “volcanic hybrids” in volcanic environments. We document in the text that two

different interpretations are proposed in understanding their origin (particularly the long-period component of their characteristic waveforms) which is indeed controversial (Aki et al., 1977; Julian, 1994; Chouet, 1996; Harrington and Brodsky, 2007; Bean et al., 2014; Denlinger and Moran, 2014; Harrington and Benson, 2011; Harrington et al., 2015). Therefore, the term “hybrid” is not necessarily meant to imply one distinct source process over another, but is rather meant to describe the nature of the signal. Given that the EHW waveforms exhibit the same features as volcanic hybrids, i.e., high frequency energy followed by low frequency ringing and long apparent durations relative to their magnitude, they may be classified as a hybrid signal just as well as any other exhibiting similar features. As such, we use the same approach as a number of studies to make inferences about the source of these signals, which includes looking at duration and frequency characteristics of both the broadband- and low-frequency part of the waveform. In doing so, we observe that the ringing of EHW (the apparent duration, T_a) at individual stations is dictated by the travel path. The distribution of T_a consistently suggests two volumes of low-velocity crustal heterogeneity near the wellbore (Fig. S6). The apparent waveform duration of the typical induced earthquakes is consistent with the inferred location of velocity heterogeneities, i.e., the coda wave of a typical induced event would look similar to the low-frequency ringing portion of EHWs if its ray path travels primarily through the low-velocity heterogeneity (black star, Fig. S6). Once corrected for the path effect, the apparent ringing duration of EHWs is positively correlated to the hypocentral distance (Fig. S8), suggesting that the extended low-frequency portion is likely the result of dispersion rather than pressure gradient-driven fluid flow in a crack.

We will explain why the ringing duration of EHWs are not likely governed by the focal mechanisms in the next Comment #4.

4. I wonder if the coda duration patterns for the stations are focal mechanism-related. The potential effects of focal mechanisms are not discussed. For example, the proximal EHW in Fig S5 shows longer durations in an NNW-SSE region through stations 3,5 and 7, and the stations with longest duration for the further EHW are 1,2,7,8; all in the same region of the array. This may warrant some consideration, and in my mind better explains the observations: the fact that these wavetrains are low frequency and are described to have retrograde motion suggests that they are surface waves; I therefore think that it is difficult to see how they could be generated at depth on velocity heterogeneities as hypothesized by the authors. However, certain focal mechanisms could generate stronger energy in different directions, with resulting variations in surface wave energy and duration.

Because the EHWs are all small in magnitude ($M < 2$), it is not possible to robustly constrain focal mechanism solutions with these surface stations. As such, the observable effects of source directivity would likely also be negligible. To further explore the effects of radiation pattern on apparent ringing duration (T_a), we have now included a consistency check documented in Fig. S7 and surrounding discussion (Lines 629-673), in which the inconsistency of the azimuthal correlation between amplitude and T_a rules out a significant influence of radiation pattern on the T_a . For example, if we assume that the observed T_a is indeed controlled by the focal mechanism, a positive correlation should exist between the T_a and the radiated energy, as well as between the waveform amplitude and the radiated energy. That is, if we compare two stations at the same epicentral distance, the stations at azimuths receiving more radiated surface wave energy are more likely to record waveforms with larger amplitude and longer T_a . Under this hypothesis, we should observe the consistency between the T_a and amplitude, but this is absent from our observation.

We show the variation of T_a and amplitude at nearby stations for two EHW events in Fig. S7. For the well-proximal EHW, MG01 and MG03 have similar distances and comparable amplitudes (38 nm/s vs. 34 nm/s, Fig. S7C), but the values of T_a are largely different (1.9s vs. 4.6s, Fig. S7A). The anticorrelation invalidates the hypothesis that the T_a is governed by the radiated energy. Similarly, for the EHW among the southern cluster, stations MG02 and MG05 show inconsistent results when we infer the amount of radiative energy from their waveform amplitudes or T_a (Fig. S7F). The discrepancy between waveform amplitude and T_a for both EHW examples suggests that focal mechanism, if not irrelevant, is not the key factor controlling the T_a pattern (Figs. S7E-F).

We added the above discussion in Lines 217-218 and Lines 629-673. Fig. S7 is also included in the Supplementary Materials.

On the other hand, to test whether the low-velocity heterogeneity can generate low-frequency ringing, we use SPECFEM3D, a finite element software for simulating seismic wave propagation through a heterogeneous media, to calculate the synthetic waveforms for the case of the southern EHWs (Fig. S6B). Fig. R1A (below) describes the model set-up, and Figs. R1B-C shows the simulated waveform at station MG02. Note that many EHW waveform features, including the longer ringing duration and hybrid-frequency contents, are reproduced. In contrast, the waveform calculated with a layered model without a low-velocity heterogeneity does not show the hybrid-frequency contents (Fig. R2). We further note that a detailed study combining inversion and forward modeling is required to map the 3D distribution of such velocity heterogeneities and evaluate their effects, but this is beyond the scope of this work. This simplified example is sufficient to at least demonstrate that the existence

of low-velocity heterogeneities could possibly lead to hybrid-frequency waveforms. Bean et al. (2008) also show similar waveform protraction by introducing a low velocity layer into their modeling.

Fig. R1. Synthetic waveforms of EHW signals from SPEC3D. (A) A schematic diagram showing the model setup of synthetic waveform experiment. We apply the location of the distal EHW example in Fig. S6 as the source. Focal mechanism is set as the normal fault along the fitted structure in Fig. 2B. The velocity model of Crust 1.0 (65) is adopted with a low-velocity heterogeneity shown in red. We assume the v_p and v_s of the low-velocity zone as 80% and 50% of that of the surrounding material (O'Connell et al., 1974), and attenuation of $Q_p=80$ and $Q_s=30$. We also assume the material is 100% saturated with 5%-30% porosity (Duitama-Leal et al., 2016). (B) Synthetic Vertical component waveforms at station MG02. (C) Corresponding spectrogram of waveforms in (B).

Fig. R2. Synthetic waveforms of signals without the presence of low-velocity heterogeneity. The same as Fig. R1 but the layered model does not include a low-velocity volume.

5. Different possible error sources in the analysis are not adequately considered. Some events appear to be located > 4 km away from the wells. Is this really likely? Higher resolution monitoring arrays do not usually observe induced events as far away from HF wells (e.g., Bao and Eaton, 2016; Eaton et al. 2018; Eyre et al., 2019a; Kozłowska et al., 2018; Clarke et al., 2019; McKean et al., 2019). Location errors would affect both magnitude and attenuation estimate and could lead to the spread observed between “typical” and EHW events. Also, focal mechanisms could greatly affect magnitude estimates given that the Authors have assumed a constant factor to estimate magnitudes. This is valid and common but could still lead to significant errors, especially when comparing two main clusters which may have common internal features leading to different systematic errors for each cluster, even though the results are averaged over an array of stations. This could explain some observations (e.g. Fig 4). Some consideration of these aspects should be given in the discussion. These and other errors could also play into the slip velocity versus distance from wellbore pattern suggested in the manuscript. For example, the

Authors' principle argument is about the observations from the two main clusters. But the differences in these clusters may be because of differences in the sampling of the focal mechanism by the array or due to differences in location errors. Also, events furthest from the array likely have the highest location errors and are more affected by errors in attenuation estimation etc. So some of these aspects may explain some of the observations presented and need to be properly considered. Finally, how reliable is the EGF method for this data? It seems to me that the events do not match some of the required criteria for good EGFs, i.e. $>$ an order of magnitude magnitude difference (Frankel and Kanamori, 1983), located within one source dimension (Abercrombie, 2015), share the same focal mechanism (Kane et al., 2013).

We note that both EHWs and typical induced events are detected before the HF stimulation, which could be possibly attributed to three other co-located wellbores that were active since 2013. In fact, the surge of seismicity in the northeast BC area since 2008 is largely associated with the drastic increase of HF injection (Fig. 1A; 2). Therefore, from a regional perspective, all the detected events here can be considered injection related. Consequently, it is rational to assume that the detected EHWs have similar seismogenic causes and to study them as a whole. We also apply the classical spatiotemporal analysis to associate seismic events with specific HF injection stages. Specifically, only events within 4 km from the wellbore and within 100 days after the commencement of HF injection are considered as directly related. The 4-km distance limit is chosen to reflect the maximum range of stress perturbation due to HF injections in our study area, according to Yu et al. (2019).

The location errors are marked in Fig. 2B. We require all the locations to have less than 0.5 km horizontal and 1 km vertical errors. Many of them are significantly smaller (Table S1). Therefore, the errors of M_0 and f_c related to the location errors should be limited. Please refer to Comment #1 (a) for further details.

Please refer to Comment #4 for the concern related to focal mechanisms; Comment #9 for the concern related to the spectral ratio methods.

Other main comments:

6. L135: As described above, I need more convincing than currently provided that “EHW spectral corner frequencies (f_c) are discernibly lower than those of typical induced events within the same magnitude range obtained from the same dataset”

Please refer to our responses to Comments #1 (c-d) and #5 of the first Reviewer.

7. L139-145: So they don't act specifically like “slow” events: if this was the case I would be much more convinced of the Authors' arguments. I appreciate that it doesn't rule it out as described in the discussion, but the evidence seems very thin.

The EHWs have longer source durations than the typical induced events, but, the moment-duration scaling is better fit by the $M_0 \propto T^3$, rather than the $M_0 \propto T$ relationship. This supports the circular crack rupture model, and does not conflict with the source features related to longer source durations.

We also refer to our responses to Comments #2 – #5 of the first Reviewer for additional evidence in support of our argument.

8. L324: I have some issues with this migration analysis currently as it is measured from the first event after injection. Which event is this and is it close to the injection? Also, the wells are over 1km long horizontally so some diffusivity patterns may be influenced by errors introduced by this simplification (maybe the 3 events at $D = 1.5 \text{ m}^2/\text{s}$?). These factors will influence the reliability. Also, it seems like only hypocentral distance is included, but diffusivity will also be influenced by depth. A repeat of Figure 2A with progression over time would also help (in the supplementary materials?).

The first well-located event after injection is now marked by a small gray arrow in Fig. 2B. It occurred on July 13, 2015, 10:52:22 UT (122.7811°W, 57.2483°N, Depth: 1.79 km).

In our previous version, we used the epicentral distance in the EWH migration analysis (the original Fig. 2B). In the revised Fig 2D, the Y axis corresponds to the hypocentral distance with respect to the first event (P1). The two results are essentially the same. Our conclusion remains valid that the migration of EHWs mainly follows a fluid diffusivity value of $\sim 0.2 \text{ m}^2/\text{s}$.

To test the possible effect due to the horizontal extent of the injection wellbore, we repeat the migration analysis for the five events located near station MG08 using the injection point closest to the vertical wellbore as the initial migration reference (122.8005°W, 57.2565°N, Depth: 2.2 km on July 20th, 2015, marked with a green triangle and the label “P2” in Fig. R3A). In this case, the two events previously following the diffusion front of $D = 1.5 \text{ m}^2/\text{s}$ are now more likely instantaneously induced by the injection (Fig. R3B). According to our geomechanical modeling result (the new Fig. 5, more details in our response to Comment #1 of Reviewer #2), either pore pressure diffusion or poroelastic stress perturbation is insufficient to induce the two events. Therefore, we suggest aseismic slip loading as a viable physical mechanism to shorten the process of stress accumulation. The other three events previously following the diffusion front of $D = 0.2 \text{ m}^2/\text{s}$ now indicate a slightly lower diffusivity. Since the change is not significant, and the estimated D values are more consistent with other EHWs and typical induced events (Fig. 2C; 22), we prefer to use P1 as the migrating initial point for all the EHW.

We revised the text in Lines 163-167 as “The high diffusivity values suggest they are either hydraulically connected to the volume near the well, or that they occur as the result of an instantaneous response to follow-up injection. Further investigation of the role of diffusivity in controlling the spatiotemporal distribution of pore pressure and poroelastic stress is given in the Discussion.” Rather than repeated images of 2A (now 2B) over time, we have added a color bar to show the time progression of events.

Fig. R3. Spatiotemporal distribution of EHWs near MG08. (A) 3D spatial distribution. Dots: 5 relocated EHWs near MG08. Dot color refers to the origin time. Gray arrow (P1): the initial reference of the diffusion in Fig. 2C. Green reverse triangle (P2): the initial migrating reference near the vertical wellbore for the 5 relocated EHW events near MG08. (B) Migration of EHWs near MG08. Yellow and green dots: migration of EHWs with start point using P1 and P2, respectively. Parabolic envelopes: the “diffusion front” calculated using diffusivity values indicated on the labels.

9. L344: ratio of 0.5 seems low for EGF, most studies advocate for a range of 1.0-2.0 (e.g. Frankel and Kanamori, 1983).

Many studies use a one-unit magnitude difference as a rule of thumb for selecting spectral ratio pairs, but it is still possible to resolve corner frequencies of a pair of

earthquakes with similar magnitudes (e.g., 0.6M, Folesky et al., 2016; Wu et al., 2018; 0.5M, Uchide and Imanishi, 2016; Ruhl et al., 2017; Holmgren et al., 2019 and 0.3M, Clerc et al., 2016). Huang et al. (2019) reveals that in some cases, master earthquakes can release smaller slip than their eGfs, and thus master earthquakes may have lower magnitudes (but larger source dimensions). Finding distinguishable spectral ratios that allow resolvable differences in spectral corner frequencies requires a balance between the moment ratio and noise levels. Our dataset with event-station distance as close as 10 km (mostly within 3 km) generally has very good quality with a high SNR, which contributes to the more relaxed magnitude difference requirement (Yu et al., 2020).

However, since spectral ratios are more sensitive to noise levels when the moment ratio is small, a stricter quality control criterion is necessary. In case when the magnitude difference is 0.5M unit, we also require the moment ratio between the master (target) and eGf events to be larger than 2. In practice, by visual inspection, the combined requirement of magnitude difference and moment ratio is very effective in retaining both the most qualified event pairs and the highest spectral ratio quality. We have tried other magnitude difference values, such as 0.7M unit and 1M unit (which are commonly applied to studies of induced earthquakes, Yu et al., 2020), but the improvement is negligible with respect to the result using the criteria described above. The criteria to find event pairs are the first step of event selection, and we try to avoid removing too many event pair candidates before knowing if there are resolvable differences in corner frequency. Ultimately, we ensure the robustness of our spectral ratio analysis by only accepting the results with excellent fitting quality.

10. L402-414: From this I'm still unclear on exactly how the two low velocity zones are found from the raypaths and durations. Is it inversion, trial and error? Please better clarify the methodology, especially as this appears to be a new technique. Also, consider my major comment 5 which may also explain the observations.
 - a) The fact that EHWs seem to occur at certain times of day but other induced earthquakes don't slightly worries me in terms of their source. The only explanation that I can think of if they are truly HF-induced events is if they are strongly correlated with injection times. However, I'm not sure how common it is for HF operations to only occur in a few hours time window consistently for several days, to my knowledge they are usually conducted on a tight time schedule to optimize costs. Is there any injection data for this treatment that can be used for correlation.

We collected the injection data of the five wellbores (Well ID: #30515-#30519) from geoLOGIC dataset (<https://gdcweb.geologic.com>). The operating dates within our observation period are summarized in Fig. 2A. There is a ten-day period of injection preparation and pressure testing (Jul 2, 2015 - Jul 11, 2015), followed by a ten-day period of hydraulic fracturing (Jul 11, 2015 - Jul 20, 2015). After waiting for 12 days (no injecting), there is a 16-day period of backflow production (Aug 2, 2015 - Aug 17, 2015; running 24 hours per day). We didn't consider the nearby active wells here because they are all more than 5 km away, i.e. too distant to influence the seismicity near our studied well pad (13).

We also count the cumulative number of injection stages in each hour respectively aiming at the Upper Montney (U), Upper Middle Montney (UM) and Lower Middle Montney (LM), as shown in Fig. R4. The U injections are more frequent during the daytime (peak at 10:00 am), the UM injections are generally less frequent, and LM injections are more frequent during the night hours (peaks at 2:00 am and 21:00 pm). We then compare the temporal distribution of injections with the origin hours of events occurred during the 10-day HF period. Since the event number of EHWs is too limited to show any statistical significance, we also consider the temporal distribution of typical injection-induced earthquakes (IIE) as a reference, assuming the two types of events have similar patterns of the origin hour. There is no clear consistency between the injection and earthquake origin hours, neither for EHWs (Fig. R4A) nor for typical induced earthquakes (Fig. R4B). Interestingly, when we compare the hourly distribution of events that occurred during the period without any well operations, both the EHWs and the typical induced events show some consistency with the hours of LM injections (Fig. R5). Thus, the aforementioned consistency between the EHW origin hours and the LM injections is probably a coincidence rather than a manifestation of any causal relation.

The distribution of origin hours for both EHWs and IIE does not appear to be significantly related to the injecting hours, but more likely a background seismic pattern. It is an interesting observation, but beyond the scope of this study. We therefore decided to remove the related part from the manuscript (Text S2; and the original Fig. 2A showing the 3D spatial distribution of origin hours of EHWs).

Fig. R4. Comparison during the 10-day hydraulic fracturing period about (A) Hourly distribution of the EHWs occurred (Blue bars) and the cumulative numbers of injection stages aiming at the Upper Montney (U, yellow line), Upper Middle Montney (UM, green line) and Lower Middle Montney (LM, thick red line), respectively. (B) The same as (A) but for the typical injection-induced earthquakes (IIE) that occurred during the HF period.

Fig. R5. The same as Fig. R4 except that the EHW (A) and IIE (B) occurred during the time period without any injecting operations (prepare, pressure testing, fracking or flowback).

- b) On a related note, the injection is stated as taking place between 11-13 July. Why are there events detected before this in Table S1? This is not mentioned but seems extremely important to me! Are there any other HF or disposal wells nearby that any of these events could be related to? Or could they be natural? And to a lesser extent, what about the events months after injection- this is uncommon for HF induced seismicity? If some of these events are natural it could significantly alter the interpretations.

We note that both EHWs and typical induced events are detected before the HF stimulation. It is possible that they are related to three other co-located wellbores being active since 2013. Please see our response to Comment #5 for further details.

As shown in Fig. 2A, the hydraulic fracturing treatment took place between Jul 11, 2015 - Jul 20, 2015. The number of both EHWs and typical induced events are both the highest during this period. The number of typical induced events is relatively higher during the preparation and testing period, the flowback period, and the period between the HF and the flowback. The number of EHWs is not statistically significant due to the limited number of detections.

- c) EHW detection – is the STA/LTA designed so that only EHW events are detected, or do typical earthquakes also get picked up? How do you determine whether these events are truly EHWs or whether they are typical events? I would think a frequency band of 10-20 Hz should pick up both types of events, why not use a lower frequency for the EHWs? I get the impression that the distinguishment is maybe based mainly on the long “ringing” coda, but you have stated that this is a path and not a source effect, so separating event types based on this seems unintuitive.

We have rephrased the text in the “EHW Detection” section to make it clearer that the detection process entails two steps. In Step 1, we apply the general STA/LTA detection that identifies a variety of seismic signals, including both induced and natural earthquakes. We then visually identify an EHW cluster of events among the detections with waveforms showing the following two characteristics: (1) an impulsive broadband onset with visible P- and S-phase arrivals, but with slightly broader pulse widths compared to typical induced events in the same area, and (2) a sustained lower-frequency ringing (~10 seconds; < 5-7 Hz) after the body-wave phases. Such identified events are

named “earthquakes characterized by hybrid-frequency waveforms” or “EHWs”. Indeed the visual identification criteria may not be entirely quantitative. However, if the interpretation is correct that EHWs represent a continuum of waveform types representing a continuum of slip (similar to volcanic hybrid counterparts), some qualitative assessment of waveforms will be necessary. Because the main purpose of this paper is to document EHWs and try to understand the physical mechanism responsible for their seismogenesis. The main conclusions will not be obscured even if the entirety of the hypothesized continuum of events is not documented.

In Step 2, we apply a Multi-station Matched-Filter (MMF) detection to find additional EHWs. The detected EHW waveforms in the first step are used as templates.

We revised introduction (Lines 93-100) and the “EHW Detection” section by adding the above information (Lines 520-526).

- d) It would be interesting to compare locations of the EHWs with locations of typical events, and this may provide further insights on source mechanisms, how path effects may play a role etc depending if there are any patterns. A figure showing this would be nice, although I appreciate the other catalogue was published in an earlier paper.

We added the locations of typical induced events reported in Yu et al. (2019) to Fig. 2B. The typical induced events also show three comparable clusters, i.e., one proximal to the horizontal well, one to the south of the wellbore (co-located with the southern cluster of EHWs), and one near the MG08 station. The migration patterns of the EHWs and the typical induced events are also comparable (e.g., $D=0.2 \text{ m}^2/\text{s}$). The similar spatial distributions of the two types of events indicate that they may share common driving mechanisms.

- e) Also looking at the previous paper of the same monitoring campaign, there were events with similar low stress drops as the EHWs (or are they the same EHWs?).

In our previous study using the same dense seismic array (Yu et al., 2020), the low stress drop values ($\sim 1 \text{ MPa}$) of typical induced events are not EHWs. They only located in the proximity of the wellbores, mainly due to the higher stress change near the wellbore. We also marked the stress drop values of the typical induced events in Fig. 4A for better comparison. The f_c values of EHWs with similar M_0 are generally lower than that of typical induced events. The differences are greater at proximal distances to the well (see in the inset of Fig. 4A).

Minor comments:

11. L20 – “several kms away” – not apparent whether it would act over these distances.

The well-event distance could reach 1-2 kms (e.g., the induced event sequences in Bao and Eaton, 2016). Here we revised it as “a few kilometers away” (Line 19).

12. L47 – “distances of several km” -> “distances of up to several km”.

This sentence has now been replaced with “some observations of induced seismicity migration speeds outpace pore pressure diffusion” (Lines 50-51).

13. L112-116 – why the diffusivity

We add the following transitional sentence “The EHWs exhibit a migration pattern that could help evaluate the causal relationship between the fluid flow and the origin time of EHWs.” in Lines 156-157.

14. L161: “encounter” -> “encounters” Similarly, rupture

We revised it accordingly (Line 236).

15. L204-205: “speeds of the southern EHW cluster are generally higher than those closer to the wellbore, suggesting a positive correlation with distance to the wellbore.” – possible, but given only two clusters its difficult to draw firm conclusions as it could just be something else about the locations of the two different clusters – is there any evidence of this for events within each cluster? Also see main comments.

We agree that the indication of “a positive correlation” is not well-supported only based on three groups. Thus, we revised this sentence in Lines 277-281 as “The stress drop of EHWs near the wellbore is lower than that at further distances, which is consistent with the trend of typical induced earthquakes (Yu et al., 2020). Moreover, stress drop values of the further clusters (near MG08 and the southern cluster) show a wider range between 0.03 and 1 MPa, suggesting a wider slip continuum.”

16. L217-218: “Either case is thought to be the seismic manifestation of slow slip in aseismic to seismic fault transition zones” – this sentence does not really make sense, by definition slow slip = slow rupture, a manifestation of this may be the interpretation of low stress drops too.

This sentence has been removed. We revised the first paragraph of “Discussion” to highlight that the co-location of EHWs and typical induced events are the key that

distinguishes our study from previous stress drop observations (see more details in Comments #2).

17. L252-254: “Fluid migration is supported by the abnormally high hydraulic diffusivity ($\sim 0.2 \text{ m}^2/\text{s}$) observed in the tight shale formation between the wellbore and the southern cluster (Fig. 2B) (13).” – I would think that an abnormally high diffusivity may suggest that a different mechanism than fluid migration may be the cause?

We agree, and that is why we have added a modeling test to demonstrate the necessity to invoke the role of aseismic slip loading in explaining the observations. We refer to our response to Comment #1 of the second Reviewer.

18. L267-269: “One possible difficulty in the detection of EHW is that the characteristic low-frequency, low-amplitude ringing will be quickly buried by body wave dispersion and coda noise when recorded on stations at regional distances from injection wells.” – but my understanding of your conclusions is that this “ringing” is a path effect and therefore not indicative of the source process. The events shown here still have a broad frequency onset that should be just as easily detected for events of the same magnitude at distal stations. More important are events with low corner frequencies, and this is not necessarily a common technique, and rarely analyzed for enough (or even any) events to see such a pattern.

The possible difficulty in the EHW detection at distal stations doesn't mean that the signal is totally buried, but that their waveforms would be difficult to differentiate from those of typical induced events if the low-frequency portion of the waveform is dispersed or exhibits low amplitudes near or below the background noise. Fig. R6 shows an example of EHW waveforms recorded at epicentral distances of 10s and 100s of km. We can still observe the P and S arrivals clearly, while the hybrid frequency content is ambiguous, therefore the larger epicentral distances mask the distinguishing features of the waveforms.

Nevertheless, there are emerging reports of long-period transients similar to EHWs at stations in close proximity to injection (Milkereit et al., 2017; Zang et al., 2017).

Figure R6. Representative EHW waveforms recorded at regional distance. The exemplified EHW occurred on July 26, 2015 02:18:55 (UT) (A) 3 component waveforms recorded on station NBC5 (Epicenter distance = 33.6 km). Theoretical P and S arrivals are marked. (B) Same as (A) but for recordings on station NBC6 (Epicenter distance = 273 km).

19. L397-401: Basically, an argument for why get lower frequencies in a fluid filled, heterogeneous environment. Maybe this should be considered in terms of interpreting lower corner frequencies for the events. I know that the EGF method in theory removes the path effect but given the relatively small study region some of the EGFs seem to be a significant distance from the events analyzed (0.7km), i.e. beyond the one source dimension suggested by Abercrombie (2015) for EGFs given the low event magnitudes. Given the HF and related pore pressure changes are isolated to relatively small regions, path effects could change significant over a small distance and this could also be a contributing factor in the results?

Here we require the inter-event distance of <1 km (accounting for location error) as one of the criteria for event pair selection. But the decisive factor that ensures the co-location assumption is satisfied is the full waveform similarity of the event pair as exhibited by the cross-correlation coefficient (6 s window length waveform cross-correlation coefficient higher than 0.7), applying a bandpass filter of 1-20 Hz. According to the results of previous studies, the corner frequency estimation is considered stable when the CC threshold is set at 0.7 or higher (Abercrombie et al., 2017; Ruhl et al., 2017).

We revised the related description accordingly in Line 577-583.

20. Figure 1A: Too small, some parts are very difficult to see (e.g. grey rectangle, all red stars etc.). and some labels (e.g. top-right inset)

We modified Fig. 1A with clearer symbols and moved the top-right inset into Fig. S1.

21. Figure 2B: Define “distance”, i.e. from first event.

We marked the initial point (the first event) in Fig. 2B with a gray arrow. Detailed information is now provided in “EHW Migration” section.

22. L653: “Thick blue curve: HF well” -> “Thick blue lines: HF wells”

We revised it accordingly.

23. Figure 3C; S4: Y-axis label “duratuon” -> “duration”

We revised it accordingly.

24. L663: “displacement of two events” -> “displacement of the two events”

It has been removed.

25. Figure 4: Why show typical induced event in (A) but not (B)?

We assumed a constant rupture velocity for typical induced events, which equals to 90% of shear wave velocity (0.9β , Sato and Hirasawa, 1973; Imanishi et al, 2014; Huang et al., 2016; Yu et al., 2020). We add the reference of the constant rupture speed (2.33 km/s) in Fig. 4B.

Supplementary Materials:

26. L34: “short duration of low-frequency” -> “short duration of the low-frequency”.

We revised it accordingly (Line 42).

Reviewer #2 (Remarks to the Author):

Thank you for the opportunity to review this manuscript. It is a very interesting study that describes and analyzes a new, potentially important signal of induced shear slip during HF stimulation. Overall, the data analysis is robust and rigorous, but the wording, interpretations, and figures could be strengthened to emphasize/clarify the significance of the results. Specific comments are provided in the annotated .pdf and several general comments are provided below.

1. I am not sure about the idea that the EHW ‘bridges’ the gap between inferred aseismic slip and seismic slip at greater distances. I understand the intent, but another way to approach this would be to directly address the physical processes that operate at different distances. For example, the authors could outline 3 regimes: wellbore, near-wellbore, and far away from wellbore and discuss how each of the mechanisms described in the discussion may or may not play a role.

After carefully examining the EHWs spatial distribution, we now interpret the EHW mechanisms in the context of three (newly) separated groups of varying proximity to the wellbore: 1) directly next to the well bore, 2) near the station MG08, and 3) in the southern cluster. The grouping helps highlight changes in source parameters of EHWs with distance that are also consistent with observations of typical induced events. We also add a new Fig. S3 to provide four pairs of representative examples between the co-located EHWs and typical induced events. The new plot shows the quality of spectral ratio fitting, as well as the differences in source parameters between the two.

To evaluate the role of different physical processes suggested by the observations that related to the HF injections in generating EHWs, we have added a numerical modeling test to calculate the evolution of coupled pore pressure and poroelastic stress. The modeling test allows us to reject the null hypothesis that the classical mechanisms of coupled pore pressure and poroelastic stress change are sufficient to account for the temporal and spatial distribution of EHWs. Thus, the most viable mechanism to generate them is by triggering through aseismic slip.

We use the COMSOL Multiphysics software to perform the modeling (see details in Method). In our layered model (Fig. 5A), we embed a 300 m shale layer to represent the Montney formation. To match the migration of seismicity, we must assume the permeability within the shale layer to be $1 \times 10^{-14} \text{ m}^2$, equivalent to a hydraulic diffusivity of $0.2 \text{ m}^2/\text{s}$. It is much higher than values reported for tight shale formations ($10^{-23} \sim 10^{-16} \text{ m}^2$) (Neuzil, 1994; Flewelling and Sharma, 2014), indicating that the southern/near MG08 clusters are probably hydraulically connected to the

volume near the wellbore (Bao and Eaton, 2016; Peña Castro and Roth et al., 2020). Our simulation applies injection information of real HF stages (Table S2). To evaluate the triggering capacity for microseismicity, we require the Coulomb failure stress change (ΔCFS) to be at least ~ 0.1 bar (King et al., 1994; Stein, 1999; Sumy et al., 2014). Similar values are used by previous efforts to study dynamic triggering in the WCSB (Wang et al., 2015; 2019).

We specifically look into the evolution of ΔCFS at three locations as near the horizontal wellbore, the center of the fitted southern structure and near station MG08, respectively (Figs. 5B-D). Since we don't have robust focal mechanism solutions of EHWs, we alternatively assuming the ruptures as (a) near the wellbore: normal slip along fault striking along the $S_{H\text{max}}$ (strike = 60° , dip = 90° , rake = 0°), (b) near station MG08: thrust slip along the preferred striking (strike = 30° , dip = 90° , rake = 90°), (c) at the center of the fitted structure: thrust slip along the fitted plane (strike = 150° , dip = 66° , rake = 90°).

Accordingly, the ΔCFS proximal to the wellbore caused by coupled pore pressure and poroelastic stress change is ~ 10 bars and sufficient to induce EHWs (Fig. 5C). At further distances, near the station MG08 (Fig. 5B) or the source location of the southern EHW cluster (Fig. 5D), it takes about 50 days for ΔCFS to reach the value of 0.05 bar because of the slow build-up of pore pressure change. The ΔCFS thereafter may be eventually capable of triggering those further EHW clusters. However, the eight EHWs that occurred within the first 50 days are unlikely to be induced by the low values of ΔCFS (0.01~0.04 bar). Our modeling results indicate that the coupled effect of pore pressure and poroelastic stress change is insufficient to induce those "early" EHWs at further distances, even as an abnormally high permeability is assumed to allow the relatively fast fluid diffusion. An alternative mechanism(s) is required to make a comparable contribution to the ΔCFS .

Our new effort of numerical modeling is presented in Lines 336-406 (Discussion), Lines 768-815 (Method), Fig. 5 and Tables S3.

Based on the modeling result, we propose that aseismic slip loading may be a viable mechanism to help explain the early occurrence of EHWs at further distances (Lines 407-422). Moreover, the build-up process of ΔCFS along the planar structure outlined by the southern cluster also favors the rupture nucleation of seismic events, including EHWs (Lines 423-428).

2. One important idea missing from the manuscript is a discussion of effective normal stress. It is somewhat implied in the discussion of dilatant strengthening and slip on

non-critically stressed faults, but it should be directly stated that low effective normal stresses can be achieved through active pumping near the wellbore or into a feature (e.g., southern cluster fault) that is hydraulically connected to the near wellbore region.

As the reviewer pointed out, since the active pumping would lower the effective normal stress through pore pressure increase, it could reactivate faults/fractures near the wellbore or on a structure that is hydraulically connected to the volume near the wellbore (e.g., the fault outlined by the southern cluster). As slip accelerates, shear deformation accompanied by dilation causes newly fractured rock volume (Guglielmi et al., 2015; Cappa et al., 2018; Segall et al., 2010), which if hydraulically poorly connected to the ambient rock would temporarily reduce the pore pressure, and hence increase the effective normal stress. Consequently, the strengthened fault tends to inhibit seismic slip or equivalently, favors aseismic/slow slip.

We now also use the change of Coulomb failure stress (ΔCFS) to discuss the controlling mechanisms of EHWs. As presented in our response to the previous comment, the build-up process of ΔCFS along the plane outlined by the southern cluster can eventually lead to the nucleation of seismic events like EHWs, but the reduction in effective normal stress is not sufficient to overcome the residual fault strength in the early stage (Fig. 5). Therefore, we expect the ruptures to be stabilized initially (Garagash and Germanovich, 2012; Zoback et al., 2012). Once the stress perturbation further loaded the fault to overcome the residual fault strength, the fault would eventually become unstable.

The above discussion now appears in the Discussion (Lines 407-428).

3. There is a missed opportunity here to add in some simple, but potentially impactful geomechanical analysis. In Fig. 1, the direction of S_{Hmax} is indicated on the map, but the authors don't return to this idea when discussing the southern cluster fault plane. I was left wondering if the best fit plane was or was not critically stressed and what magnitude of pore pressure change was needed to induce slip? Adding this simple analysis and plotting the best fit plane on the map in Fig. 1 would help connect this study to the referenced geomechanical studies in the same area. In addition, comparing the estimate of the pore pressure change to the diffusion model could yield insights into the timing of the events and strengthen the authors interpretations of the formation diffusivity.

Thanks for the suggestion. We collected the information reported by the industrial operator to estimate the principal stress in our study area. However, different conventions used by operators lead to discrepancies in reported stress values, and

accordingly, an unrealistic geomechanical analysis. We provide more details in the following.

We calculate the value of principal stresses at the injection depth (~1.9 km) following Mahani et al. (2020). Specifically, we use the averaged instantaneous shut-in pressure (ISIP, 22.5 MPa) as S_{Hmin} , and, determined S_V to be 48.7 MPa from density logs. We then use a simplified approximation of the Hubbert and Willis' equation to estimate S_{Hmax} (Ervine and Bell, 1987), i.e., $S_{Hmax}=2*\text{leak-off pressure} - \text{pore pressure}$. In our case, the leak-off pressure is 40.7 MPa, and reservoir pore pressure (unperturbed) is 19 MPa (Eaton and Schultz, 2018). The estimated S_{Hmax} is then 62.5 MPa. The corresponding Mohr circle diagram is shown in Fig. R7 as black lines.

When we take the reservoir pore pressure into consideration, the Mohr circles move leftward, shown as the blue Mohr's circles in Fig. R7. The Coulomb failure criterion (with zero cohesion) is also marked as the red line. We use a typical value of 0.6 constrained for sedimentary rocks for the coefficient of internal friction, as documented in Zoback (2010). The Coulomb failure criterion crosses the blue Mohr circles, meaning that certain geometry of faults/fractures (with angles roughly greater than 30° from S_{Hmax}) would keep rupturing under the background stress regime. Considering the Montney area is seismically quiet before the drastically increase of HF injections, such geomechanical scenario seems unrealistic.

According to our personal communications with colleagues at the British Columbia Oil and Gas Commission, it turns out that there is no standard way for oil and gas companies to conduct and report pressure test results. The value of ISIP, for example, would be largely biased using different shutdown processes, including water hammer shutdown, slow shutdown of pumps (common to avoid excessive wear), the clean, sharp shutdown, etc. The reported values of pressure tests could thus be less reliable than other parameters such as injection volumes, injection rate, and operation time.

We note that the reported ISIP values in southern Montney (~150 km to the south of our study area) are in the range of 50–70 MPa (Mahani et al., 2020), much higher than our case. This independent evidence seems to also suggest that the reported ISIP in our case may be incorrect.

Therefore, we decide to focus on the modeling of the spatiotemporal distribution of stress perturbations due to multi-stage fluid injection, as detailed in Comment #1 of the second Reviewer.

Furthermore, we propose that aseismic slip loading may play an important role in facilitating the spatiotemporal distribution of EHWs. In this case, the constitutive response would be better described by the rate and state-dependent frictional law. A fixed value of the frictional coefficient ($\mu=0.6$ or some similar value) may not be the optimal choice.

Fig. R7. In-situ 3D Mohr diagram and coulomb failure criterion. Blue/Black circles: Mohr circles with/without considering the reservoir pore pressure. Blue/Black dots: the stress state along the southern fault structure outlined by EHWs inferred from borehole logs.

4. Is moment tensor inversion possible for this dataset? There are recent studies of HF seismicity that attempt to extract best fit planes and estimate principal stress directions (e.g., Kuang, W., Zoback, M., & Zhang, J. (2017). Estimating geomechanical parameters from microseismic plane focal mechanisms recorded during multistage hydraulic fracturing. *Geophysics*, 82(1), KS1–KS11. <https://doi.org/10.1190/geo2015-0691.1>). If possible, it would be interesting to perform the geomechanical analysis described in comment 3 on the EHWs and typical events.

Accurate determination of the moment-tensor solutions for small events ($M < 2$) is challenging without high-quality borehole seismic data. In our case, the quality of seismic waveforms recorded at the temporal surface broadband stations is limited. Nonetheless, we tried to use the open-source software *Grond* (Heimann et al., 2018) to estimate the moment tensors of EHWs. Unfortunately, we obtain no stable, satisfactory solution.

5. The discussion of EHWs in context of seismic hazard needs to be reworked. The purpose of HF stimulation is to induced slip on faults in order to enhance the surface

area of the pre-existing fracture network. In this context, there seem to be two conflicting objectives. If one was somehow able to reproduce the near-wellbore conditions of EHWs elsewhere in the formation, would the induced accomplish the necessary permeability enhancement? The authors could potentially recast Fig. 4A in terms of source dimension and slip in order to say something about whether aseismic slip achieves the same scale of surface area creation. On a related note - If the dataset of EHWs was larger, it would be interesting to see if the Gutenberg-Richter distribution was similar or different to that of the typical events.

Thanks for the inspiration. We add the following discussion in Lines 449-454 “The purpose of HF stimulation is to enhance permeability of the tight shale layer by enlarging the surface area of the fracture network. In this context, the source process of EHWs can be more efficient and safer in accomplishing the objective of HF stimulation due to their relatively larger rupture area and slower rupture speed (Fig. 4), compared to the typical induced events with similar magnitudes.”

It is indeed interesting to check the Gutenberg-Richter distribution of EHWs when it is possible. In a typical tectonic environment, it has been reported that slow earthquakes follow consistent Gutenberg-Richter distribution with b-value equal to 1 (e.g., Wech et al., 2010; Michel et al., 2019; Dal Zilio et al., 2020), based on both observational and simulated results. Unfortunately, the number of samples in our EHW catalogue is far too few to provide a meaningful G-R relationship.

Comments from the annotated .pdf

6. L19: “moderate” -> “moderately sized”.
We revised it as moderate-sized (Line 19).
7. L20: “hydraulic fracturing stimulations” -> “hydraulic stimulations”.
We revised it accordingly (Line 20).
8. L21: “aseismic/slow slip signals” -> “signals of aseismic slip”.
We revised it as “aseismic slip signals” (Lines 20-21).
9. L22: “a new type of induced seismic signal” -> “a new type of signal”.
We revised it accordingly (Line 22).
10. L23-25: “Earthquakes characterized by hybrid-frequency waveforms (EHWs) differ from ordinary induced earthquakes..” -> “Earthquakes characterized by these hybrid-frequency waveforms (EHWs) differ from typical induced earthquakes..”.
We revised it accordingly (Lines 23-25).

11. L27-28: “The characteristics described above are identical to low-frequency earthquakes found in plate boundary zones.” -> “The characteristics of EHWs are identical to those of low-frequency earthquakes found in plate boundary zones.”

We revised it as “The source characteristics of EHWs are identical to those of low-frequency earthquakes found in fault transition zones.” (Line 27-28).

12. L28-30: The final sentence of the abstract is not clear. The EHWs are described as signals of aseismic slip, but the authors don’t identify here whether they occur close to or far away from the wellbore. The authors also do not describe how the aseismic slip relates to seismic slip at greater distances.

Laboratory and experimental work, including larger meso-scale experiments, have shown that injection loading can generate a continuum of slip behavior ranging from aseismic near the wellbore to seismic at further distances. In this paper, we document the observation of EHWs and propose that they may be the manifestation of slow ruptures caused by the HF injection. Unfortunately, we did not observe a continuum of slip behavior due to the limited observational resolution. We revised this sentence (Line 30-31) as “EHWs could thus represent the manifestation of slow rupture transitioning from aseismic to seismic slip.”

13. L33: “can induce” -> “induces”.

We revised it accordingly (Line 35).

14. L34-36: The sentence “The most common ... United States.” needs revision (e.g., M4+ events result from wastewater disposal).

We revised the sentence as “The most common perception is arguably that the M4+ events generally result from large fluid volumes related to wastewater disposal, particularly in the Central and Eastern United States (Ellsworth, 2013).” (Lines 36-38).

15. L46: “volumes” -> “volumes compared to wastewater disposal”.

We revised the sentence in Lines 51-53 as “Moreover, the relatively small volumes of total injected fluids are often likely insufficient or unable to generate significant poroelastic stress perturbations (Peña Castro and Roth et al., 2020; Eyre et al., 2019b).”

16. L45-50: It is unclear why/how injection volume/time or the occurrence of distant earthquakes precludes the possibility of M3+ event.

Please refer to our response to Comment #1 of the third Reviewer.

17. L55: “(e.g., high content of clay and total organic carbon)” reference needed here:
Kohli, A. H., & Zoback, M. D. (2013). Frictional properties of shale reservoir rocks.
Journal of Geophysical Research: Solid Earth, 118.
<https://doi.org/10.1002/jgrb.50346>.

Thanks. The reference is added accordingly (Line 62).

18. L71: “prolific” -> “common”.

We revised it as “commonplace” (Line 89).

19. L72: “Montney Play” -> “Montney shale”?

It has been revised as “Montney Shale formation” (Lines 91-92).

20. L73: “manifest” -> “represent”.

We revised it accordingly (Line 93).

21. L75: Remove “that are described by”.

We removed it accordingly (Line 94).

22. L77-78: Remove “(later referred to simply as “typical” induced events)”.

We removed it accordingly (Line 96).

23. L80: “for” -> “for these”.

We replaced this sentence with “In the following, we document and quantify the distinctive features of the EHW waveforms.” (Line 100-102)

24. L82: “...between the HF injection and the distribution of the EHWs.” -> “...between the injection and the timing of the EHWs”.

It has been revised (now in Lines 102-103) as “between the injection and spatiotemporal distribution of the EHWs”.

25. L83: “from source or path effects” -> “from source or from path effects”.

We revised it accordingly (Line 104).

26. L85: “further look into” -> “investigate”.

We revised it accordingly (Line 106).

27. L87: “At last” -> “Last”.

We revised it accordingly (Line 109).

28. L94: “surrounding” -> “around”.

We revised it accordingly (Line 117).

29. L96: Remove “potentially”.

We revised it accordingly (Line 119).

30. L98: “local to” -> “near”.

We revised it accordingly (Line 121).

31. L99: “constrained” -> “well constrained”.

We revised it accordingly (Line 141).

32. L106: “specific” -> “common”.

This sentence has been removed. Please refer to our response to Comment #10 (a) of the first Reviewer.

33. L107-109: Remove “which only occurs ... (see Text S2).”

This sentence has been removed.

34. L106-111: Are any of the typical events contemporaneous with the EHWs?

This part has been removed. Please refer to our response to Comment #10 (a) of the first Reviewer.

35. L124: “while other” -> “other”.

We revised it as “The second” (Line 175).

36. L127: “and thus independent of the duration of brittle failure” Not sure about this statement. The characteristics of brittle failure impact fault hydrology (e.g., wear, fracturing, etc.).

Here the “duration” refers to “the duration of low-frequency ringing”. We revised the sentence (now in Lines 176-178) as “In the latter interpretation, the ringing duration is thought to be dictated by the pressure gradient across the crack, and is independent of the duration of brittle failure.”

37. L130-131: Move the last sentence to next paragraph.

We revised it accordingly (Lines 182-183).

38. L161: “encounter” -> “encounters”.

We revised it accordingly (Line 236).

39. L229: “rupture” -> “rupture characteristics”.

This sentence no longer exists.

40. L230: “The intended purpose of ...”. This is somewhat misleading as HF stimulation is intended to create slip on pre-existing fault/fracutres.

This sentence no longer exists.

41. L227-237: This paragraph needs to be reworked. The description of the near-wellbore mechanism is confusing. What is missing is any discussion of the effective normal stress and how it might vary with hypocentral distance.

Please refer to our response to Comment #2 of the second Reviewer.

42. L239: “paced” -> “being outpaced”.

This sentence no longer exists.

43. L242: Add reference: Zoback, M. D., Kohli, A. H., Das, I., & McClure, M. (2012). The importance of slow slip on faults during hydraulic fracturing stimulation of shale gas reservoirs. Society of Petroleum Engineers. <https://doi.org/10.2118/155476-MS>.

We revised it accordingly (Line 426).

44. L250-251: This is confusing. It seems appropriate here to discuss the idea of low effective normal stress near the wellbore due to active pumping. This is needed for both mechanisms and explains how non-critically stressed faults may slip.

Please refer to our response to major Comment #2 of the second Reviewer.

45. L264-265: “the first clue to how injection operations can be controlled to keep sliding aseismic” Is this aligned with the goals of HF stimulation? Could the same permeability enhancement be achieved with only aseismic slip?

Please refer to our response to major Comment #5 of the second Reviewer.

46. Fig. 1: Rearrange figure (A) and caption to start description from largest map inset to main figure.

We have modified the top-right inset plot to better display the regional geography. The map showing the regional station distribution is now moved to Fig. S1. We also add the following sentence to the caption of Fig. 1A as “The top-right inset shows the geographic location of our study area (red rectangle).”

47. Fig. 2: Identify southern cluster events in (B).

We added a schematic map on the right panel to identify the southern cluster events.

48. Fig. 3: Label EHW/typical events clearly to emphasize the data. Create figure legend.

We have revised Fig. 3 accordingly. Please refer to our response to major Comment #1(b) of the first Reviewer for more details.

49. Fig. 4: Remove the purple bars in (A). Red border on S cluster points looks confusing, try using another symbol.

We have revised Fig. 4 accordingly. The red border symbols are replaced with purple ones.

Reviewer #3 (Remarks to the Author):

A – Report Summary

The authors observed 31 slow slip induced seismic signals related to an industrial HF treatment in the Montney Play, Canada. This is an interesting new catalogue of such events which are difficult to measure in deep field contexts. The authors then make a “classical” source analysis of these events, and compare to collocated induced seismic events. Their clearest finding is that there is a significant difference in the corner frequency. They then used this measured parameter to estimate EHWs source stress drop and rupture velocity showing some analogy with tectonic slow slip events in literature. To summarize, this study is robust but classical. The high merit of the authors is to bring to the scientific community a nice catalogue of exotic events (they call EHWs) which are well located around a hydrofracking treatment well. Unfortunately, they are very little information about the hydrofracking treatment (volume, protocol, time duration, etc) that could allow the community to eventually reanalyze these data. This is to our opinion a strong weakness of the paper. We would recommend this contribution more as a report for the quality of the observations, if they can be completed with some crucial information on the HF treatment.

We thank the reviewer for the helpful, constructive feedback, and have now included detailed information about the hydraulic fracturing treatment parameters, as well as constructed a coupled poroelastic model of stress evolution based on the parameters to support our observation and mechanism interpretations. Please refer to responses to Comments #8 and #10(a) from the first Reviewer and to Comment #1 from the second Reviewer for further details.

- What are the noteworthy results?

A catalogue of well localized and characterized hybrid frequency waveform seismic events. These events look like slow slip events observed in tectonic context. From this comparison, the author hypothesize that they might be proxies of the transition from aseismic to seismic slip around a fluid injection.

- Will the work be of significance to the field and related fields? How does it compare to the established literature?

The originality of the work is the catalogue of 31 events, and the industrial context of a real HF treatment. The data processing is not original, and authors provide relevant references.

- Does the work support the conclusions and claims, or is additional evidence needed?

Additional evidence is needed. There are not enough information about the hydraulic treatment.

- Are there any flaws in the data analysis, interpretation and conclusions? - Do these prohibit publication or require revision?

Interpretation and conclusion seem to mainly rely on one parameter, and thus may look overinterpreted.

- Is the methodology sound? Does the work meet the expected standards in your field?

The works meets the usual standards in the field of IS analyses.

- Is there enough detail provided in the methods for the work to be reproduced?

No. There are no information on the hydraulic source.

B - Additional detailed comments in the text

Introduction:

1. Lines 45 to 50 – “However, the combination of factors that are associated with HF injection, such as the relatively small total injected fluid volumes, the short injection time history, and the often nearly instantaneous onset of seismicity at distances of several kilometers from wellbores, makes it challenging to explain the occurrence of M3+ earthquakes with the classical concept of pore pressure and/or poroelastic stress change”. We do not see the clear link between the affirmations in this sentence and the bibliographic references. In addition, the fact that an industrial HF treatment involves a relatively small total injected fluid volume is not true.

We agree that the HF treatment volumes of injected fluid are not always relatively small. The situation we want to describe here is that an M3+ earthquake is induced where the injected volumes are generally too small to cause a sufficient poroelastic stress change (Peña Castro and Roth et al., 2020, Eyre and Eaton, 2019b). We also revised the references. We have revised the sentence in Lines 47-53 as “However, geomechanical considerations often make it challenging to explain the M3+ HF induced earthquakes with the classical concepts of pore pressure and/or poroelastic stress change. For example, some observations of induced seismicity migration speeds outpace pore pressure diffusion (Eyre and Eaton, 2019b). Moreover, the relatively small volumes of total injected fluids are often likely insufficient or unable to generate significant poroelastic stress perturbations (Peña Castro and Roth et al., 2020, Eyre and Eaton, 2019b).”

2. Lines 54-56 –“Under the optimal geomechanical (e.g., if the hosting fault is critically stressed) and compositional (e.g., high content of clay and total organic carbon) conditions, aseismic slip fronts can move ahead of the pore-pressure diffusion and may interact with nearby, larger faults to trigger significant events.” Here again, the fact that the compositional fault conditions may play a role in the preferential development of aseismic slip front ahead of the pressurized zone is not as clear as stated by the authors. The references cited do not really support the idea (they do not deny that there could be a link either).

We have removed the text that misleadingly linked the fault condition and the aseismic slip front ahead of the pore pressure diffusion, and added two references. The sentence now is revised (Lines 61-64) as “Under the optimal geomechanical (e.g., critically stressed host fault) and compositional conditions (e.g., high clay and total organic carbon content; Kohli and Zoback, 2013), aseismic slip fronts may interact with nearby, larger faults to trigger significant events (Eyre and Eaton, 2019b).”

Correlation of EHWs with HF treatment:

3. Line 82 – “We first check the correlation between the HF injection and the distribution of the EHWs.” We are not fully convinced by the correlation, may be because it is not enough documented in the paper (Text S2 is not bringing significant additional information to the main text). It would for example be interesting to get a time variation of the injected water volume, with the time occurrence of the EHWs and IS plotted.

Thanks for the suggestion. We have now added a new panel Fig. 2A to show the temporal distribution of EHWs and typical induced events. The new plot marks the HF operation dates and the injected volumes (in green dots). We have also added a new Table S2 to provide the corresponding parameters of the injection operations. We removed the test related to the origin hour of EHWs, as it is out of the scope of this study, and proved to be unrelated to injecting hours upon closer inspection. Please refer to our response to Comment #10 (a) of the first Reviewer.

4. Figure 2 – Dot color figuring the EHWs hour is not accurate enough to clearly the chronology of these events. The ‘classical’ IS events should be shown in the figure as it is in Fig. S2.

Fig. 2B has been replaced with the colored dots indicating the origin time of EHWs. The typical induced earthquakes from the same dataset have also been added. Please refer to our response to Comment #10 (a) of the first Reviewer for additional details.

5. Lines 112 – 116 – “As shown in Fig. 2B, the migration of EHWs mainly follows a fluid diffusivity value of ~ 0.2 m²/s (see in Methods), which is consistent with the estimation based on typical induced events (13). A few EHWs located near the station MG08 are compatible with a higher diffusivity value of ~ 1.5 m²/s, suggesting the existence of a fluid conduit that connects to the well.” Seems to me like a lot of interpretations here. Fig. 2B mainly shows a high dispersion of the plot...

Please refer to our response to Comment #8 of the first Reviewer.

EHWs Source mechanisms:

6. Lines 119 to 127 should be moved to the later discussion.

The main purpose of our study is to understand the mechanism responsible for these earthquakes characterized by hybrid-frequency waveforms (“EHWs”). The sentences to which the Reviewer refers briefly provide the background on how similar hybrid-frequency seismic signals are observed and characterized in volcanic environments. The introduction also draws parallels to how the different distinguishing features of the waveforms can be analyzed to infer their source mechanisms. They form an essential part of the motivation to study the consistency of the size-duration scaling relationships between the initial broadband onset and the extended low-frequency part of the waveform, and are one of the key features to delineate the physical mechanism responsible for the EHW signals.

7. Except the difference in the corner frequencies, it is not very clear that there is a difference between the EHWs and the classical earthquakes. The moment-duration scaling is not conclusive.

Please refer to our response to Comment #1 (d) of the first Reviewer.

8. Lines 146 – 162 - The apparent duration of the low-frequency portion of the EHW waveform is interpreted as related to geological heterogeneities and dispersion. This is mainly based on the poor correlation ($R = 60\%$) of the T_a vs hypocentral distance. Maybe the authors underestimate the possibility of fluid driven effect. Overall, the data make interpretations hard to achieve here.

We would like to point out that our interpretation of the apparent duration of the low-frequency ringing of the EHW waveforms (T_a) is not totally based on the positive correlation between T_a and hypocenter distance at station MG08 (R -squared value = 60%, the original Fig. S6, now Fig. S8 in the revised Supporting Materials). Instead, it is based on several lines of arguments.

First, we carefully exclude the possibility that the variation of T_a is governed by the focal mechanism (please see our response to Comment #4 of the first Reviewer for

more detail). Then, we hypothesize that the variation of T_a stems from path effects, as the recorded seismic waves with travel paths in heterogeneous, fluid-filled volumes would likely exhibit longer apparent ringing durations relative to recordings with travel paths in less fractured rock (Lines 218-229). We refer to our response to the Comment #3 of the first Reviewer for detailed reasoning, as well as Comment #4 of the first Reviewer for the proof that velocity heterogeneity is capable of generating the low-frequency ringing.

When the wave energy encounters a less pronounced low-velocity heterogeneity along the ray path, the protracted low-frequency ringing would likely disappear. Correspondingly, a positive correlation between the hypocentral distance and the T_a is observed at one specific station (MG08), which experiences the least influence from the inferred volumes of velocity heterogeneity (Fig. S8), suggesting that the extended low-frequency portion is likely the result of dispersion rather than pressure gradient-driven fluid flow in a crack (Lines 238-243).

EHW rupture characteristics:

9. Lines 171 – 181. These lines give details about a basic way to estimate rupture characteristics. They should shorten this paragraph only stressing on the fact that they use collocated classical IS events to compare ruptures between EHWs and IS.

We have revised this paragraph to avoid presenting too many technical details (Lines 247-269).

10. In scenario 1, they take the same rupture speed for both events types, which is a strong assumption. Nevertheless, it shows that EHWs stress drop values are statistically 16 times smaller than for typical events. In sentence of line 190, it is not clear if this is the stress drop of the collocated IS of the same field test or an average value from literature.

All the collocated typical induced earthquakes here are from the same field test. Their source characteristics are studied using the same dataset and method, and the results are already published (Yu et al., 2020). We have modified the text in Lines 266-267 and Line 277 to make clear that the results are from the same field deployment.

11. In the second case, it would be much more appropriate to compare to the stress drop estimated from the collocated events...In scenario 2, author assume the same stress drop between EHW and IS. This way they calculate a much smaller EHWs rupture velocity than IS, which is in the range of tectonically induced LFE. One question arises whether this paragraph about the EHW rupture characteristics should not be in

the discussion chapter since it relies on many assumptions and only one field estimated parameter which is the corner frequency f_c . In addition, it is not fully clear that the comparison is done with the IS rupture characteristics of that experiment or with more general IS characteristics from the bibliography.

The main reason that we present the “EHW Rupture Characteristics” as the last “Results” section, rather than in the “Discussion”, is to better explain the differences of source characteristics between EHWs and typical induced events in detail. It also naturally follows the logic flow after the “EHW Source Mechanism” section.

Discussion:

12. Lines 208 – 218 are more a summary of the results than a discussion. Line 217-218 is not really proved, unless the authors can exclude any fluid coupling effects on rupture (they arguments are not fully clear to do this since they rely mainly on a poor T_a vs hypocentral distance correlation).

The part (previously in Lines 208-218) is now shortened as in Lines 294-296 “We observe that the EHWs exhibit evidently longer source durations than the typical induced earthquakes (Fig. 2C), albeit over a limited magnitude range of $M_W \sim 0.5$ to 2.0.”

We agree that our original statement “Either case is thought to be the seismic manifestation of slow slip in aseismic to seismic fault transition zones.” is not well supported in the first paragraph of Discussion. We removed this sentence and reorganized the first paragraph by first stating that the co-location of the EHWs and typical induced events is the key observation that distinguishes ours from previous studies (Lines 296-335). Then we explain that the EHW signals can be best explained by the manifestation of lower stress drop values and/or slower rupture speeds of the source process that are fundamentally different from the typical induced events (more details in our response to the next Comment #13).

13. Lines 219-226 – I think that the author make a strong shortcut in their discussion by writing that these observed EHWs mark the boundary between aseismic and seismic slip just because they have common features with tectonic slow slip events.

We have now significantly strengthened this part of discussion.

We first verify the role of aseismic slip loading in inducing EHWs by conducting geomechanical modeling. Our results show that the classical mechanisms of coupled pore pressure and poroelastic stress change are insufficient to address the distribution of EHWs (Lines 336-406). Please refer to our response to Comment #1 of the second Reviewer for more details.

Next, we propose that aseismic slip loading could be a viable mechanism (Lines 407-422). The dilatant strengthening effect could help explain the early occurrence of the induced EHWs at further distances. Besides, the build-up process of ΔCFS along the plane outlined by the southern cluster also favors the rupture nucleation of seismic events, including EHWs. (Lines 423-428).

Moreover, we point out that the stress drop and rupture speed of EHWs along the fitted southern plane exhibit wide ranges from typical low-frequency earthquakes to seismic rupture behaviors (Fig. 4). We also discuss the common source features between EHWs and those tectonically driven slow slip phenomena (Lines 429-446).

Finally, we suggest that EHWs could be the candidate of slow ruptures transitioning aseismic slip to seismic rupture.

14. Lines 229-232 – Pore pressure reduction inducing dilatant strengthening might depend a lot of the treatment injection rate. The authors should give information about the HF treatment adopted in their field study.

It is indeed true that a higher injection rate leads to stronger dilatancy effects. With the injection rate of $\sim 9 \text{ m}^3/\text{min}$, the pressure front of aseismic slip loading could propagate twice as fast as the diffused fluid (Yang and Dunham, 2020). We added this in Lines 417-419.

The complete information of HF treatments is now given in Table S2. The injection parameters are also adopted in our stress evolution modeling.

15. Line 252 – The hydraulic diffusivity value shown here was calculated from the fitting of a simple diffusivity model to the epicentral distance vs time plot. It would much better to use the hydraulic data to confirm or cross-check this high value.

The hydraulic diffusivity of $0.2 \text{ m}^2/\text{s}$ is equivalent to a permeability of $1 \times 10^{-14} \text{ m}^2$ (estimated using Eqs. 7-8 from Yu et al., 2019). It is much higher than values reported for tight shale formations ($10^{-23} \sim 10^{-16} \text{ m}^2$) (Neuzil, 1994; Flewelling and Sharma, 2014), indicating that the southern/near MG08 clusters are probably hydraulically connected to the volume near the wellbore (Bao and Eaton, 2016; Peña Castro and Roth et al., 2020). We added this in Lines 342-347.

Reference

- Abercrombie, R. (1995). Earthquake source scaling relationships from 1 to 5 ml using seismograms recorded at 2.5-km depth: *Journal of Geophysical Research: Solid Earth*, 100, 24015–24036.
- Abercrombie, R., and Leary P. (1993). Source parameters of small earthquakes recorded at 2.5 km depth, Cajon Pass, southern California: implications for earthquake scaling: *Geophysical Research Letters*, 20, 1511–1514.
- Abercrombie, R.E., Poli, P. and Bannister, S. (2017). Earthquake directivity, orientation, and stress drop within the subducting plate at the Hikurangi Margin, New Zealand. *Journal of Geophysical Research: Solid Earth*, 122(12), pp.10-176.
- Aki, K., Fehler, M. and Das, S. (1977). Source mechanism of volcanic tremor: Fluid-driven crack models and their application to the 1963 Kilauea eruption. *Journal of volcanology and geothermal research*, 2(3), pp.259-287.
- Babaie Mahani, A., Esfahani, F., Kao, H., Gaucher, M., Hayes, M., Visser, R. and Venables, S. (2020). A systematic study of earthquake source mechanism and regional stress field in the southern Montney unconventional play of northeast British Columbia, Canada. *Seismological Research Letters*, 91(1), pp.195-206.
- Bao, X., and Eaton D. W. (2016). Fault activation by hydraulic fracturing in western Canada: *Science*, 354, 1406–1409.
- Bean, C.J., De Barros, L., Lokmer, I., Métaxian, J.P., O'Brien, G. and Murphy, S. (2014). Long-period seismicity in the shallow volcanic edifice formed from slow-rupture earthquakes. *Nature Geoscience*, 7(1), pp.71-75.
- Bean, C., Lokmer, I. and O'Brien, G. (2008). Influence of near-surface volcanic structure on long-period seismic signals and on moment tensor inversions: Simulated examples from Mount Etna. *Journal of Geophysical Research: Solid Earth*, 113(B8).
- Cappa, F., Guglielmi, Y., Nussbaum, C. and Birkholzer, J. (2018). On the relationship between fault permeability increases, induced stress perturbation, and the growth of aseismic slip during fluid injection. *Geophysical Research Letters*, 45(20), pp.11-012.
- Chouet, B.A. (1996). Long-period volcano seismicity: its source and use in eruption forecasting. *Nature*, 380(6572), pp.309-316.
- Clarke, H., Verdon J. P., Kettleby T., Baird A. F., and Kendall J.-M. (2019). Real-Time Imaging, Forecasting, and Management of Human-Induced Seismicity at Preston New Road, Lancashire, England: *Seismological Research Letters*, 90, 1902–1915.
- Clerc F., Harrington R. M., Liu Y., and Gu Y. (2016). Stress drop estimates and hypocenter relocations of induced seismicity near Crooked Lake, Alberta, *Geophys. Res. Lett.* 43, no. 13, 6942–6951.
- Dal Zilio, L., Lapusta, N., & Avouac, J.-P. (2020). Unraveling scaling properties of slow-slip events. *Geophysical Research Letters*, 47, e2020GL087477.
- Denlinger, R.P. and Moran, S.C. (2014). Volcanic tremor masks its seismogenic source: Results from a study of noneruptive tremor recorded at Mount St. Helens, Washington. *Journal of Geophysical Research: Solid Earth*, 119(3), pp.2230-2251.

- Duitama-Leal, Alejandro, Almanza, Ovidio, & Montes-Vides, Luis. (2016). Modeling attenuation and dispersion of acoustic waves in porous media containing immiscible non viscous fluids. *DYNA*, 83(199), 78-85. <https://dx.doi.org/10.15446/dyna.v83n199.56238>
- Eaton, D.W., Igonin, N., Poulin, A., Weir, R., Zhang, H., Pellegrino, S. and Rodriguez, G. (2018). Induced seismicity characterization during hydraulic-fracture monitoring with a shallow-wellbore geophone array and broadband sensors. *Seismological Research Letters*, 89(5), pp.1641-1651.
- Eaton, D. W., and R. Schultz (2018). Increased likelihood of induced seismicity in highly overpressured shale formations, *Geophys. J. Int.* 214, no. 1, 751–757, doi: 10.1093/gji/ggy167.
- Ellsworth, W.L. (2013). Injection-induced earthquakes. *Science*, 341(6142).
- Ervine, W. B., & Bell, J. S. (1987). Subsurface in situ stress magnitudes from oil-well drilling records: an example from the Venture area, offshore eastern Canada. *Canadian Journal of Earth Sciences*, 24(9), 1748-1759.
- Eyre, T.S., Eaton, D.W., Zecevic, M., D’Amico, D. and Kolos, D. (2019a). Microseismicity reveals fault activation before M w 4.1 hydraulic-fracturing induced earthquake. *Geophysical Journal International*, 218(1), pp.534-546.
- Eyre, T.S., Eaton, D.W., Garagash, D.I., Zecevic, M., Venieri, M., Weir, R. and Lawton, D.C. (2019b). The role of aseismic slip in hydraulic fracturing–induced seismicity. *Science advances*, 5(8), p.eaav7172.
- Fehler, M. and Phillips, W.S. (1991). Simultaneous inversion for Q and source parameters of microearthquakes accompanying hydraulic fracturing in granitic rock. *Bulletin of the Seismological Society of America*, 81(2), pp.553-575.
- Flewelling, S.A. and Sharma, M. (2014). Constraints on upward migration of hydraulic fracturing fluid and brine. *Groundwater*, 52(1), pp.9-19.
- Fischer, T. and Guest, A. (2011). Shear and tensile earthquakes caused by fluid injection. *Geophysical Research Letters*, 38(5).
- Folesky J. Kummerow J. Shapiro S. A. Häring M., and Asanuma H (2016). Rupture directivity of fluid-induced microseismic events: Observations from an enhanced geothermal system, *J. Geophys. Res.* 121, no. 11, 8034–8047.
- Frankel, A. and Kanamori, H. (1983). Determination of rupture duration and stress drop for earthquakes in southern California. *Bulletin of the Seismological Society of America*, 73(6A), pp.1527-1551.
- Garagash, D.I. and Germanovich, L.N. (2012). Nucleation and arrest of dynamic slip on a pressurized fault. *Journal of Geophysical Research: Solid Earth*, 117(B10).
- Goertz-Allmann, B.P., Goertz, A. and Wiemer, S. (2011). Stress drop variations of induced earthquakes at the Basel geothermal site. *Geophysical Research Letters*, 38(9).
- Guglielmi, Y., Cappa, F., Avouac, J.P., Henry, P. and Elsworth, D. (2015). Seismicity triggered by fluid injection–induced aseismic slip. *Science*, 348(6240), pp.1224-1226.

- Harrington, R.M. and Benson, P.M. (2011). Analysis of laboratory simulations of volcanic hybrid earthquakes using empirical Green's functions. *Journal of Geophysical Research: Solid Earth*, 116(B11).
- Harrington, R.M. and Brodsky, E.E. (2007). Volcanic hybrid earthquakes that are brittle-failure events. *Geophysical Research Letters*, 34(6).
- Harrington, R.M., Kwiatak, G. and Moran, S.C. (2015). Self-similar rupture implied by scaling properties of volcanic earthquakes occurring during the 2004-2008 eruption of Mount St. Helens, Washington. *Journal of Geophysical Research: Solid Earth*, 120(7), pp.4966-4982.
- Heimann, S., Isken, M., Kühn, D., Sudhaus, H., Steinberg, A., Daout, S., ... & Dahm, T. (2018). Grond: A probabilistic earthquake source inversion framework.
- Holmgren, J.M., Atkinson, G.M. and Ghofrani, H. (2019). Stress drops and directivity of induced earthquakes in the Western Canada Sedimentary Basin. *Bulletin of the Seismological Society of America*, 109(5), pp.1635-1652.
- Hough, S.E. (2014). Shaking from injection-induced earthquakes in the central and eastern United States. *Bulletin of the Seismological Society of America*, 104(5), pp.2619-2626.
- Huang, Y., Beroza, G.C. and Ellsworth, W.L., 2016. Stress drop estimates of potentially induced earthquakes in the Guy-Greenbrier sequence. *Journal of Geophysical Research: Solid Earth*, 121(9), pp.6597-6607.
- Huang, Y., De Barros, L., & Cappa, F. (2019). Illuminating the rupturing of microseismic sources in an injection-induced earthquake experiment. *Geophysical Research Letters*, 46(16), 9563-9572.
- Ide, S., Beroza, G.C., Shelly, D.R. and Uchide, T. (2007). A scaling law for slow earthquakes. *Nature*, 447(7140), pp.76-79.
- Imanishi, K., Takeo, M., Ellsworth, W.L., Ito, H., Matsuzawa, T., Kuwahara, Y., Iio, Y., Horiuchi, S. and Ohmi, S. (2004). Source parameters and rupture velocities of microearthquakes in Western Nagano, Japan, determined using stopping phases. *Bulletin of the Seismological Society of America*, 94(5), pp.1762-1780.
- Julian, B.R. (1994). Volcanic tremor: Nonlinear excitation by fluid flow. *Journal of Geophysical Research: Solid Earth*, 99(B6), pp.11859-11877.
- Kane, D.L., Kilb, D.L. and Vernon, F.L. (2013). Selecting empirical Green's functions in regions of fault complexity: A study of data from the San Jacinto fault zone, southern California. *Bulletin of the Seismological Society of America*, 103(2A), pp.641-650.
- King, G.C., Stein, R.S. and Lin, J. (1994). Static stress changes and the triggering of earthquakes. *Bulletin of the Seismological Society of America*, 84(3), pp.935-953.
- Kohli, A.H. and Zoback, M.D. (2013). Frictional properties of shale reservoir rocks. *Journal of geophysical research: solid earth*, 118(9), pp.5109-5125.
- Kozłowska, M., Brudzinski, M.R., Friberg, P., Skoumal, R.J., Baxter, N.D. and Currie, B.S. (2018). Maturity of nearby faults influences seismic hazard from hydraulic fracturing. *Proceedings of the National Academy of Sciences*, 115(8), pp. E1720-E1729.

- Kwiatek, G., Bulut, F., Bohnhoff, M. and Dresen, G. (2014). High-resolution analysis of seismicity induced at Berlín geothermal field, El Salvador. *Geothermics*, 52, pp.98-111.
- McKean, S.H., Priest, J.A., Dettmer, J. and Eaton, D.W. (2019). Quantifying fracture networks inferred from microseismic point clouds by a Gaussian mixture model with physical constraints. *Geophysical Research Letters*, 46(20), pp.11008-11017.
- Michel, S., Gualandi, A., & Avouac, J.-P. (2019). Similar scaling laws for earthquakes and Cascadia slow-slip events. *Nature*, 574(7779), 522–526.
- Milkereit, C., Dahm, T., Cesca, S., Lopez, J., Nooshiri, N. and Zang, A. (2017). April. Long-period tilt-induced accelerations associated with hydraulic fracturing. In EGU General Assembly Conference Abstracts (p. 14472).
- Neuzil C. (1994), How permeable are clays and shales? *Water Resour. Res.*, 30, 145–150.
- Peña Castro, A.F., Roth, M.P., Verdecchia, A., Onwuemeka, J., Liu, Y., Harrington, R.M., Zhang, Y. and Kao, H. (2020). Stress Chatter via Fluid Flow and Fault Slip in a Hydraulic Fracturing-Induced Earthquake Sequence in the Montney Formation, British Columbia. *Geophysical Research Letters*, 47(14), p.e2020GL087254.
- Ruhl C. J. Abercrombie R. E., and Smith K. D. (2017). Spatiotemporal variation of stress drop during the 2008 Mogul, Nevada, earthquake swarm, *J. Geophys. Res.* 122, 8163–8180.
- Sato, T. and Hirasawa, T. (1973). Body wave spectra from propagating shear cracks. *Journal of Physics of the Earth*, 21(4), pp.415-431.
- Satoh, T. (2006). Influence of fault mechanism, depth, and region on stress drops of small and moderate earthquakes in Japan. *Structural Engineering/Earthquake Engineering*, 23(1), pp.125s-134s.
- Segall, P., Rubin, A.M., Bradley, A.M. and Rice, J.R. (2010). Dilatant strengthening as a mechanism for slow slip events. *Journal of Geophysical Research: Solid Earth*, 115(B12).
- Stein, R.S. (1999). The role of stress transfer in earthquake occurrence. *Nature*, 402(6762), pp.605-609.
- Sumy, D.F., Cochran, E.S., Keranen, K.M., Wei, M. and Abers, G.A. (2014). Observations of static Coulomb stress triggering of the November 2011 M5. 7 Oklahoma earthquake sequence. *Journal of Geophysical Research: Solid Earth*, 119(3), pp.1904-1923.
- Sumy, D.F., Neighbors, C.J., Cochran, E.S. and Keranen, K.M. (2017). Low stress drops observed for aftershocks of the 2011 Mw 5.7 Prague, Oklahoma, earthquake. *Journal of Geophysical Research: Solid Earth*, 122(5), pp.3813-3834.
- Uchide T., and Imanishi K. (2016). Small earthquakes deviate from the omega-square model as revealed by multiple spectral ratio analysis, *Bull. Seismol. Soc. Am.* 106, no. 3, 1357–1363.
- Wang, B., Harrington, R.M., Liu, Y., Kao, H. and Yu, H. (2019). Remote Dynamic Triggering of Earthquakes in Three Unconventional Canadian Hydrocarbon Regions Based on a Multiple-Station Matched-Filter Approach Remote Dynamic Triggering of Earthquakes in Three Unconventional Canadian Hydrocarbon Regions. *Bulletin of the Seismological Society of America*, 109(1), pp.372-386.

- Wang, B., Harrington, R.M., Liu, Y., Yu, H., Carey, A. and van der Elst, N.J. (2015). Isolated cases of remote dynamic triggering in Canada detected using cataloged earthquakes combined with a matched-filter approach. *Geophysical Research Letters*, 42(13), pp.5187-5196.
- Wech, A.G., Creager, K.C., Houston, H. and Vidale, J.E., 2010. An earthquake-like magnitude-frequency distribution of slow slip in northern Cascadia. *Geophysical Research Letters*, 37(22).
- Wu Q. Chapman M. C., and Chen X. (2018). Stress drop variations of induced earthquakes in Oklahoma, *Bull. Seismol. Soc. Am.* 108, no. 3A, 1107–1123.
- Yang, Y. and Dunham, E.M. (2020). Effect of Porosity and Permeability Evolution on Injection-Induced Aseismic Slip.
- Yu, H., Harrington, R.M., Kao, H., Liu, Y., Abercrombie, R.E. and Wang, B. (2020). Well proximity governing stress drop variation and seismic attenuation associated with hydraulic fracturing induced earthquakes. *Journal of Geophysical Research: Solid Earth*, 125(9), p.e2020JB020103.
- Yu, H., Harrington, R.M., Liu, Y. and Wang, B. (2019). Induced seismicity driven by fluid diffusion revealed by a near-field hydraulic stimulation monitoring array in the Montney Basin, British Columbia. *Journal of Geophysical Research: Solid Earth*, 124(5), pp.4694-4709.
- Zang, A., Stephansson, O., Stenberg, L., Plenkers, K., Specht, S., Milkereit, C., Schill, E., Kwiatek, G., Dresen, G., Zimmermann, G. and Dahm, T. (2017). Hydraulic fracture monitoring in hard rock at 410 m depth with an advanced fluid-injection protocol and extensive sensor array. *Geophysical Journal International*, 208(2), pp.790-813.
- Zhang, H., Eaton, D.W., Li, G., Liu, Y. and Harrington, R.M. (2016). Discriminating induced seismicity from natural earthquakes using moment tensors and source spectra. *Journal of Geophysical Research: Solid Earth*, 121(2), pp.972-993.
- Zoback, M.D. (2010). *Reservoir geomechanics*. Cambridge University Press.
- Zoback, M.D., Kohli, A., Das, I. and McClure, M.W. (2012). The importance of slow slip on faults during hydraulic fracturing stimulation of shale gas reservoirs. In *SPE Americas Unconventional Resources Conference*. Society of Petroleum Engineers.

REVIEWER COMMENTS

Reviewer #1 (Remarks to the Author):

Dear Editor and Authors,

I have gone through the paper again. Thank you to the Authors, who have done a very thorough job of addressing my and the other Reviewers' comments and gone through a substantial amount of work to improve the manuscript. In my opinion, the manuscript is far stronger now and closer to publication.

The Authors have now much more clearly presented, with stronger evidence, that there are two distinct event families here, and I believe that it is now more apparent that this difference in corner frequency is most likely caused by a difference in the rupture velocity. This in itself is a very interesting finding. The evidence for all of their results and interpretation is much more clearly presented and should be reproducible.

Nevertheless, in my opinion, the manuscript still has some important flaws in terms of the interpretation related to my original comments that in my opinion affects the potential impact of the paper for a broader audience, strong though it already is for the more specific field of induced seismicity. I do now believe that some relatively moderate changes could lead to a much stronger article and interpretation that is definitely worthy of publication in Nat Comm.

The key issues in terms of the interpretation are that it seems to me that there are three key observations that lead to the classification of these unusual events as "EHWs":

1. Low frequency coda.
2. Onset with lower corner frequency than typical.
3. Co-location of EHWs and typical events.

The Authors therefore identify the events due to their waveform characteristics above, i.e. events are identified due to their broadband onset with low frequency ringing coda. Events that appear like this are classically referred to as "hybrid" especially in volcanic environments, and therefore it seems reasonable to refer to these events similarly. However, the Authors then appear to discredit the importance of the low frequency coda, as they demonstrate that this is caused by propagation effects. I think it is this part of the manuscript that really causes some of the significant issues that I raised in my first review. For example, is it reasonable to identify events using an aspect of their waveform that is later discounted for being unrelated to the mechanism? Also, can we truly categorise these as "hybrid" events if they are principally events with low corner frequency, and the coda that in my opinion is key to allowing them to be referred to as hybrid is very much dependent on the site-specific propagation path? Is the one line of evidence (lower corner frequency) enough to support a different categorization of these events to typical induced seismicity (also mentioned by Reviewer 3), where some events of lower stress drops have already been observed in other locations (albeit interpreted as due to different causes)?

I should state here that I very much support the Authors interpretations that the low corner frequency is caused by slow rupture and that the coda is principally a propagation effect. The Authors have done a good job of presenting those arguments. However, in my opinion, the Authors are missing a key link in this conundrum that could strengthen their interpretation:

- Why do the low corner frequency events appear to always generate low frequency, ringing coda? If it does always happen, then this gives a much stronger argument for calling them hybrid events.
- Why don't the co-located, typical induced events generate this coda? If it's a path effect then we should observe it for both? I don't think it is related to the source frequency as typical events should also have significant low frequency energy (i.e. Brune source model).

In my opinion, if the Authors could present a stronger scientifically-reasoned interpretation that solves the above conundrum, they could more legitimately suggest that they have observed "hybrid" events, as the current explanation only partly addresses this. This is therefore my key suggestion for the paper. Admittedly, it's a complicated conundrum to solve; however, I have come up with one possible explanation

that I can suggest that the Authors could perhaps better develop to suit their interpretations:

Events with low rupture velocity have been hypothesized to occur in regions of low effective stress (Passelegue et al 2020), with low shear wave velocity (Johnson 1978), or with certain rock properties (Kohli and Zoback, 2013; Eyre et al., 2019). So therefore, according to the Authors' interpretation of low rupture velocity, the EHWs probably occur in formations with high pore pressure and/or in weaker (shale?) formations. All of these conditions are likely in unconventional reservoir formations (Montney formation is known to be overpressured in many regions). The overpressure will also increase during fluid injection. Both of these conditions (high pore pressure, weaker) suggest lower seismic velocities. So perhaps the waves are getting trapped within this low-velocity layer where they may be generated, leading to the "resonating" coda, as proposed by Bean et al. (2008) for shallow low-velocity layers and others (citations 24-28). This fits very nicely with the Authors' explanation in lines 216-229. For regional arrays such as these, depth errors for seismic event locations are usually quite high ($\gg 500$ m) in comparison to the thicknesses of the individual geological formations (of the order of ~ 100 m). The "collocated" IIEs could therefore be happening in different formations just above or below these units with different properties/stress conditions and with different raypaths that are less likely to allow the seismic energy to be trapped within these weaker formations due to an expected smaller angle of incidence. This explanation may suggest that the entire waveform appearance, including resonating coda, would generally be expected in this region, and make a stronger argument for referring to them as "hybrid events".

Of course, this was a quickly put together scenario and the Authors have different ideas, but something along these lines or that can similarly explain these observations would really strengthen the paper in terms of demonstrating that the low frequency coda is an intrinsic part of the signal which appears to be the case, and that they can satisfactorily be referred to as hybrid events. It also gives novel insights in terms of interpreting similar hybrid events in other environments (e.g. volcanoes). I do though appreciate that this is not in line with the Authors' interpretations of two shallower low velocity heterogeneities: perhaps they can somehow better resolve why these interpreted heterogeneities only affect the EHWs and not IIEs (something to do with different angle of incidence changing susceptibility for waves to becoming trapped perhaps?). Some modelling such as has already been provided but addressing this conundrum would significantly improve the manuscript.

I also have a few other comments on the manuscript:

Other major comment:

There seems to be a pretty constant seismicity rate for IIEs and EHWs in the region in Fig 2A (average of around 2-6 EHWs/20-day period?), with superimposed peak around injection (i.e. 8 EHWs/20-day period). In my opinion, this really calls into question some of the interpretation around the spatiotemporal migration of events (as much as I like the interpretation!). In particular, events in the MG08 and southern clusters appear to be ongoing throughout, with no clear noticeable uptick (according to the evidence presented) related to injection or flowback at the wells shown here. The small observed injection-related uptick in EHWs could be explained by the three proximal events. The Authors do not present evidence that the MG08 and southern cluster are not related to other operations, and furthermore, recent studies have shown that HF-induced seismicity can continue for many months after injection, including in the BC Montney (e.g. Salvage & Eaton, 2021; Eyre et al., 2020). It is therefore difficult to determine whether the migration patterns suggested in Fig 2D and the EHW Migration section are reasonable. A plot of Fig 2D but with prior seismicity included would be useful; in particular I'm not convinced by the $D=1.5$ curve through 3 points. The $D = 0.2$ curve looks perhaps more convincing, but now that the authors are including more interpretation on the migration pattern (e.g. Figure 5), it is important to see whether this is clearly a migration pattern or could be related to the background seismicity that appears to be ongoing in these clusters.

To summarise, the Authors have done a good job of ruling out pore pressure and poroelastic stress changes as triggering mechanisms for the distant clusters and thus conclude the only mechanism can be aseismic slip. I think that this interpretation is reasonable, however, they also need to rule out the hypothesis that these events would have occurred anyway due to increased background seismicity rate in these locations

caused by other past injections. In my opinion, this is a current weakness, but if the Authors incorporate changes relating to my main comment then in my opinion this section becomes less critical to the main article anyway, as I still think that the manuscript would be very strong as an observation and interpretation of the first hybrid events relating to induced seismicity, without needing to interpret too much around the event migrations when there are so few events that are difficult to reliably link with the specific injection wells/schedule. The observations of the further clusters are still important and very interesting, but perhaps less interpretation should be made in terms of their relation to the "current" injection well if it is possible that the events are more likely related to other "recent" injection wells.

Minor comments:

Reply to Reviewers Comment 1a: I'm not following the Author's argument here. Magnitude is related to energy. Slower = energy released over longer time = lower amplitude needed for same magnitude? This seems to be opposite of their argument.

L19: located a few kilometers away from the wellbore -> located within a few kilometers of the wellbore
L22-23: "Here we report a new type of signal consisting of an impulsive, broadband onset followed by protracted low-frequency ringing." I maybe wasn't explicit enough in my first review, but I think there is still some confusion here especially for a casual reader. This sentence makes the events sound like normal (broadband) events with lower frequency coda. I think in the abstract you need to clearly state that the ringing is a path effect, and the interesting thing about the events in terms of mechanism are the observations in the following sentence. This sentence may infer that the paper is about understanding the coda and linking it to slow slip, which is not the case. However, this part may need rewriting with respect to my main comment.

L546 - "Hypocenter" - reference?

NB Line numbers refer to tracked changes document.

Thomas Eyre

References:

Bean, C., Lokmer, I., and O'Brien, G. (2008), Influence of near-surface volcanic structure on long-period seismic signals and on moment tensor inversions: Simulated examples from Mount Etna, *J. Geophys. Res.*, 113, B08308, doi:10.1029/2007JB005468.

Thomas S. Eyre, Megan Zecevic, Rebecca O. Salvage, David W. Eaton; A Long-Lived Swarm of Hydraulic Fracturing-Induced Seismicity Provides Evidence for Aseismic Slip. *Bulletin of the Seismological Society of America* 2020;; 110 (5): 2205–2215. doi: <https://doi.org/10.1785/0120200107>

Johnson T.L. (1978) Rupture and Particle Velocity During Frictional Sliding. In: Byerlee J.D., Wyss M. (eds) *Rock Friction and Earthquake Prediction. Contributions to Current Research in Geophysics (CCRG)*, vol 6. Birkhäuser, Basel. https://doi.org/10.1007/978-3-0348-7182-2_25

Passelègue, F.X., Almakari, M., Dublanchet, P. et al. Initial effective stress controls the nature of earthquakes. *Nat Commun* 11, 5132 (2020). <https://doi.org/10.1038/s41467-020-18937-0>

Salvage, R. O. and Eaton, D. W.: Unprecedented quiescence in resource development area allows detection of long-lived latent seismicity, *Solid Earth*, 12, 765–783, <https://doi.org/10.5194/se-12-765-2021>, 2021.

Reviewer #4 (Remarks to the Author):

Comments on the manuscript "Fluid-injection induced earthquakes characterized by hybrid-frequency waveforms manifest the transition from aseismic to seismic slip"

Induced earthquakes have been a global concern since last decade or so. Understanding the mechanisms of inducing earthquakes in different circumstances is critical to advance our knowledge in earthquake physics and to mitigate risks of potential induced earthquakes. In this study, the authors conducted a careful analysis on event waveforms and source parameters of induced earthquakes in Montney, BC, western Canada. After examining waveform similarity and source parameters, they found a group of earthquakes which were termed hybrid-frequency events (EHW). They claimed a distinct contrast in source durations (wave pulse width) and thus corner frequencies for ordinary induced earthquakes and the EHWs, which had similar magnitudes. Consequently, the stress drop values or rupture speeds of the EHW events must be smaller than those of ordinary induced earthquakes in the same region. Then they attributed the hybrid-frequency events to possible aseismic slip, a new mechanism that was suggested to induced earthquakes by fracking. To verify such a hypothesis, they conducted numerical modeling to inspect the level of pore pressure change and poroelastic stress perturbation and concluded that the combination of the two mechanisms was too small to trigger the early EHWs in distance, indicating that aseismic slip was likely the cause of additional stress perturbation to trigger earthquakes.

Major findings of this study are to identify a group of earthquakes with apparently lower corner frequencies than ordinary induced earthquakes. This is very much like the analog of low-frequency earthquakes within tectonic tremors VS tectonic earthquakes. The former was recognized as a manifestation of slow slip events that were observed at the transition zone from velocity-weakening and velocity-strengthening.

Overall the technical analysis was rigorously done and the interpretation was sound. I am not one of the original reviewers on this manuscript, but I find that the authors have done a good job in addressing all questions raised previously. Indeed, this version of the manuscript has been substantially improved from the original submission. With the following rather Minor points to be addressed, I recommend acceptance of this manuscript.

Introduction should include the 2019 shallow Rongxian earthquake, which was thought to be the first deadly earthquake induced by fracking. Responsible mechanisms exclude pore pressure diffusion but were still under investigation. Aseismic slip is a plausible one, which however can not be verified due to the lack of near-field monitoring data.

Yang, H., P. Zhou, N. Fang, G. Zhu, W. Xu, J. Su, F. Meng, and R. Chu (2020), A shallow shock: the 25 February 2019 ML 4.9 earthquake in the Weiyuan shale gas field in Sichuan, China, *Seismo. Res. Lett.*, 91(6), p.3182-3194, doi:10.1785/0220200202

Low frequency ringing. It is fine, but I think "elongated low frequency coda" or "low frequency coda" is more common. Indeed, I have to go back to check the figure again and again when reading the term "ringing", because the very first thought of mine was normal mode. The apparent duration T_a was also confusing to me whether it referred to the onset pulse width when reading the text. If it refers to coda waves, then "coda duration" is a concise and clear term.

EHW source mechanism: I think the authors have made tremendous efforts to validate that the low frequency coda waves were generated along ray paths while the onset pulse width (corner frequencies) were originated from the source. Then the logic would be easier to move the two paragraphs (line 192-220) in front of line 164. It is actually important to highlight the lack of dependency of T_a and f_c (Fig. S5), after concluding that the T_a was indeed caused by the heterogeneities along ray path.

Line 315-319: the discussion of potential effects of dilatancy was great. For any given fault, whether the slip is aseismic or not is up to whether the slow slip can accelerate into seismic regime. As such, the statement of "the strengthened fault tends to inhibit seismic slip" is a bit misleading, as dilatancy effects are more profound in low slip rates and the fault is not close to seismic slip rate yet. I think to rephrase it to "the strengthened fault tends to slow down/ inhibit the slip acceleration" was what the author meant here.

Figure 1: I do not think panels C and E are very informative. Of course this is my personal taste, but the low

frequency coda was more obvious on the seismogram (D). An aligned column comparison of these seismograms with the zoom-in plot of P- and S-wave windows should suffice the purpose of highlighting the difference. Indeed, Fig. 1C and 1E were never quoted in the text.

A clear illustration to me would be to show four seismograms, with one regular induced earthquake recorded at two stations, and one EHW at two stations that show variant duration of the low-frequency coda. Then it is self-explanatory that the code is from heterogeneities along paths, not from the source.

I am curious though, if the nearly co-located ordinary induced earthquakes share similar low-frequency coda at any stations. Then I got the answer from Fig. S6. First I suggest to highlight the title of subpanels C, D, E in Figure S6, by possibly using different colors. Second, I think showing a seismogram with coda for the ordinary induced earthquakes in Figure 1 would be very helpful.

Line 327-329: If stress perturbation is insufficient to overcome the stress excess (the difference between frictional strength and shear stress), then the fault is stable. The statement of "any ruptures would be stabilized" seems to contradict with such understanding, because "ruptures" have not been developed yet in such a case. I think the authors mean "the fault is stable" here and "the fault may slip seismically" for the case of having the stress perturbation exceeding the stress excess.

Hongfeng Yang

Reviewer #3 (Remarks to the Author):

Report Summary

In their revised version, the authors made at least three significant changes to their manuscript.

First, they have made it clearer how accurate the collocation between EHW and seismic events was by providing some “representative” examples in Fig. S3. In addition, they invoke that given some differences in shear waves velocities at the respective depths of the “collocated” events, the event magnitudes should give the same estimates. Thus, they conclude this is validating their conclusion that EHW have different source properties compared to seismic events. By providing examples of spectral EHW characteristics, they also make it more clear that they look different from induced earthquakes. On this first point, the authors bring in more convincing data that the EHW source process might be different.

Second big add from the authors is a geomechanical modeling approach to estimate if the poroelastic stress transfer mechanism could explain EHW triggering. They use a coupled hydromechanical elastic solution with COMSOL to demonstrate that, even with an unrealistic high initial shale permeability, they can only explain that the borehole nearfield events may have been created enough Coulomb stress transfer. They, thus, argue that this is a way to strengthen their hypothesis of a slow rupture building additional stress for the EHW that triggered further away. We agree that the poroelastic stress transfer hypothesis may be considered unrealistic, although the COMSOL model which, from our understanding, does not include poroelastic change in the shales permeability may be a little too simplistic. In addition, the model is not clear about how the background stress is applied. The main question here is that this model obviously does not demonstrate the possibility of the slow slip hypothesis since it does not include rupture.

Third, following reviewer’s suggestions, the authors have reinterpreted the EHW mechanisms in the context of three differently located groups. This is allowing them to give more focus on the “southern cluster” where EHW and seismic events apparently coexist on the same structure (given the location accuracy). I would have suggested to the authors to give even more details about this zone. When did the EHW occur after the injection started. It is not clear, but seems to exceed 20 days. Thus after injection end. Fig 2A histogram is adding complexity for the reader in a way that it does not show a clear correlation between injections and EHW or seismic events, as mentioned by the authors. This is quite disturbing given the title proposed for the paper.

To conclude, although the authors made a significant work to improve their paper, it still seems to me that their interpretation of the data is a bit overinterpreted, or at least that we cannot clearly state that EHW events mark a transition from aseismic to seismic slip. Overinterpreted in a way that we may trust that some different source mechanisms are observed, but nowhere there is a direct demonstration that it is related to slow slip. We recognize that this is a hard job to demonstrate (that would request more advanced numerical modeling or a clear correlation between injection and EHW and seismicity); and we acknowledge again the high merit of the authors to bring to the scientific community a nice catalogue of exotic events (they call EHWs) which are well located around a hydrofracking treatment well.

For these reasons, we would definitely recommend this contribution more as a report for the quality of the observations, especially since they have been completed with some crucial information on the HF treatment. I recommend rejection of this manuscript for Nature Communications.

Manuscript Number: NCOMMS-20-41246

Title: Fluid-injection induced earthquakes characterized by hybrid-frequency waveforms manifest the transition from aseismic to seismic slip

Authors: H. Yu, R. M. Harrington, H. Kao, Y. Liu, and B. Wang

We would first like to thank three reviewers for their positive feedback and further comments that are very helpful to improve the quality of the manuscript. All three additional reviewers agree that the major contribution of this study is the identification of a new class of injection-related earthquakes with apparently lower corner frequencies relative to typical induced earthquakes. In the newly revised version, we provide a consistent and convincing interpretation that these earthquakes characterized by hybrid-frequency waveforms (EHW) likely represent the manifestation of slow rupture transitioning from aseismic to seismic slip, based on the observational evidence and the support of other modeling and experimental results.

As suggested by the first and the fourth reviewers, it is important to emphasize phenomena that may occur on a widespread basis. Underscoring such signals may help advance observational, experimental, and modeling studies related to aseismic slip loading. Moreover, the similarity between EHWs and tectonic slow earthquakes will attract attention from a wide range of interdisciplinary groups in the Earth science community, including experimental rock physics, seismology, and geomechanics. Overall, the expected high impact of this study would fit perfectly with the broad scope and readership of Nature Communications.

In the revised manuscript, we made the following four major revisions to address the concerns raised by reviewers:

- ✧ Clarify that the protraction of coda affects both EHWs and typical induced events, and is therefore consistent with path attenuation. In addition, we differentiate the two features of EHW's coda waves as 1) the protracted duration and 2) lower frequency content. The former is a path effect common to EHWs and typical induced earthquakes, while the latter is a manifestation of the longer EHW source duration. (Fig. 1B-E; Lines 22-27, 197-229, 290-292)
- ✧ Clarify that all EHWs are injection-related from a regional perspective to provide context showing it is rational to study their seismogenic origin as a whole. Demonstrate that EHWs following the migration pattern are linked to the specific HF wells in the study. (Fig. 2C; Lines 155-183)

- ✧ Reorganize the section of “EHW Source Mechanism” to improve the logical flow. We first verify that protracted coda duration stems from path effects rather than pressure gradient-driven fluid flow in a crack (Lines 197-229), then provide evidence that EHWs result from slow rupture through analyzing the broadband portion of the waveforms, including the moment-duration scaling (Lines 230-247) and their rupture characteristics (Lines 249-282).

Our point-to-point responses to individual comments are given below in the order of Reviewer #1, Reviewer #3, and Reviewer #4. The original reviewer comments are in black, and our responses in blue. A “track-changes” version is also included for your convenience. Please refer to the PDF document of the revised manuscript with tracked changes for the corresponding line numbers mentioned in this response letter.

REVIEWER COMMENTS

Reviewer #1 (Remarks to the Author):

I have gone through the paper again. Thank you to the Authors, who have done a very thorough job of addressing my and the other Reviewers' comments and gone through a substantial amount of work to improve the manuscript. In my opinion, the manuscript is far stronger now and closer to publication.

The Authors have now much more clearly presented, with stronger evidence, that there are two distinct event families here, and I believe that it is now more apparent that this difference in corner frequency is most likely caused by a difference in the rupture velocity. This in itself is a very interesting finding. The evidence for all of their results and interpretation is much more clearly presented and should be reproducible.

Nevertheless, in my opinion, the manuscript still has some important flaws in terms of the interpretation related to my original comments that in my opinion affects the potential impact of the paper for a broader audience, strong though it already is for the more specific field of induced seismicity. I do now believe that some relatively moderate changes could lead to a much stronger article and interpretation that is definitely worthy of publication in Nat Comm.

The key issues in terms of the interpretation are that it seems to me that there are three key observations that lead to the classification of these unusual events as “EHWs”:

1. Low frequency coda.
2. Onset with lower corner frequency than typical.
3. Co-location of EHWs and typical events.

The Authors therefore identify the events due to their waveform characteristics above, i.e. events are identified due to their broadband onset with low frequency ringing coda. Events that appear like this are classically referred to as “hybrid” especially in volcanic environments, and therefore it seems reasonable to refer to these events similarly. However, the Authors then appear to discredit the importance of the low frequency coda, as they demonstrate that this is caused by propagation effects. I think it is this part of the manuscript that really causes some of the significant issues that I raised in my first review. For example, is it reasonable to identify events using an aspect of their waveform that is later discounted for being unrelated to the mechanism? Also, can we truly categorise these as “hybrid” events if they are principally events with low corner frequency, and the coda that in my opinion is key to allowing them to be referred to as hybrid is very much

dependent on the site-specific propagation path? Is the one line of evidence (lower corner frequency) enough to support a different categorization of these events to typical induced seismicity (also mentioned by Reviewer 3), where some events of lower stress drops have already been observed in other locations (albeit interpreted as due to different causes)?

The coda wave of EHWs have two distinctive features, the first is that they have relatively lower-frequency content, the second is the prolonged duration. The first feature is unique to the coda of EHWs. Overall, the EHWs contain lower-frequency energy relative to typical induced events, including broader P- and S-phases and longer-period coda waves. Both the broader P- and S-pulses and the lower-frequency coda could be the manifestation of the longer source durations of EHWs. The second feature (variation of coda duration) exists for both EHW and typical induced events, and the variation of duration correlates with source-station travel path. The latter feature suggests that if two EHW and typical event types have similar ray paths, the longer coda wave is due to the dispersive effect along the ray path.

To better illustrate the two coda features of EHWs, we modify Fig. 1 by providing a direct comparison between the representative examples of EHW and typical induced event. We also revise the main text to more clearly explain the two features separately (Lines 22-27, 197-229, 290-292).

I should state here that I very much support the Authors interpretations that the low corner frequency is caused by slow rupture and that the coda is principally a propagation effect. The Authors have done a good job of presenting those arguments. However, in my opinion, the Authors are missing a key link in this conundrum that could strengthen their interpretation:

- Why do the low corner frequency events appear to always generate low frequency, ringing coda? If it does always happen, then this gives a much stronger argument for calling them hybrid events.

We appreciate the comment, as it vividly shows that we need to clarify the points outlined in the response above, namely that the coda is protracted for all events with a given travel path, but that the frequency content of the coda is lower for EHWs than for typical induced events.

The fine-scale fluid-pressurized structures would enhance the dispersive effect by disproportionately attenuating higher frequency energy (Lines 206-208). This effect is likely more significant for EHWs than for typical induce events, due to the lower relative amount of high-frequency energy (suggested by lower corner frequency in their source spectra). As a result, the dispersive effect is even more pronounced in codas of EHWs with

source-station ray paths through such structures. The text modifications noted in the previous response and revisions to Fig. 1 aim at making the above point clear.

- Why don't the co-located, typical induced events generate this coda? If it's a path effect, then we should observe it for both? I don't think it is related to the source frequency as typical events should also have significant low frequency energy (i.e. Brune source model).

The revised text (with line numbers indicated in the first response) now clarifies that the co-located, typical induced events do generate prolonged coda (Fig. 1D) as well. The typical induced event coda duration is also consistent with the inferred location of velocity heterogeneities (Fig. S4D). The frequency contents of the typical induced event coda are not as low as those of EHWs though (e.g., Figs. 1D-E) (also clarified in the new text noted in the response to the first comment).

In my opinion, if the Authors could present a stronger scientifically-reasoned interpretation that solves the above conundrum, they could more legitimately suggest that they have observed "hybrid" events, as the current explanation only partly addresses this. This is therefore my key suggestion for the paper. Admittedly, it's a complicated conundrum to solve; however, I have come up with one possible explanation that I can suggest that the Authors could perhaps better develop to suit their interpretations:

Events with low rupture velocity have been hypothesized to occur in regions of low effective stress (Passelegue et al 2020), with low shear wave velocity (Johnson 1978), or with certain rock properties (Kohli and Zoback, 2013; Eyre et al., 2019). So therefore, according to the Authors' interpretation of low rupture velocity, the EHWs probably occur in formations with high pore pressure and/or in weaker (shale?) formations. All of these conditions are likely in unconventional reservoir formations (Montney formation is known to be overpressured in many regions). The overpressure will also increase during fluid injection. Both of these conditions (high pore pressure, weaker) suggest lower seismic velocities. So perhaps the waves are getting trapped within this low-velocity layer where they may be generated, leading to the "resonating" coda, as proposed by Bean et al. (2008) for shallow low-velocity layers and others (citations 24-28). This fits very nicely with the Authors' explanation in lines 216-229. For regional arrays such as these, depth errors for seismic event locations are usually quite high ($\gg 500$ m) in comparison to the thicknesses of the individual geological formations (of the order of ~ 100 m). The "collocated" IIEs could therefore be happening in different formations just above or below these units with different properties/stress conditions and with different raypaths that are less likely to allow the seismic energy to be trapped within these weaker formations due to an expected smaller angle of incidence. This explanation may suggest that the entire waveform appearance, including resonating coda, would generally be expected in this region, and make a stronger argument for referring to them as "hybrid events".

Of course, this was a quickly put-together scenario, and the Authors have different ideas, but something along these lines or that can similarly explain these observations would really strengthen the paper in terms of demonstrating that the low frequency coda is an intrinsic part of the signal which appears to be the case, and that they can satisfactorily be referred to as hybrid events. It also gives novel insights in terms of interpreting similar hybrid events in other environments (e.g. volcanoes). I do though appreciate that this is not in line with the Authors' interpretations of two shallower low velocity heterogeneities: perhaps they can somehow better resolve why these interpreted heterogeneities only affect the EHWs and not IIEs (something to do with different angle of incidence changing susceptibility for waves to becoming trapped perhaps?). Some modelling such as has already been provided but addressing this conundrum would significantly improve the manuscript.

We thank the reviewer for suggesting resonating wave in a low-velocity, overpressurized layer as an alternative source mechanism to understand EHWs, and we do agree that a high-resolution study would be warranted to investigate possible slow-rupture, waveguide generated signals. However, the limited number of EHWs and the insufficient hypocenter depth resolution do not allow us to explicitly confirm or exclude the existence of fluid resonance within shale layers. We would also like to take an Occam's razor approach, as our revised text now makes it clear to the reader, that all events with a given travel path exhibit protracted coda. Invoking additional fluid resonance effects to explain the signal is not necessary (or justified, in light of the of the depth resolution and correspondingly thin shale layers).

In summary, the similar patterns of coda duration in both EHWs and typical induced events (Fig. S4) suggest that coda duration variation is most easily explained by a dispersive effect along the path. Moreover, Fig. S4 shows the consistency of coda duration variations with two velocity heterogeneities. These two regions could consist of fine-scale fluid-pressurized structures that would enhance the dispersive effect. Such structures may be particularly pronounced as injection into a low-permeable shale formation would create localized fracture network, increase the fluid pressure, and thus increase attenuation for rays travelling through such regions.

Based on the simplicity of the above argument that requires no specific assumptions about hypocentral depth, and that all types of events with specific travel paths experience coda elongation, we prefer to interpret the prolonged coda duration as a path effect. To help emphasize the above interpretation, we reorganize the text concerning our analysis of the source mechanisms on Lines 197-229.

I also have a few other comments on the manuscript:

Other major comment:

There seems to be a pretty constant seismicity rate for IIEs and EHWs in the region in Fig 2A (average of around 2-6 EHWs/20-day period?), with superimposed peak around injection (i.e., 8 EHWs/20-day period). In my opinion, this really calls into question some of the interpretation around the spatiotemporal migration of events (as much as I like the interpretation!). In particular, events in the MG08 and southern clusters appear to be ongoing throughout, with no clear noticeable uptick (according to the evidence presented) related to injection or flowback at the wells shown here. The small observed injection-related uptick in EHWs could be explained by the three proximal events. The Authors do not present evidence that the MG08 and southern cluster are not related to other operations, and furthermore, recent studies have shown that HF-induced seismicity can continue for many months after injection, including in the BC Montney (e.g. Salvage & Eaton, 2021; Eyre et al., 2020). It is therefore difficult to determine whether the migration patterns suggested in Fig 2D and the EHW Migration section are reasonable. A plot of Fig 2D but with prior seismicity included would be useful; in particular I'm not convinced by the $D=1.5$ curve through 3 points. The $D = 0.2$ curve looks perhaps more convincing, but now that the authors are including more interpretation on the migration pattern (e.g. Figure 5), it is important to see whether this is clearly a migration pattern or could be related to the background seismicity that appears to be ongoing in these clusters.

To summarise, the Authors have done a good job of ruling out pore pressure and poroelastic stress changes as triggering mechanisms for the distant clusters and thus conclude the only mechanism can be aseismic slip. I think that this interpretation is reasonable, however, they also need to rule out the hypothesis that these events would have occurred anyway due to increased background seismicity rate in these locations caused by other past injections. In my opinion, this is a current weakness, but if the Authors incorporate changes relating to my main comment then in my opinion this section becomes less critical to the main article anyway, as I still think that the manuscript would be very strong as an observation and interpretation of the first hybrid events relating to induced seismicity, without needing to interpret too much around the event migrations when there are so few events that are difficult to reliably link with the specific injection wells/schedule. The observations of the further clusters are still important and very interesting, but perhaps less interpretation should be made in terms of their relation to the “current” injection well if it is possible that the events are more likely related to other “recent” injection wells.

We thank the reviewer for the helpful comment, and we have now removed the original Fig. 2A (Temporal comparison between seismicity and injection operations). We also revise the migration plot (now a new Fig. 2C) to include EHWs that occurred before HF injection and during the well stimulation period.

The new Fig. 2C more clearly shows the temporal relationship between injection and EHWs. Specifically, two EHWs occurred before the well preparation/testing/stimulation (P/T/S), seven EHWs nearly concurrent with the onset of the July HF stimulation but located at ~3 km from the well. The subsequent EHWs migrate outward approximately following a hydraulic diffusivity of ~0.2 m²/s, which is consistent with estimations based on typical induced events in our study area (22).

We agree with the reviewer that the EHWs that occurred prior to injection and during injection on short timescales and at intermediate distances are likely associated with the local critical stress state that may have been primed by previous injection activity in the study area. Specifically, there were four HF operations during 2013 and 2014, including one in September 2013 with three horizontal wellbores extending from the same HF well pad in our study area. From the perspective of injection-related EHWs, it is rational to assume that all detected EHWs have the similar seismogenic origin and to study them as one group. Here we only consider EHWs below the migration curve as being reliably linked to the specific well. We recognize the number of “migrating” EHWs may be insufficient to confidently outline a detailed migration pattern. But, the similarity of migration patterns between EHWs and typical induced events (D=0.2 m²/s) makes it logical to assume that they share common driving mechanisms.

We compile the above argument in the revised text (Lines 155-183).

Minor comments:

1. Reply to Reviewers Comment 1a: I’m not following the Author’s argument here. Magnitude is related to energy. Slower = energy released over longer time = lower amplitude needed for same magnitude? This seems to be opposite of their argument.

In our study, the seismic moment is estimated as $M_0 = \frac{4\pi\rho c^3 R\Omega_0}{U_{\phi\theta}}$ (Eq. 2), where c is the shear velocity, not rupture velocity (although it is usually linked to the rupture velocity by a constant), and Ω_0 is the static dislocation of the ruptured fault which is linearly related to the seismic moment. The value of Ω_0 is not related to the details of rupture process (e.g., rupture velocity, source duration, or f_c).

In our previous response, the smaller c value (2.59 km/s) we applied for the EHW is due to its shallower focal depth (1.93 km). A lower shear wave velocity means a smaller shear modulus, that is, it requires a larger fault dislocation to release the same elastic energy. As $M_0 \propto c^3\Omega_0$, the EHW ruptured in a layer with a slower shear velocity c requires a larger Ω_0 (low-frequency amplitude) to have a similar moment as a typical

induced event.

Directly comparing the amplitudes of EHW and typical induced events in Fig. 1 may be a bit misleading, as their focal depths are slightly different. Nevertheless, the point of Fig. 1B-E is to show the differences in frequency content between the two types of earthquakes. We add two sentences in the caption to emphasize the last point.

2. L19: located a few kilometers away from the wellbore -> located within a few kilometers of the wellbore.

The sentence is revised accordingly in Lines 18-20 as “Aseismic slip loading has recently been proposed as a complementary mechanism to induce moderate-sized earthquakes located within a few kilometers of the wellbore over the timescales of hydraulic stimulation”.

3. L22-23: “Here we report a new type of signal consisting of an impulsive, broadband onset followed by protracted low-frequency ringing.” I maybe wasn’t explicit enough in my first review, but I think there is still some confusion here especially for a casual reader. This sentence makes the events sound like normal (broadband) events with lower frequency coda. I think in the abstract you need to clearly state that the ringing is a path effect, and the interesting thing about the events in terms of mechanism are the observations in the following sentence. This sentence may infer that the paper is about understanding the coda and linking it to slow slip, which is not the case. However, this part may need rewriting with respect to my main comment.

Sentences are now revised in Lines 22-27 as “Here we report a new type of earthquakes characterized by hybrid-frequency waveforms (EHWs). Distinguishing features from typical induced earthquakes include broader P and S-pulses and relatively lower-frequency coda content. Both features may be causally related to lower corner frequencies, implying longer source durations, thus, either slower rupture speeds, lower stress drop values, or a combination of both”.

4. L546 – “Hypocenter” – reference?

Thanks for the reminder. We add it in Line 437.

NB Line numbers refer to tracked changes document.

Thomas Eyre

Reviewer #3

Report Summary

In their revised version, the authors made at least three significant changes to their manuscript. First, they have made it clearer how accurate the collocation between EHW and seismic events was by providing some “representative” examples in Fig. S3. In addition, they invoke that given some differences in shear waves velocities at the respective depths of the “collocated” events, the event magnitudes should give the same estimates. Thus, they conclude this is validating their conclusion that EHW have different source properties compared to seismic events. By providing examples of spectral EHW characteristics, they also make it clearer that they look different from induced earthquakes. On this first point, the authors bring in more convincing data that the EHW source process might be different.

Although the reviewer did not raise any issue here, he/she may have misunderstood our reasoning (as highlighted above), we want to point out that the similar magnitude estimates of EHWs and typical induced events are not one of the reasons we consider EHWs as a new type of seismic signals. Instead, EHWs differ from the co-located typical induced events in the following aspects, as (1) their waveforms having broader P and S-pulses and protracted low-frequency coda waves (Figs. 1B-E), (2) their longer source durations (Fig. 3), equivalent to either slower rupture speeds, lower stress drop values, or a combination of both (Fig. 4). Moreover, we consider the two aspects are causally linked.

We choose the two events with similar magnitudes in Fig. 1 with the intention to eliminate any doubt that the broader P- and S-pulses and lower-frequency coda of EHWs are resulted from magnitude differences.

Second big add from the authors is a geomechanical modeling approach to estimate if the poroelastic stress transfer mechanism could explain EHW triggering. They use a coupled hydromechanical elastic solution with COMSOL to demonstrate that, even with an unrealistic high initial shale permeability, they can only explain that the borehole nearfield events may have been created enough Coulomb stress transfer. They, thus, argue that this is a way to strengthen their hypothesis of a slow rupture building additional stress for the EHW that triggered further away. We agree that the poroelastic stress transfer hypothesis may be considered unrealistic, although the COMSOL model which, from our understanding, does not include poroelastic change in the shale’s permeability may be a little too simplistic. In addition, the model is not clear about how the background stress is applied. The main question here is that this model obviously does not demonstrate the possibility of the slow slip hypothesis since it does not include rupture.

The aim of the COMSOL modeling is not to simulate the slow slip or earthquake rupture process, but rather to show that the timescale and magnitude of pore pressure and poroelastic changes are insufficient to explain the timing of EHWs at the distances of several kilometers from the well. Since the stress changes do not depend on the initial stress state, we assume that (1) the initial normal stress follows lithostatic gradient, (2) pore pressure follows hydrostatic gradient, and (3) shear stress is the product of friction coefficient times and normal stress (added in Lines 595-598). The model then calculates stress perturbations from the initial state due to fluid injection. Typically, induced seismicity studies invoke poroelastic stress transfer to explain seismicity more than kilometers away from the wellbore in low permeable shales where the diffusion rate is slow (e.g., Goebel et al., 2017; Peña Castro and Roth, 2020; Deng et al., 2020; Wang et al., 2021). The COMSOL modeling result demonstrates that the coupled effect of *both* pore pressure and poroelastic stress changes inferred from injection data in this study is insufficient to induce the “early” EHWs at greater distances, even with an abnormally high value of assumed permeability. Therefore, the model suggests that aseismic slip loading could be the only viable candidate capable of inducing slip, as it can effectively facilitate the pressure front propagation needed to nucleate failure. We emphasize that direct simulation of aseismic slip and/or seismic rupture is beyond the scope of our COMSOL modeling. In the manuscript we clearly state that the objective is to test the null hypothesis that classical mechanisms of pore pressure and poroelastic stress change are enough to explain the observations.

In our COMSOL modeling, we ignore the feedback effect on permeability from the coupled pore pressure and poroelastic stress. We agree that it may be a simplified scheme, but argue that dismissing the permeability change in the numerical simulations of the poroelastic coupling is very common for induced seismicity modeling (e.g., Segall and Lu 2015; Chang and Segall, 2015; Deng et al., 2016; Zhai et al., 2019; Peña Castro and Roth, 2020; Deng et al., 2020; Wang et al., 2021). The main reason is that the permeability change related to poroelastic effects decreases rapidly with increasing distance (Freeman et al., 2008; Cappa, 2009). In our case, the clusters we focus are ~2 km away from the well (Fig. 2C), so it is reasonable to dismiss the permeability change. We accordingly added a sentence in Lines 602-604 as “We note that applying a strain-independent permeability may have minor effects on predicting the stress perturbations in the proximity of the well (79)”.

Aseismic slip has only been observed in induced earthquake studies under laboratory or meso-scale experimental conditions (e.g., Guglielmi et al., 2015; Wu and McClaskey, 2018), and it would be extremely challenging to observe in situ at seismogenic depths. In spite of the fact that we do not observe aseismic slip directly, our observations of slower

rupture velocity, the spatial/temporal distribution of the EHWs, and the COMSOL modeling are all consistent with the inference of aseismic slip loading. While running an aseismic slip model under the framework of rate and state dependent frictional law to address the role of aseismic slip loading is not the aim, it has been shown as a viable mechanism (as evidenced by many recent publications, including Eyre et al., 2019; Bhattacharya and Viesca, 2019). Moreover, according to Yang and Dunham (2020), the pressure front of aseismic slip loading could propagate at twice the rate of fluid diffusion in our case, which is consistent with the occurrence of the “early” EHWs (Fig. 5D) (Lines 349-350). In short, our EHW observations, the COMSOL modeling, and the aseismic rupture modeling works by previous studies are all internally consistent in supporting our interpretation. Conducting additional aseismic rupture modeling would step into the theoretical aspect of the physical mechanism – a topic outside of the observational theme of this paper.

Third, following reviewer’s suggestions, the authors have reinterpreted the EHW mechanisms in the context of three differently located groups. This is allowing them to give more focus on the “southern cluster” where EHW and seismic events apparently coexist on the same structure (given the location accuracy). I would have suggested to the authors to give even more details about this zone. When did the EHW occur after the injection started? It is not clear but seems to exceed 20 days. Thus, after injection end. Fig 2A histogram is adding complexity for the reader in a way that it does not show a clear correlation between injections and EHW or seismic events, as mentioned by the authors. This is quite disturbing given the title proposed for the paper.

Thanks for the suggestions. As detailed in the last major comment of Reviewer 1, we remove Fig. 2A and revise the migration plot (now Fig. 2C). The new Fig. 2C includes EHWs that occurred prior to HF injection and information related to well operations. Please refer to the details of the referenced response above.

To conclude, although the authors made a significant work to improve their paper, it still seems to me that their interpretation of the data is a bit overinterpreted, or at least that we cannot clearly state that EHW events mark a transition from aseismic to seismic slip. Overinterpreted in a way that we may trust that some different source mechanisms are observed, but nowhere there is a direct demonstration that it is related to slow slip.

We recognize that this is a hard job to demonstrate (that would request more advanced numerical modeling or a clear correlation between injection and EHW and seismicity); and we acknowledge again the high merit of the authors to bring to the scientific community a nice catalogue of exotic events (they call EHWs) which are well located around a hydrofracking treatment well.

For these reasons, we would definitely recommend this contribution more as a report for the quality of the observations, especially since they have been completed with some crucial information on the HF treatment. I recommend rejection of this manuscript for Nature Communications.

All three additional reviewers unanimously agree that the revised version of the manuscript communicates the major contribution of this study, namely, it identifies a new class of injection-related earthquakes with apparently lower corner frequencies than typical induced earthquakes. The relation between EHWs and typical induced events is analogous to that between slow earthquakes and natural events in a tectonic environment. Both EHWs and tectonic slow earthquakes are most likely associated with slow slip typically observed in the frictional transition zone between velocity-weakening and velocity-strengthening regimes. As such, they are of broad interest to a wide range of interdisciplinary groups in the geoscience community, including experimental rock physics, seismology, and geomechanics.

We agree that direct evidence of aseismic slip is difficult to provide. It may require high-resolution in-situ monitoring of deformation at the injection depth and/or surface geodetic measurements, which is beyond the scope of this study. However, the observational evidence presented here supports the inference of aseismic slip loading as a driving factor: unique waveform characteristics (Fig. 1), evidently longer source durations (Figs. 3-4), the spatiotemporal correlation between typical induced earthquakes, EHWs, and injection activity, insufficient stress perturbation without considering the role of aseismic slip loading (Fig. 5), as well as the similarity between EHWs and tectonic slow earthquakes (Figs. 3-4). The observational evidence presented here, along with the support of other modeling and experimental results, collectively forms a consistent and convincing basis for the interpretation that EHWs likely represent the manifestation of slow rupture transitioning from aseismic to seismic regimes.

One additional aspect to emphasize is that it is scientifically very important to document phenomena that may be previously overlooked but become observable as a result of recent advances in the monitoring capability. EHWs generally have low amplitude signals that cannot be easily detected and recognized without dense seismic arrays. If injection commonly first induces aseismic slip near the wellbore, EHWs should occur on a widespread basis, and are likely not limited to occurring in the Montney shale formation. With the proliferation of enhanced seismic and geodetic observations at injection sites, we expect an explosion of studies in the near future to further advance our understanding of the aseismic-seismic transition and its physical implications. It is precisely for the purpose of making the biggest possible impact that we feel this pioneer research should be published in Nature Communications.

Reviewer #4 (Remarks to the Author):

Comments on the manuscript “Fluid-injection induced earthquakes characterized by hybrid-frequency waveforms manifest the transition from aseismic to seismic slip”

Induced earthquakes have been a global concern since last decade or so. Understanding the mechanisms of inducing earthquakes in different circumstances is critical to advance our knowledge in earthquake physics and to mitigate risks of potential induced earthquakes. In this study, the authors conducted a careful analysis on event waveforms and source parameters of induced earthquakes in Montney, BC, western Canada. After examining waveform similarity and source parameters, they found a group of earthquakes which were termed hybrid-frequency events (EHW). They claimed a distinct contrast in source durations (wave pulse width) and thus corner frequencies for ordinary induced earthquakes and the EHWs, which had similar magnitudes. Consequently, the stress drop values or rupture speeds of the EHW events must be smaller than those of ordinary induced earthquakes in the same region. Then they attributed the hybrid-frequency events to possible aseismic slip, a new mechanism that was suggested to induced earthquakes by fracking. To verify such a hypothesis, they conducted numerical modeling to inspect the level of pore pressure change and poroelastic stress perturbation and concluded that the combination of the two mechanisms was too small to trigger the early EHWs in distance, indicating that aseismic slip was likely the cause of additional stress perturbation to trigger earthquakes.

Major findings of this study are to identify a group of earthquakes with apparently lower corner frequencies than ordinary induced earthquakes. This is very much like the analog of low-frequency earthquakes within tectonic tremors VS tectonic earthquakes. The former was recognized as a manifestation of slow slip events that were observed at the transition zone from velocity-weakening and velocity-strengthening.

Overall, the technical analysis was rigorously done, and the interpretation was sound. I am not one of the original reviewers on this manuscript, but I find that the authors have done a good job in addressing all questions raised previously. Indeed, this version of the manuscript has been substantially improved from the original submission. With the following rather Minor points to be addressed, I recommend acceptance of this manuscript.

1. Introduction should include the 2019 shallow Rongxian earthquake, which was thought to be the first deadly earthquake induced by fracking. Responsible mechanisms exclude pore pressure diffusion but were still under investigation. Aseismic slip is a plausible one, which however cannot be verified due to the lack of near-field monitoring data.

Yang, H., P. Zhou, N. Fang, G. Zhu, W. Xu, J. Su, F. Meng, and R. Chu (2020), A shallow shock: the 25 February 2019 ML 4.9 earthquake in the Weiyuan shale gas field in Sichuan, China, *Seismo. Res. Lett.*, 91(6), p.3182-3194, doi:10.1785/0220200202.

We add the reference accordingly in Lines 45-46 and Line 49.

2. Low frequency ringing. It is fine, but I think “elongated low frequency coda” or “low frequency coda” is more common. Indeed, I have to go back to check the figure again and again when reading the term “ringing”, because the very first thought of mine was normal mode. The apparent duration T_a was also confusing to me whether it referred to the onset pulse width when reading the text. If it refers to coda waves, then “coda duration” is a concise and clear term.

Thanks for the suggestion. We replace “low-frequency ringing” with “low-frequency coda”, and “apparent ringing duration T_a ” with “coda duration T_a ”.

3. EHW source mechanism: I think the authors have made tremendous efforts to validate that the low frequency coda waves were generated along ray paths while the onset pulse width (corner frequencies) were originated from the source. Then the logic would be easier to move the two paragraphs (line 192-220) in front of line 164. It is actually important to highlight the lack of dependency of T_a and f_c (Fig. S5), after concluding that the T_a was indeed caused by the heterogeneities along ray path.

We reorganize the section of “EHW Source Mechanism” as the reviewer suggested. We first verify that the protracted duration of the coda wave is likely the result of dispersion rather than pressure gradient-driven fluid flow in a crack (Lines 197-229), then analyze the scaling between seismic moment and source duration based on the broadband portion of the waveforms (Lines 230-247).

4. Line 315-319: the discussion of potential effects of dilatancy was great. For any given fault, whether the slip is aseismic or not is up to whether the slow slip can accelerate into seismic regime. As such, the statement of “the strengthened fault tends to inhibit seismic slip” is a bit misleading, as dilatancy effects are more profound in low slip rates and the fault is not close to seismic slip rate yet. I think to rephrase it to “the strengthened fault tends to slow down/ inhibit the slip acceleration” was what the author meant here.

Thanks for the clarification. We rephrase the sentence in Lines 346-348 as “Consequently, the strengthened fault tends to hinder, or even inhibit slip acceleration that, in turn, creates a mechanical condition in favor of aseismic/slow slip”.

5. Figure 1: I do not think panels C and E are very informative. Of course, this is my personal taste, but the low frequency coda was more obvious on the seismogram (D). An aligned column comparison of these seismograms with the zoom-in plot of P- and S-wave windows should suffice the purpose of highlighting the difference. Indeed, Fig. 1C and 1E were never quoted in the text.

A clear illustration to me would be to show four seismograms, with one regular induced earthquake recorded at two stations, and one EHW at two stations that show variant duration of the low-frequency coda. Then it is self-explanatory that the code is from heterogeneities along paths, not from the source.

I am curious though, if the nearly co-located ordinary induced earthquakes share similar low-frequency coda at any stations. Then I got the answer from Fig. S6. First, I suggest highlighting the title of subpanels C, D, E in Figure S6, by possibly using different colors. Second, I think showing a seismogram with coda for the ordinary induced earthquakes in Figure 1 would be very helpful.

Following the reviewer's comment, we remove the original Fig. 1C and 1E and replace them with the waveforms of two representative events recorded at station MG02. Both events show longer coda durations at a greater epicentral distance. We also note in the caption that the coda waves of the EHW contain a greater proportion of lower-frequency content.

We also adjust Fig. S4 (previous Fig. S6) accordingly.

6. Line 327-329: If stress perturbation is insufficient to overcome the stress excess (the difference between frictional strength and shear stress), then the fault is stable. The statement of "any ruptures would be stabilized" seems to contradict with such understanding, because "ruptures" have not been developed yet in such a case. I think the authors mean "the fault is stable" here and "the fault may slip seismically" for the case of having the stress perturbation exceeding the stress excess.

We revise the sentence in Lines 356-359 as "The fault is stable in the early stage when the stress perturbation is insufficient to overcome the residual fault strength. As the stress perturbation further loads the fault to overcome the residual fault strength, the fault would eventually slip seismically (20, 56)".

Hongfeng Yang

References

- Bean, C., Lokmer, I., and O'Brien, G. (2008), Influence of near-surface volcanic structure on long-period seismic signals and on moment tensor inversions: Simulated examples from Mount Etna, *J. Geophys. Res.*, 113, B08308, doi:10.1029/2007JB005468.
- Cappa, F. (2009), Modelling fluid transfer and slip in a fault zone when integrating heterogeneous hydromechanical characteristics in its internal structure, *Geophysical Journal International*, 178(3), 1357-1362.
- Chang, K. W., and P. Segall (2016), Injection-induced seismicity on basement faults including poroelastic stressing, *Journal of Geophysical Research: Solid Earth*, 121(4), 2708-2726, doi:<https://doi.org/10.1002/2015JB012561>.
- Deng, K., Y. Liu, and X. Chen (2020), Correlation Between Poroelastic Stress Perturbation and Multidisposal Wells Induced Earthquake Sequence in Cushing, Oklahoma, *Geophysical Research Letters*, 47(20), e2020GL089366.
- Deng, K., Y. Liu, and R. M. Harrington (2016), Poroelastic stress triggering of the December 2013 Crooked Lake, Alberta, induced seismicity sequence, *Geophysical Research Letters*, 43(16), 8482-8491, doi:10.1002/2016gl070421.
- Freeman, T. T., R. J. Chalaturnyk, and I. I. Bogdanov (2008), Fully coupled thermo-hydro-mechanical modeling by COMSOL Multiphysics, with applications in reservoir geomechanical characterization, paper presented at COMSOL Conf.
- Goebel T. H. W., Weingarten M., Chen X., Haffener J., Brodsky E. E. (2017). The 2016 MW5.1 Fairview, Oklahoma earthquakes: Evidence for long-range poroelastic triggering at >40 km from fluid disposal wells. *Earth Planet. Sci. Lett.*, 472, 50-61.
- Johnson T.L. (1978) Rupture and Particle Velocity During Frictional Sliding. In: Byerlee J.D., Wyss M. (eds) *Rock Friction and Earthquake Prediction. Contributions to Current Research in Geophysics (CCRG)*, vol 6. Birkhäuser, Basel. https://doi.org/10.1007/978-3-0348-7182-2_25
- Passelègue, F.X., Almakari, M., Dublanchet, P. et al. (2020). Initial effective stress controls the nature of earthquakes. *Nat. Commun.*, 11, 5132 <https://doi.org/10.1038/s41467-020-18937-0>
- Peña Castro, A. F., M. P. Roth, A. Verdecchia, J. Onwuemeka, Y. Liu, R. M. Harrington, Y. Zhang, and H. Kao (2020), Stress Chatter via Fluid Flow and Fault Slip in a Hydraulic Fracturing - Induced Earthquake Sequence in the Montney Formation, British Columbia, *Geophysical Research Letters*, 47(14), e2020GL087254, doi:10.1029/2020gl087254.
- Salvage, R. O. and Eaton, D. W. (2021) Unprecedented quiescence in resource development area allows detection of long-lived latent seismicity, *Solid Earth*, 12, 765–783, <https://doi.org/10.5194/se-12-765-2021>.
- Segall, P., and S. Lu (2015), Injection-induced seismicity: Poroelastic and earthquake nucleation effects, *Journal of Geophysical Research: Solid Earth*, 120(7), 5082-5103, doi: <https://doi.org/10.1002/2015JB012060>.
- Thomas S. Eyre, Megan Zecevic, Rebecca O. Salvage, David W. Eaton (2020). A Long-Lived Swarm of Hydraulic Fracturing-Induced Seismicity Provides Evidence for Aseismic Slip. *Bulletin of the Seismological Society of America*; 110 (5): 2205–2215. doi: <https://doi.org/10.1785/0120200107>

Wang, B., A. Verdecchia, H. Kao, R. M. Harrington, Y. Liu, and H. Yu (2021), A Study on the Largest Hydraulic Fracturing Induced Earthquake in Canada: Numerical Modeling and Triggering Mechanism, *Bulletin of the Seismological Society of America*, doi:10.1785/0120200251.

Zhai, G., M. Shirzaei, M. Manga, and X. Chen (2019), Pore-pressure diffusion, enhanced by poroelastic stresses, controls induced seismicity in Oklahoma, *Proceedings of the National Academy of Sciences*, 116(33), 16228-16233, doi:10.1073/pnas.1819225116.

REVIEWER COMMENTS

Reviewer #1 (Remarks to the Author):

I reiterate my previous review comments that this manuscript introduces very interesting observations of events with unusual characteristics, that are now well presented and should be worthy of publication in Nature Communications. The Authors have now addressed my main issue with the manuscript. More specifically, they have now made it much clearer that the longer coda durations also affect the typical induced events, which was not discussed in the previous versions of the manuscript, and thus their interpretation is now much more reasonable to me. This has greatly improved the strength of the manuscript.

However, I still have major issues with the event migration analysis, which in my opinion was not adequately addressed from the previous review round from comments from myself and other Reviewers. Fixing these may constitute moderate revisions.

I am still concerned with the Authors' interpretation, given the fact that events appear to occur before the injection starts, especially in the southern cluster. Additionally, from my experience it seems unusual to have events in three distinct clusters, with two mostly focused > 2km from the hydraulic fracturing wells, which is further from the injection wells than we typically see with dense monitoring arrays. Given the Authors' revised comments on seismicity prior to the injection in the manuscript (i.e. that there were operations on the same pad in 2013 that may have caused the latent seismicity that was still occurring prior to injection), I did a quick search of the well database using Geoscout (similarly to the Authors). There are actually two other wellpads within this region that were active more recently, in January and July 2014 (Figure R1, see supplementary review pdf) – the Authors allude to this in their rebuttal. The locations of these wellpads seem to correlate well with the locations of the southern and MG08 clusters of EHWs. In my opinion, it could be possible that the EHW activity in these clusters could be, at least in part, related to past activity at these two wellpads, rather than entirely the "current" wellpad, especially as the Authors suggest that it could be related to operations at the "current" wellpad even earlier, in 2013. Latent seismicity from multiple operations has been shown to occur in the Montney in other areas (Salvage & Eaton, 2021) and the Authors acknowledge this. However, I disagree with the comment in their rebuttal "EHWs below the migration curve as being reliably linked to the specific well", as they may be a continuation of either latent seismicity or the earlier triggered events in those clusters.

Just because the events fit below a curve does not infer causality. The Authors' fit three of five events (MG08 cluster), 7 of 13 events (southern cluster), and 2 of four random events using the fluid diffusion curve. This is only just over half of the events that occurred, suggesting little statistical significance to me, and really calling into question the fit and subsequent interpretations, especially in terms of the migration being able to be fitted with a diffusion curve when seismicity is already occurring in that location prior to injection (southern cluster).

Nevertheless, event counts may well increase in these two clusters due to the "current" operations (the evidence appears to show this), as similar observations of interactions between wellpads spaced kilometers apart have been made in other areas, e.g. the Duvernay (Bao and Eaton, Science, 2016; e.g. Fig. 3). To my knowledge, the causality of such behaviour is still yet to be well explained, and is difficult to explain over the distances of kms seen (the Authors have shown this with their modelling work). However, it is likely related to the previous injections bringing the faults closer to a critically stressed state in the immediate vicinity of the older operations, such that only a very small stress perturbation could lead to a seismic response in these locations. It may also help to better explain why there are three separate clusters of events with "aseismic" regions between them. Neglecting to discuss these other wellpads and operations with respect to the current seismicity is a major oversight of the paper, and may significantly affect the Authors' and readers' interpretations. Following the above consideration, is aseismic slip still the best explanation for the triggering of distances of ~3km? It's possible, although that would require a large slow slip event and therefore I'm not necessarily convinced, but the Authors should at least acknowledge other possibilities related to these operations at other wells: e.g. much smaller poroelastic stress changes required for regions

of past operations; dynamic triggering that has been suggested to occur for some past operations (Wang et al., 2019), etc. Lower stresses need to induce reactivation were also discussed by Wang et al. (2019): "The occurrence of both direct and delayed triggering following transient stress perturbations of <10kPa in all three regions suggests that local faults may remain critically stressed over periods similar to the time frame of our study (~2 yrs) or longer, potentially due to high pore pressures maintained in tight shale formations following injection."

So to conclude, in my opinion this part of the manuscript is currently a bit misleading, despite the fact that the Authors do reject the pore pressure diffusion interpretation in the end. That is not to say that the diffusion curve fitting is obsolete, as in many ways it can be useful to aid the interpretation, but that the Authors' need to tone down their interpretations and consider other possibilities that are not included here, especially the influence of latent seismicity from previous operations, and take into account some of the points outlined above. The Authors need to include the locations (in one of the figures) and timings of these previous operations in the manuscript. It appears to me that any triggering of the MG08 and southern clusters is not at all related to fluid diffusion, as any triggering of reactivation additional to the ongoing latent seismicity likely takes place either during or very shortly after injection, at timescales much faster than diffusion processes, and there may be other possible mechanisms not currently discussed by the Authors.

Minor comments (line numbers from tracked changes document):

L81: British Colombia -> British Columbia

L295: along -> within

Thomas Eyre

References:

Bao, X., Eaton, D.W., 2016. Fault activation by hydraulic fracturing in western Canada. *Science*, 354(6318), 1406-1409.

Bei Wang, Rebecca M. Harrington, Yajing Liu, Honn Kao, Hongyu Yu; Remote Dynamic Triggering of Earthquakes in Three Unconventional Canadian Hydrocarbon Regions Based on a Multiple-Station Matched-Filter Approach. *Bulletin of the Seismological Society of America* 2018;; 109 (1): 372–386. doi: <https://doi.org/10.1785/0120180164>.

Salvage, R. O. and Eaton, D. W.: Unprecedented quiescence in resource development area allows detection of long-lived latent seismicity, *Solid Earth*, 12, 765–783, <https://doi.org/10.5194/se-12-765-2021>, 2021.

Fig. R1. (See uploaded pdf) Wellpads active within 2014 and 2015 in the region. They appear to correlate well with the EHW cluster locations taken from the manuscript (B) (note figure parts not accurately scaled), showing similar clustering to that seen around other operations in other regions (e.g. Bao and Eaton, 2016).

Reviewer #4 (Remarks to the Author):

I appreciate the efforts from the authors, who had done a great job in address my and other reviewers' comments on the previous version. Pleased to find that the authors have addressed all my previous concerns and incorporated my suggestions appropriately. This version can be accepted, with one following minor point that can be easily addressed.

After attributing the coda to the travel path effects, the author interpreted possible heterogeneities along the path due to injection into the shale formation. In Fig. S4, the inferred structural heterogeneities should have their legend, as the color-coded ray path makes the figure pretty busy. A possibly better plot is to show the variation of coda versus back azimuth, which may strengthen the interpretation of heterogeneities along the ray path, in addition to the plot of distance cross section (Fig. S4BCD).

Manuscript Number: NCOMMS-20-41246

Title: Fluid-injection induced earthquakes characterized by hybrid-frequency waveforms manifest the transition from aseismic to seismic slip

Authors: H. Yu, R. M. Harrington, H. Kao, Y. Liu, and B. Wang

We would first like to thank both reviewers for their positive feedback and further comments that are very helpful to improve the quality of the manuscript.

To address their concerns, the revised manuscript mainly includes the following two main revisions:

- ✧ We now clarify the relation between past injection and EHWs that occurred before new injection activity (hence latent EHWs) in the local area by (a) adding the well information of previous injection activity (Fig. 2B), (b) introducing the spatial correlation between EHW clusters and previous injection activity (Lines 140-146), (c) discussing the possible mechanism of latent EHWs (Lines 335-344), and (d) proposing the possibility of EHWs that lie far below the migration curve being latent EHWs (344-346)
- ✧ Add subplots to show the coda duration variation versus back azimuth in Fig. S4.

Our point-to-point responses to individual comments are given below in the order of Reviewer #1 and Reviewer #4. The original reviewer comments are in black, and our responses in blue. A “track-changes” version is also included for your convenience. Please refer to the PDF document of the revised manuscript with tracked changes for the corresponding line numbers mentioned in this response letter.

REVIEWER COMMENTS

Reviewer #1 (Remarks to the Author):

I reiterate my previous review comments that this manuscript introduces very interesting observations of events with unusual characteristics, that are now well presented and should be worthy of publication in Nature Communications. The Authors have now addressed my main issue with the manuscript. More specifically, they have now made it much clearer that the longer coda durations also affect the typical induced events, which was not discussed in the previous versions of the manuscript, and thus their interpretation is now much more reasonable to me. This has greatly improved the strength of the manuscript.

However, I still have major issues with the event migration analysis, which in my opinion was not adequately addressed from the previous review round from comments from myself and other Reviewers. Fixing these may constitute moderate revisions.

I am still concerned with the Authors' interpretation, given the fact that events appear to occur before the injection starts, especially in the southern cluster. Additionally, from my experience it seems unusual to have events in three distinct clusters, with two mostly focused > 2 km from the hydraulic fracturing wells, which is further from the injection wells than we typically see with dense monitoring arrays. Given the Authors' revised comments on seismicity prior to the injection in the manuscript (i.e., that there were operations on the same pad in 2013 that may have caused the latent seismicity that was still occurring prior to injection), I did a quick search of the well database using Geoscout (similarly to the Authors). There are actually two other wellpads within this region that were active more recently, in January and July 2014 (Figure R1, see supplementary review pdf) – the Authors allude to this in their rebuttal. The locations of these wellpads seem to correlate well with the locations of the southern and MG08 clusters of EHWs. In my opinion, it could be possible that the EHW activity in these clusters could be, at least in part, related to past activity at these two wellpads, rather than entirely the “current” wellpad, especially as the Authors suggest that it could be related to operations at the “current” wellpad even earlier, in 2013. Latent seismicity from multiple operations has been shown to occur in the Montney in other areas (Salvage & Eaton, 2021) and the Authors acknowledge this. However, I disagree with the comment in their rebuttal “EHWs below the migration curve as being reliably linked to the specific well”, as they may be a continuation of either latent seismicity or the earlier triggered events in those clusters.

Just because the events fit below a curve does not infer causality. The Authors' fit three of five events (MG08 cluster), 7 of 13 events (southern cluster), and 2 of four random events using the fluid diffusion curve. This is only just over half of the events that occurred, suggesting little statistical significance to me, and really calling into question the fit and subsequent interpretations, especially in terms of the migration being able to be fitted with

a diffusion curve when seismicity is already occurring in that location prior to injection (southern cluster).

Nevertheless, event counts may well increase in these two clusters due to the “current” operations (the evidence appears to show this), as similar observations of interactions between wellpads spaced kilometers apart have been made in other areas, e.g. the Duvernay (Bao and Eaton, Science, 2016; e.g. Fig. 3). To my knowledge, the causality of such behaviour is still yet to be well explained, and is difficult to explain over the distances of kms seen (the Authors have shown this with their modelling work). However, it is likely related to the previous injections bringing the faults closer to a critically stressed state in the immediate vicinity of the older operations, such that only a very small stress perturbation could lead to a seismic response in these locations. It may also help to better explain why there are three separate clusters of events with “aseismic” regions between them. Neglecting to discuss these other wellpads and operations with respect to the current seismicity is a major oversight of the paper, and may significantly affect the Authors’ and readers’ interpretations. Following the above consideration, is aseismic slip still the best explanation for the triggering of distances of ~3km? It’s possible, although that would require a large slow slip event and therefore I’m not necessarily convinced, but the Authors should at least acknowledge other possibilities related to these operations at other wells: e.g. much smaller poroelastic stress changes required for regions of past operations; dynamic triggering that has been suggested to occur for some past operations (Wang et al., 2019), etc. Lower stresses need to induce reactivation were also discussed by Wang et al. (2019): “The occurrence of both direct and delayed triggering following transient stress perturbations of <10 kPa in all three regions suggests that local faults may remain critically stressed over periods similar to the time frame of our study (~2 yrs) or longer, potentially due to high pore pressures maintained in tight shale formations following injection.”

So to conclude, in my opinion this part of the manuscript is currently a bit misleading, despite the fact that the Authors do reject the pore pressure diffusion interpretation in the end. That is not to say that the diffusion curve fitting is obsolete, as in many ways it can be useful to aid the interpretation, but that the Authors’ need to tone down their interpretations and consider other possibilities that are not included here, especially the influence of latent seismicity from previous operations, and take into account some of the points outlined above. The Authors need to include the locations (in one of the figures) and timings of these previous operations in the manuscript. It appears to me that any triggering of the MG08 and southern clusters is not at all related to fluid diffusion, as any triggering of reactivation additional to the ongoing latent seismicity likely takes place either during or very shortly after injection, at timescales much faster than diffusion processes, and there may be other possible mechanisms not currently discussed by the Authors.

Thanks for the detailed explanation and instructions on how to best address the concern about the migration pattern of EHWs. Accordingly, our revision effort is summarized as following:

First, we add the locations and timings of previous operations in the study area during the period between 2013 and 2014, as shown in Fig. 2B.

Second, we introduce the spatial correlation between previous injections and the EHW clusters in Lines 140-146 as “EHWs preceding the July HF stimulation are suspected to be latent seismicity related to previous nearby injection activity (e.g., 25, 26). Specifically, HF injection was conducted at four well pads between 2013 and 2014 (W1–W4 in Fig. 2B). Among them, W1 was operational in September 2013 and is co-located with the monitored well pad of this study. W3 and W4 appear to be close to the cluster near MG08 and the southern cluster, respectively.”

Third, we discuss the possible mechanism of these latent EHWs in Lines 335-344 as “EHWs that occurred before the July 2015 HF injection (mainly from the southern cluster) are likely latent seismicity (Fig. 2C). Given the timing, they cannot be interpreted to have been triggered by pore pressure diffusion or poroelastic stress transfer from injections. Rather, we propose that aseismic slip driven by fluids from prior injections (W1–W4) may play a role. The fluids trapped in fault zones within low-permeability formations could retain a localized, elevated stress state for periods of months to years (26, 53). The altered stress state may help generate aseismic slip to repeatedly load neighboring unstable areas along faults, and thereby lead to latent EHWs/typical induced events occurring at relatively steady rates (25, 26).”

Fourth, we point out that “The localized elevated stress-state scenario may also apply to EHWs that occur behind the migration front (Fig. 2C), as they could be the on-going process of either latent EHWs or earlier triggered events.” (Lines 344-346).

Minor comments (line numbers from tracked changes document):

1. L81: British Colombia -> British Columbia

We correct the typo throughout the manuscript.

2. L295: along -> within

It is revised in Line 269.

Thomas Eyre

References

- Bao, X., Eaton, D.W., 2016. Fault activation by hydraulic fracturing in western Canada. *Science*, 354(6318), 1406-1409.
- Bei Wang, Rebecca M. Harrington, Yajing Liu, Honn Kao, Hongyu Yu; Remote Dynamic Triggering of Earthquakes in Three Unconventional Canadian Hydrocarbon Regions Based on a Multiple-Station Matched-Filter Approach. *Bulletin of the Seismological Society of America* 2018; 109 (1): 372–386. doi: <https://doi.org/10.1785/0120180164>.
- Salvage, R. O. and Eaton, D. W.: Unprecedented quiescence in resource development area allows detection of long-lived latent seismicity, *Solid Earth*, 12, 765–783, <https://doi.org/10.5194/se-12-765-2021>, 2021.

Fig. R1. Wellpads active within 2014 and 2015 in the region. They appear to correlate well with the EHW cluster locations taken from the manuscript (B) (note figure parts not accurately scaled), showing similar clustering to that seen around other operations in other regions (e.g. Bao and Eaton, 2016).

Reviewer #4 (Remarks to the Author):

I appreciate the efforts from the authors, who had done a great job in address my and other reviewers' comments on the previous version. Pleased to find that the authors have addressed all my previous concerns and incorporated my suggestions appropriately. This version can be accepted, with one following minor point that can be easily addressed.

After attributing the coda to the travel path effects, the author interpreted possible heterogeneities along the path due to injection into the shale formation. In Fig. S4, the inferred structural heterogeneities should have their legend, as the color-coded ray path makes the figure pretty busy. A possibly better plot is to show the variation of coda versus back azimuth, which may strengthen the interpretation of heterogeneities along the ray path, in addition to the plot of distance cross section (Fig. S4BCD).

Thanks for the suggestion. We have revised Fig. S4 by (a) adding the legend for two inferred structural heterogeneities on the top right corner, and (b) adding diagrams of coda duration variation versus station-to-epicenter back azimuth to the three distance cross sections respectively (Figs. S4B-D).

REVIEWERS' COMMENTS

Reviewer #1 (Remarks to the Author):

The Authors have done a good job of addressing my and the other Reviewer's comments. I greatly appreciate their efforts through these multiple review rounds. All of my concerns have now been addressed and the manuscript can now be accepted. I just have one very minor suggestion:

For completeness, it may be good to include active periods of injection (month, year) at wells W2-W4, perhaps in a small table for all wells (i.e. W1-W4) in the supplementary materials. Currently, only operations at W1 are properly detailed, with operations at the other wells stated to occur in the broad period 2013-2014.

Thomas Eyre

Manuscript Number: NCOMMS-20-41246

Title: Fluid-injection induced earthquakes characterized by hybrid-frequency waveforms manifest the transition from aseismic to seismic slip

Authors: H. Yu, R. M. Harrington, H. Kao, Y. Liu, and B. Wang

We would first like to thank the reviewer for the positive feedback and further helpful comment to improve the quality of the manuscript. Our point-to-point response to the minor suggestion from Reviewer #1 is given below.

The original reviewer comments are in black, and our responses in blue. A “track-changes” version is also included for your convenience. Please refer to the PDF document of the revised manuscript with tracked changes for the corresponding line numbers mentioned in this response letter.

REVIEWER COMMENTS

Reviewer #1 (Remarks to the Author):

The Authors have done a good job of addressing my and the other Reviewer’s comments. I greatly appreciate their efforts through these multiple review rounds. All of my concerns have now been addressed and the manuscript can now be accepted. I just have one very minor suggestion:

For completeness, it may be good to include active periods of injection (month, year) at wells W2-W4, perhaps in a small table for all wells (i.e. W1-W4) in the supplementary materials. Currently, only operations at W1 are properly detailed, with operations at the other wells stated to occur in the broad period 2013-2014.

Thanks for the suggestion. We now add Table S3 to list the active periods of injections for wells W1–W5, where W5 is the monitored well in this study (now denoted in Fig. 2b).

Thomas Eyre